# Fast training of accurate physics-informed neural networks without gradient descent

**Chinmay Datar**[1,2,3*], **Taniya Kapoor**[4], **Abhishek Chandra**[5], **Qing Sun**[1,3], **Erik Bolager**[1,3],
**Iryna Burak**[1,3], **Anna Veselovska**[1], **Massimo Fornasier**[1,2,3,6], **Felix Dietrich**[1,3,6]
[1]Technical University of Munich (TUM)  [2]TUM Institute for Advanced Study
[3]Munich Center for Machine Learning  [4]Wageningen University & Research
[5]KTH Royal Institute of Technology  [6]Munich Data Science Institute

## Abstract

Solving time-dependent Partial Differential Equations (PDEs) is one of the most critical problems in computational science. While Physics-Informed Neural Networks (PINNs) offer a promising framework for approximating PDE solutions, their accuracy and training speed are limited by two core barriers: gradient-descent-based iterative optimization over complex loss landscapes and non-causal treatment of time as an extra spatial dimension. We present *Frozen-PINN*, a novel PINN based on the principle of space-time separation that leverages random features instead of training with gradient descent, and incorporates temporal causality by construction. On nine PDE benchmarks, including challenges like extreme advection speeds, shocks, and high-dimensionality, Frozen-PINNs achieve superior training efficiency and accuracy over state-of-the-art PINNs, often by several orders of magnitude. Our work addresses longstanding training and accuracy bottlenecks of PINNs, delivering quickly trainable, highly accurate, and inherently causal PDE solvers, a combination that prior methods could not realize. Our approach challenges the reliance of PINNs on stochastic gradient-descent-based methods and specialized hardware, leading to a paradigm shift in PINN training and providing a challenging benchmark for the community.

## 1 Introduction

Partial Differential Equations (PDEs) provide a unifying framework for modeling complex dynamical systems across physics, biology, and engineering, yet developing efficient methods to solve them remains a longstanding challenge (Farlow, 1993). Deep neural networks have recently shown significant promise for approximating solutions of PDEs because of the mesh-free construction of basis functions, high expressivity of neural networks (Rudi & Rosasco, 2021), their ability to represent functions in high dimensions (E, 2020; Wu & Long, 2022; Han et al., 2018), and powerful software for automatic differentiation (e.g., Pytorch (Paszke et al., 2017), TensorFlow (Abadi et al., 2015), DeepXDE (Lu et al., 2021b)). Earlier work on solving PDEs using neural networks (Dissanayake & Phan-Thien, 1994; Lagaris et al., 1998) was recently popularized in the form of Physics-informed neural networks (PINNs) (Raissi et al., 2019; Karniadakis et al., 2021; Sirignano & Spiliopoulos, 2018). PINNs incorporate physical constraints by minimizing a loss function involving the PDE, boundary condition, and initial condition residuals during training. Despite their promise, we identify two root causes limiting the performance of PINNs in terms of accuracy and training time.

**1. Inherent challenges posed by the PINN optimization problem:** Many studies (Wang et al., 2021; 2022) show that even in very simple settings, the PINN loss is quite challenging to minimize using iterative gradient-descent-based optimization methods leveraging the classical back-propagation algorithm (Rumelhart et al., 1986). Krishnapriyan et al. (2021) show that incorporating PDE-based soft constraints into the PINN loss function yields a highly nontrivial loss landscape, rendering optimization particularly challenging. Wang et al. (2022) analyze PINN training dynamics via the Neural Tangent Kernel (NTK) and highlight issues with spectral bias and different convergence rates across different loss components. Rathore et al. (2024) show that differential operators in the PDE

---

*Corresponding author, chinmay.datar@tum.de.

residual loss induce "ill-conditioning", characterized by steep and shallow gradients in different directions near the optimum, complicating the optimization.

Efforts to improve PINN training, such as balancing loss terms (Yao et al., 2023), effective regularization (Lu et al., 2021c; Yu et al., 2022), architectural innovations (Wang et al., 2024b), and improved optimizers (Müller & Zeinhofer, 2023; Liu et al., 2024), have been explored. We assert that such approaches address the symptoms rather than the root cause that makes training PINNs extremely challenging: the PINN optimization problem is high-dimensional (large number of trainable parameters), multi-objective (simultaneous minimization of PDE, and initial and boundary condition losses), and non-convex, with inherently conflicting loss terms (Liu et al., 2024) and further complicated by treating time as an additional dimension in space.

**2. Non-causal treatment of time as an extra spatial dimension:** The temporal structure of initial value PDEs is inherently Markovian as the solution at each subsequent time step depends solely on the solution at the preceding time step. Most PINN-based approaches fail to incorporate *temporal causality* explicitly, and time is treated as an extra dimension in the input layer. This leads to neural bases spanning the entire space-time domain, exacerbating the optimization. Such approaches struggle to capture high-frequency temporal dynamics (Krishnapriyan et al., 2021), and solving PDEs over a long time horizon, without resorting to domain decomposition techniques (Meng et al., 2020).

Previous studies have sought to enforce temporal causality by progressively penalizing residuals in time (Wang et al., 2024d), training distinct models across disjoint intervals with integral-form losses within each interval (Jung et al., 2024), or applying implicit time-differencing with transfer learning to sequentially update PINNs on each interval (Li et al., 2024). Nonetheless, such approaches are difficult to implement, require precise tuning of temporal windows and weight scheduling, and remain computationally demanding (Kim & Son, 2025; Li et al., 2024; Penwarden et al., 2023). See Appendix A for an extended literature review and Appendix B.1 for a detailed discussion on PINNs.

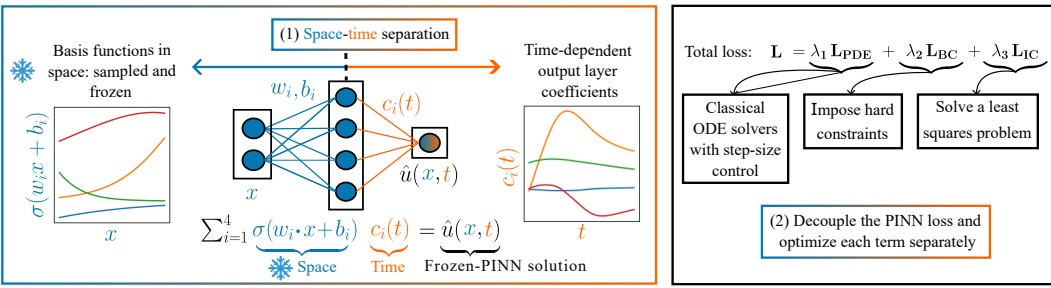

Figure 1: **Core ideas of Frozen-PINNs:** (1) **Space–time separation**: For $x \in \mathbb{R}^d$, spatial bases $\phi_i = \sigma(w_i \cdot x + b_i)$ with $\sigma = \tanh$, $w_i \in \mathbb{R}^d$, $b_i \in \mathbb{R}$ are sampled and frozen (shown for $d = 1$); output layer parameters $c_i(t)$ are evolved via ODEs. Each pair $(\phi_i, c_i)$ is color-matched. (2) **Loss decoupling**: PDE, boundary, and initial condition losses $L_{\text{PDE}}, L_{\text{BC}}, L_{\text{IC}}$ are optimized independently.

To address the root causes of accuracy and training bottlenecks of PINNs rather than the symptoms, we investigate: *How can the PINN optimization problem be simplified while enforcing temporal causality for time-dependent PDEs?* We propose "Frozen-PINN" based on space-time separation —a novel approach that *simplifies the PINN optimization problem* and *enforces temporal causality by construction*. We achieve this by: (a) sampling and freezing space-dependent hidden layer parameters to reduce the dimensionality, (b) decoupling the PINN loss and optimizing each term separately, and (c) computing time-dependent output layer parameters using least squares and adaptive Ordinary Differential Equation (ODE) solvers, replacing gradient-descent-based training (see Figure 1). In Figure 8, we contrast Frozen-PINNs with classical PINNs. Our key contributions are:

1. **Training algorithm:** Frozen-PINNs break the longstanding training and accuracy bottlenecks of PINNs, making PINNs rapidly trainable, temporally causal, and highly accurate, a combination realized for the first time, defining a new state-of-the-art, to our knowledge.

2. **Extensive empirical evaluation:** Across nine challenging PDE benchmarks and rigorous ablation studies, we show that Frozen-PINNs achieve up to 4-5 orders of magnitude faster training than state-of-the-art (SOTA) PINNs, attain high-precision accuracies that are comparable to efficient mesh-based methods in low dimensions, which most SOTA neural PDE

solvers fail to match, and scale efficiently to high-dimensional problems where mesh-based solvers fail.

3. **Adaptive solution-driven network parameters:** We use solution data from previous time-steps to compute efficient neural network parameters. This extends previous work on random feature methods (Bolager et al., 2023) for self-supervised PDE learning tasks.

4. **Model compression:** We introduce an *SVD layer* that reduces the number of neurons in the last hidden layer of the network by up to 20 times and speeds up training up to 75 times.

## 2 SOLVING TIME-DEPENDENT PDES USING FROZEN-PINNS

In this section, we discuss the theoretical details of Frozen-PINNs.

### 2.1 FROZEN-PINN ANSATZ

In this work, we consider time-dependent PDEs on domain $\Omega \subset \mathbb{R}^d$ for space dimension $d$ with boundary $\partial\Omega$, seeking solutions $u : \Omega \times \mathbb{R} \to \mathbb{R}$ of PDEs defined by linear operators $\mathcal{L}$ and $\mathcal{B}$ that only involve derivative operators in space, forcing $f : \Omega \to \mathbb{R}$, boundary $g : \partial\Omega \to \mathbb{R}$, initial condition $u_0 : \Omega \to \mathbb{R}$, and a nonlinear operator $\gamma\mathcal{N}$ for $\gamma \in \mathbb{R}$ ($\gamma = 0$ for linear PDEs):

$$u_t(x,t) \; + \; \mathcal{L}u(x,t) + \gamma\mathcal{N}(u)(x,t) = f(x), \; x \in \Omega, \; t \in [0,T], \tag{1a}$$

where $u_t$ denotes the time derivative of $u$, with boundary and initial conditions given by

$$\mathcal{B}u(x,t) = g(x), \qquad x \in \partial\Omega, \; t \in [0,T], \quad \text{and,} \quad u(x,0) = u_0(x), \quad x \in \Omega, \tag{1b}$$

respectively. We parameterize the approximation of the solution to the PDE (Equation (1)) with a Frozen-PINN having a single hidden layer with $M$ neurons and activation function $\sigma = \tanh$ as

$$\hat{u}(x,t) = C(t)[\Phi(x), \mathbb{1}] = c(t)\sigma(Wx^\top + b) + c_0(t). \tag{2}$$

Here, $c(t) \in \mathbb{R}^{1 \times M}$ and $c_0(t) \in \mathbb{R}$ are time-dependent parameters, $W \in \mathbb{R}^{M \times d}$ and $b \in \mathbb{R}^{M \times 1}$ are space-dependent parameters, and $C := [c, c_0] \in \mathbb{R}^{1 \times (M+1)}$. The activation functions are stacked in $\Phi = [\phi_1, \ldots, \phi_M]$, where $\phi_m(x) = \sigma(w_m x^\top + b_m)$. Note that our approach does not require the PDE solution to be separable in space and time. We next discuss how to sample parameters $W$ and $b$.

### 2.2 COMPUTING HIDDEN LAYER PARAMETERS WITHOUT GRADIENT DESCENT

We sample space-dependent hidden layer parameters in Frozen-PINNs using either ELM or SWIM. Hidden layer parameters are frozen (kept independent of time) after sampling (except Section 3.4).

**ELM (Data-agnostic)**: In the Extreme Learning Machine (ELM) approach (Huang et al., 2006), the weights are sampled from a Gaussian distribution, and biases are sampled from a uniform distribution in $[-\eta, \eta]$ for each hidden layer, where $\eta$ is a hyper-parameter.

**SWIM (Data-dependent)**: The Sample Where It Matters (SWIM) approach follows Bolager et al. (2023) and samples weights and biases using a data-dependent distribution. Each pair $(w_m, b_m)$ is computed using two collocation points $x^{(1)}, x^{(2)} \in \Omega$: $w_m = s_1 \frac{x^{(2)} - x^{(1)}}{\|x^{(2)} - x^{(1)}\|^2}$, $b_m = -\langle w_m, x^{(1)} \rangle + s_2$, where $s_1, s_2 \in \mathbb{R}$ depend on the activation function. In the unsupervised setting, one can choose pairs of collocation points from a uniform distribution over all possible pairs of collocation points, which is the default setting in this paper, as we do not know the solution of the PDE beforehand. In the supervised setting (Section 3.4, Section 3.7), collocation pairs $(x^{(1)}, x^{(2)})$ are sampled with density $\|f(x^{(2)}) - f(x^{(1)})\|/\|x^{(2)} - x^{(1)}\|$. Neuron weights and biases are set so that the $\tanh$ output is $-0.5$ at $x^{(1)}$ and $+0.5$ at $x^{(2)}$, ensuring centers of activations $\tanh$ lie inside the domain and are aligned with the direction $x^{(1)} \to x^{(2)}$, unlike ELM. The suitability of each of the proposed approaches depends on the true PDE solution's gradient distribution. See Appendix B.2.1 for details. In Figure 2, we illustrate the difference between the basis functions sampled with ELM and SWIM.

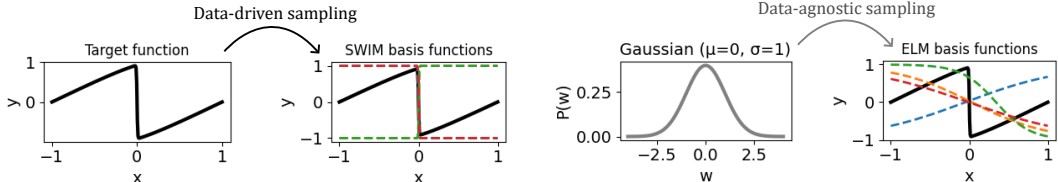

Figure 2: Sampling in Frozen-PINNs: (Left): SWIM (data-driven, places bases with steep gradients near regions with shocks) vs. (Right): ELM (data-agnostic, no control over basis placement).

## 2.3 Solving time-dependent PDEs using Frozen-PINNs by separation of variables

We now discuss the computation of time-dependent output layer parameters $c(t)$. We insert the ansatz (Equation (2)) into the PDE Equation (1a), reformulating it as an ODE for $c(t)$, preserving the inherent causal structure of time-dependent PDEs, thereby enforcing temporal causality by design. We assemble $N_c$ collocation points in $X \in \mathbb{R}^{N_c \times d}$, sample weights and biases of $M$ neurons, compute hidden layer output $\Phi(X)$, and obtain the ODE

$$C_t(t) = R(X, C(t))[\Phi(X), \mathbb{1}]^+, \quad \text{where}$$
$$R(X, C(t)) = -C(t)\mathcal{L}[\Phi(X), \mathbb{1}] - \gamma \mathcal{N}(C(t)[\Phi(X), \mathbb{1}]) + [f(X)]^\top, \tag{3}$$

where $[\Phi(X), \mathbb{1}] \in \mathbb{R}^{(M+1) \times N_c}$ and the pseudo-inverse is denoted by $\cdot^+$. The initial condition is computed via a least squares solution: $C(0) = u(X, 0)^\top[\Phi(X), \mathbb{1}]^+$, which decouples the initial condition loss from PDE and boundary losses, simplifying the optimization problem. We compute $C(t)$ via ODE solvers with step-size control (e.g., RK45 (Dormand & Prince, 1980), LSODA (Petzold, 1983)) instead of gradient descent, and interpolate solutions at test points. See Appendix B.2.2, Appendix B.2.3 for detailed derivations of PDE-to-ODE reformulations for all PDEs considered here.

## 2.4 Approaches for satisfying boundary conditions for Frozen-PINNs

We propose two different strategies to satisfy boundary conditions for Frozen-PINNs: the first utilizes a boundary-compliant layer, and the second augments the reformulated ODE.

**Boundary-compliant layer:** Certain boundary conditions can be enforced via a linear map $A \in \mathbb{R}^{M_b \times M_s}$ ($M_s := M$) applied after the sampled hidden layer, forming a *boundary-compliant layer* (see Figure 3). Defining $\Phi_A := [A\Phi, \mathbb{1}]$ and $C(t) \in \mathbb{R}^{1 \times (M_b+1)}$, we rewrite Equation (3) to

$$C_t(t) = R(X, C(t))\Phi_A(X)^+, \quad \text{where}$$
$$R(X, C(t)) = -C(t)\mathcal{L}\Phi_A(X) - \gamma \mathcal{N}(C(t)\Phi_A(X)) + [f(X)]^\top. \tag{4}$$

Boundary conditions defined by $\mathcal{B}$ and $g$ determine the construction of $A$; see Appendix B.2.4 for details. With a boundary-compliant layer, boundary conditions are satisfied by construction, fully decoupling the PINN loss so that the ODE solver minimizes only the PDE residual. The rationale for outer basis functions is discussed in Appendix B.2.1.

**Augmented ODE:** This strategy eliminates the need for a boundary-compliant layer by augmenting the ODE with a correction term enforcing boundary conditions. For *Dirichlet boundary condition* $u(x) = g(x)$, we add $\hat{u}_t(x) = -\kappa(\hat{u}(x) - g(x))$ for $x \in \partial\Omega$ and solve the augmented system:

$$C_t(t) = \underbrace{[R(X, C(t)), -\kappa(C(t)\Phi_A(X_b) - g(X_b)^\top)]}_{\in \mathbb{R}^{1 \times (N_c + N_b)}} \underbrace{\Phi_A([X, X_b])^+}_{\in \mathbb{R}^{(N_c + N_b) \times (M_b+1)}}, \tag{5}$$

where $\kappa > 0$ is a fixed parameter, $X$ are the $N_c$ collocation points and $X_b \in \mathbb{R}^{N_b \times d}$ is a collection of $N_b$ points on the boundary $\partial\Omega$. For consistency of notation, we set $A = I$ in Equation (4) when using the augmented ODE. In practice, we skip the boundary-compliant layer if we adopt this approach. The intuition behind this technique is that the augmented ODE (Equation (5)) corrects the solution by steering $\hat{u}(x, t)$ toward $g(x)$ for $x \in \partial\Omega$ at rate $\kappa(\hat{u} - g)$, with $\kappa = 10^5$ as a default value. We empirically investigate the effect of $\kappa$ on the boundary loss and the time to solution (see Figure 15). This still partially decouples the PINN loss, with the initial condition treated separately. Depending on the PDE, domain, and boundary type, either strategy can be applied (see Appendix B.2.5).

## 2.5 SVD LAYER

As the last step in the Frozen-PINN architecture, we add a linear layer to reduce the stiffness of the associated ODE (Equation (4)) and the size of the ODE system. To achieve this, we propose orthogonalizing the basis functions using an *SVD layer*. We compute a truncated singular value decomposition of $A\Phi(X) \in \mathbb{R}^{M_b \times N_c}$ to obtain matrices $V_r$, $\Sigma_r$, and $U_r$ with $r \leq M_b$ such that $V_r\Sigma_r U_r^\top = A\Phi(X) + O(\Sigma_{r+1})$. We then define $A_r := V_r^\top A$ and use it instead of the matrix $A$ and $C(t) \in \mathbb{R}^{1 \times (r+1)}$. This ensures $A_r\Phi(X)$ are orthogonal functions on the data $X$, and the matrix $A_r\Phi(X)$ has a bounded condition number. The SVD layer accelerates computation by up to 75 times while reducing the ODE system dimension 20 times, as validated by an extensive ablation study (see Appendix C). Figure 3 visualizes the complete Frozen-PINN architecture.

## 2.6 SUMMARY OF THE TRAINING ALGORITHM FOR FROZEN-PINNS

We summarize our training process in Algorithm 1, where $\epsilon_{SVD}$ is the SVD threshold that governs the SVD-layer width. See Appendix B.2 for additional methodological details, and Appendix B.2.1 for extended discussion on PINN vs. Frozen-PINN training, comparison between sampling strategies, influence of random sampling, rationale for outer bases, and the Kolmogorov $n$-width barrier.

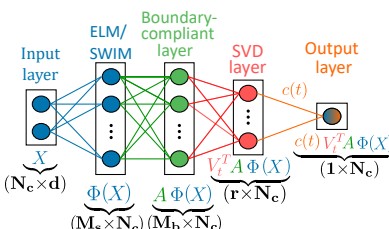

Figure 3: Architecture of Frozen-PINNs trained with a gradient-descent-free training algorithm.

---

**Algorithm 1** Frozen-PINN training algorithm

**Input:** PDE (Equation (1)), test grid points $X_{\text{test}} \times T_{\text{test}}$
**Output:** PDE Solution on the test grid points $\hat{u}(X_{\text{test}}, T_{\text{test}})$
**Parameters:** $N_c, M_s, M_b \in \mathbb{N}, \epsilon_{SVD} \in \mathbb{R}$
1: Sample $N_c$ collocation points: $X \in \mathbb{R}^{N_c \times d}$
2: Construct hidden layer params $\{w_m, b_m\}_{m=1}^{M_s}$ (SWIM/ELM) ▷ Section 2.2
3: Compute hidden layer output $\Phi(X) \in \mathbb{R}^{M_s \times N_c}$
4: Construct *boundary-compliant layer*: $A\Phi(X) \in \mathbb{R}^{M_b \times N_c}$ ▷ Section 2.4
5: Compute truncated SVD: $V_r\Sigma_r U_r^\top = A\Phi(X)$ and *SVD layer* output $V_r^\top A\Phi(X) = A_r\Phi(X)$
6: Compute neural bases: $\Phi_{A_r}(X) := (A_r\Phi(X), 1)^\top \in \mathbb{R}^{(r+1) \times N_c}$
7: Initialize *output-layer* params (least-squares): $C(0) = u(X, 0)^\top \Phi_{A_r}(X)^+$
8: Solve ODE for $C(t) \in \mathbb{R}^{1 \times (r+1)}$ using $\Phi_{A_r}$ ▷ Equation (4)
9: Evaluate $\hat{u}(X_{\text{test}}, T_{\text{test}}) = C(T_{\text{test}})\Phi_{A_r}(X_{\text{test}})$ ▷ Equation (2)

---

## 3 EMPIRICAL RESULTS

In this section, with a comprehensive empirical study across nine challenging low- and high-dimensional PDE benchmarks, we demonstrate that Frozen-PINNs consistently outperform existing state-of-the-art neural PDE solvers with orders-of-magnitude faster training in all cases and higher accuracy in almost all cases without requiring specialized hardware like GPUs. Moreover, our work includes rigorous evaluation against the classical SOTA approaches like IGA-FEM (see Appendix B.3) (Hughes et al., 2005; Cottrell et al., 2006; 2009) or FEM for low-dimensional PDEs, bridging a gap not sufficiently addressed in the literature between neural and mesh-based solvers.

Appendix C contains details of the PDEs, important ablation studies for our experiments (for the SVD layer and the width of the network), metrics used for comparison, train and test data, software and hardware environments, the absolute error plots on test points, and elaborate explanations of results. Figure 12 visually summarizes all the PDE benchmarks used for evaluation, identifies the specific challenges posed by each PDE, and shows true solutions. We perform all experiments with three seeds and report the mean and standard deviation. The code to reproduce experiments from the paper, and a refactored version that will be actively maintained are available at:

> https://gitlab.com/felix.dietrich/swimpde-paper,
> https://gitlab.com/fd-research/swimpde.

To ensure fair comparisons, we follow the two rules outlined by McGreivy & Hakim (2024): (i) we **benchmark at (almost) equal accuracy**, defining low-precision (`1e-2` to `1e-4`) and high-precision (`1e-5` to `1e-10`) regimes, configuring Frozen-PINNs to marginally outperform the best PINN baselines in the low-precision regime and aligning FEM/IGA-FEM fidelity with Frozen-PINNs in the high-precision regime; (ii) we **compare against efficient numerical methods**, including SOTA

IGA-FEM or classical FEM for low-dimensional PDEs, while highlighting neural solvers' scalability in high-dimensional benchmarks where FEM and IGA-FEM suffer from the curse of dimensionality.

### 3.1 HIGH ADVECTION SPEEDS, FAST CONVERGENCE, AND LONG-TIME SIMULATION

We benchmark the linear advection equation to demonstrate how Frozen-PINNs resolve three important well-known challenges for PINNs: (1) handling high advection speeds (Krishnapriyan et al., 2021), (2) achieving fast convergence with increasing width (Cuomo et al., 2022), and (3) long-time simulations (Lippe et al., 2024; Kapoor et al., 2024a). We describe all details in Appendix C.1.

**High advection speeds:** We solve the advection equation for increasing advection coefficients, denoted by $\beta$. Figure 4 (Left) shows that approaches using basis functions in the entire spatiotemporal domain, such as PINNs, ELM, and SWIM, completely fail as the flow velocity $\beta$ increases beyond 40. In contrast, Frozen-PINNs can accurately solve the PDE, even for extremely high values of $\beta$ (as high as $10^4$) with relative $L^2$ errors less than $10^{-4}$. Table 1 shows that for $\beta = 40$, Frozen-PINNs train 45 to 533 times faster than other alternatives at similar accuracy in the low-precision regime. With the exception of Frozen-PINNs, none of the neural PDE solvers evaluated here attain high-precision accuracy. Frozen-PINNs outperform existing neural PDE solvers by over six orders of magnitude in accuracy and approach the fidelity of IGA-FEM, which unsurprisingly is the most accurate solver.

**Fast convergence (error decay with hidden layer width):** For a low value of advection coefficient $\beta = 10$, Figure 4 (Middle) shows that errors with classical PINNs do not decay quickly with width, primarily due to the difficulties in training. In contrast, the relative $L^2$ error decays exponentially with hidden layer width for Frozen-PINNs, ultimately plateauing at a value more than four orders of magnitude smaller than that obtained with PINNs.

**Long-time simulation:** Neural PDE solvers employing joint space–time basis functions, like vanilla PINNs, encounter substantial challenges in accurately approximating dynamics over extended time spans. Here, we consider the advection equation with the advection coefficient $\beta = 1$. As shown in Figure 4 (Right), Frozen-PINNs can simulate the advection equation for 1000 seconds with a relative $L^2$ error under $0.001\%$ in just 0.94 seconds.

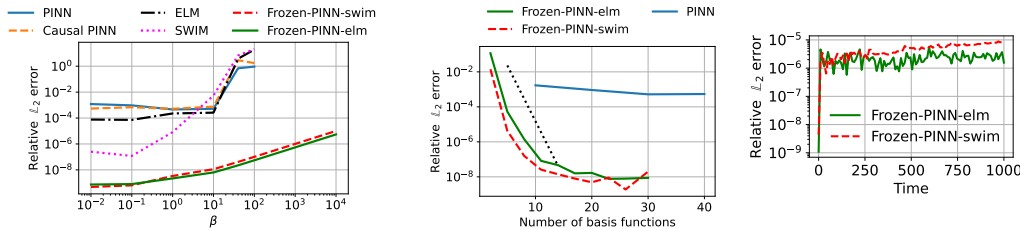

Figure 4: Illustration of experimental results for the advection equation: **(Left): high advection speeds** - effect of advection coefficient $\beta$ on the test error for different PDE solvers, **(Middle): fast convergence** - with $\beta = 10$, Frozen PINNs achieve exponential decay in test error as indicated by the reference dotted line, while standard PINNs display plateaued error decay despite increasing number of basis functions (hidden layer size), **(Right): long time simulation** - Slow error growth with time.

### 3.2 HIGHER-ORDER DERIVATIVES IN SPACE AND TIME

We consider two variants of the Euler-Bernoulli beam equation —classical Euler-Bernoulli beam equation and its extension with a Winkler foundation. See Appendix C.2 for details. The main challenge posed by both PDEs for PINNs is the higher-order differential terms (fourth- and second-order derivatives in space and time, respectively). Frozen-PINNs eliminate expensive evaluation of higher-order derivatives via backpropagation, cutting training cost by four orders of magnitude in the low-precision regime, while achieving IGA-FEM–level accuracy that is more than six orders of magnitude accurate compared to other SOTA PINN benchmarks considered here (see Table 1).

### 3.3 MULTI-SCALE SOLUTIONS

To demonstrate the capability of our method to solve PDEs with multi-scale solutions, we consider a Wave equation benchmark (Hao et al., 2024). We examine two settings: one with two distinct frequencies and another with three well-separated frequencies, which increases the spatial complexity and significantly broadens the range of scales in the solution. For the two-frequency setup, we compare the performance against prior PINN baselines in Table 1 and observe that CPU-trained Frozen-PINNs achieve 625 to 5500 times faster training than GPU-trained competing PINN variants, while simultaneously being four to five orders of magnitude more accurate. Frozen-PINNs also solve the wave equation in the three-frequency scenario (illustrated in Figure 19) extremely quickly and with high precision, reinforcing their potential for solving PDEs with complex, multiscale dynamics. Additional implementation details and extended results are provided in Appendix C.3.

### 3.4 NON-LINEARITY AND SHOCKS

In this example, we highlight how using pairs of data points to sample neural basis functions using the SWIM algorithm can be leveraged to resolve locally steep gradients in the solution of the non-linear viscous Burgers' equation, as shown in Figure 2 (Left). See Appendix C.4 for details.

Frozen-PINN-swim creates numerous basic functions with steep gradients, accurately placing them near the location of the shock, leveraging the SWIM algorithm and solutions from previous time-steps to fit neural basis functions, given enough collocation points in the domain's center (see Figure 20a (Left)). To concentrate collocation points near the shock in the domain's center, we resample them periodically after a set number of time steps, guided by a probability distribution that leverages the gradient of the approximate solution (see Figure 5a (Top)). At the resampling time $t_r \in [0, T]$, we approximate the probability density $p(x) \sim |\nabla \hat{u}(x, t_r)|$, which we then use to re-sample collocation points as illustrated in Figure 5a (Bottom), placing more collocation points near the shock region.

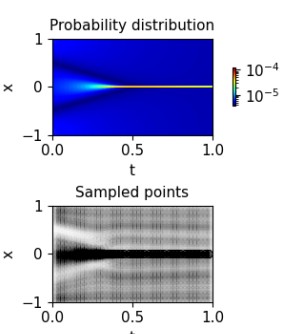

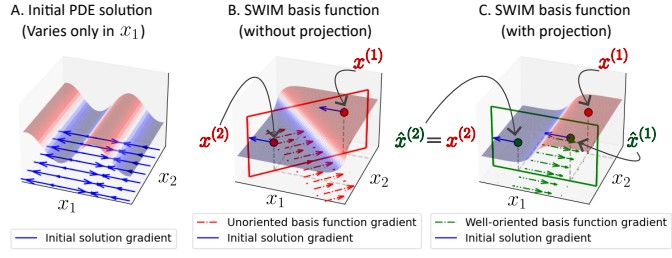

(b) Illustration of embedding directional information to orient Frozen-PINN-swim basis functions along the gradient of the initial condition: (Left): A toy initial condition $u_0 : \mathbb{R}^2 \to \mathbb{R}$ varying in a single direction, (Middle): A randomly selected pair of points $(x^{(1)}, x^{(2)})$ leads to a SWIM basis function misaligned with the gradient of the initial solution, (Right): A projected pair of points $(\hat{x}^{(1)}, \hat{x}^{(2)})$ yields a basis function aligned with the gradient of $u_0$.

(a) (Top): Probability distribution, (Bottom): Sampled collocation points.

Figure 5: Constructing useful Frozen-PINN-swim bases. (Left): shock-aware sampling (Burgers, (Section 3.4)) and (Right): direction-aware bases (reaction–diffusion, Section 3.7).

As shown in Table 1, Frozen-PINNs achieve 46 to 2945 times speedups in training time over other PINN variants in the low-precision regime. Remarkably, even in the high-precision regime, CPU-trained Frozen-PINNs remain 203 to 535 times faster than state-of-the-art GPU-trained PINNs at comparable accuracy. While optimizers like SSBroyden (Kiyani et al., 2025) can offer higher accuracy, they are extremely slow, resource-intensive, and difficult to implement. Furthermore, Frozen-PINN-swim basis functions handle shocks significantly better than Fourier or Chebyshev bases used in classical spectral methods (see Figure 22, Figure 23, Appendix C.4.1).

### 3.5 NON-LINEARITY AND COMPLICATED DOMAIN GEOMETRY

In this example, we consider a non-linear diffusion equation on a complicated domain geometry. See Appendix C.5 for details. For mesh-based methods, meshing can be resource-intensive and technically

demanding (see Figure 24), unlike neural PDE solvers. As shown in Table 1, Frozen-PINNs are 145 to 456 times faster than PINNs and 4.83 times faster than FEM at comparable low-precision accuracy, and can achieve over 1000 times better accuracy than other PINNs. Notably, Frozen-PINNs require only 350 basis functions versus around 2000 finite elements in FEM for similar accuracy (see Table 18), mainly due to the global support of neural bases. For fairness, the FEM grid points are reused as collocation points for minimizing the PDE residual in Frozen-PINNs.

### 3.6 CHAOS AND STRONG NON-LINEARITY

We tackle the highly nonlinear Kuramoto-Sivashinsky equation, which models laminar flame-front instabilities that exhibit spatiotemporal chaos. As shown in Figure 6, our Frozen-PINN captures the characteristic chaotic pattern over a long-time horizon $t \in [0, 5]$, with an average training time of only 6.9 seconds on CPU (averaged over 5 seeds). Further experimental details are provided in Appendix C.7. Since chaotic dynamics amplify small numerical differences, trajectory-level errors are not meaningful, and we assess performance based on the qualitative spatiotemporal patterns.

### 3.7 HIGH-DIMENSIONAL PDES WITH LOW-DIMENSIONAL SOLUTION MANIFOLDS

In this benchmark (Zang et al., 2020), we solve a five-dimensional non-linear reaction-diffusion equation, where the solution only changes in two dimensions that are a priori unknown. We construct SWIM basis functions aligned with the two intrinsic dimensions of variation, directly embedding directional information unlike in PINNs and ELMs, by using spatial coordinates projected onto the gradient of the initial solution to sample SWIM basis functions, as shown in Figure 5b. See Appendix C.6 for further details.

Table 1 shows that Frozen-PINN-swim is over 3400 times faster than other PINNs at comparable low-precision accuracy. It is the only method to reach the high-precision regime, achieving 2–3 orders of magnitude higher accuracy than other PINN variants and weak adversarial networks (Zang et al., 2020). These results confirm that explicitly embedding informative basis functions yields far more efficient and accurate models than relying on iterative optimization to learn them implicitly.

### 3.8 HIGH-DIMENSIONALITY

High-dimensional PDEs, such as the 100-dimensional heat equation, are computationally prohibitive for grid-based methods, which require more than $10^{30}$ grid points, considering only two points per dimension. The following examples demonstrate Frozen-PINNs' ability to solve such PDEs efficiently and accurately. We evaluate our approach on two established benchmarks: one introduced in Wang & Dong (2024), which addresses the heat equation in up to 10 dimensions on a unit hypercube, and another introduced in He et al. (2023), which focuses on a 100-dimensional variant of the heat equation on a unit ball. We discuss all details in Appendix C.8.

Frozen-PINN-elm is consistently 10–1000 times more accurate than classical PINNs for up to 100-dimensional PDEs Figure 7 (top), with error decaying rapidly with network width until saturation Figure 7 (bottom). For the 10-d heat equation, Frozen-PINN-elm trains $100 - 1000$ times faster than other PINNs while achieving higher accuracy. For the 100-d heat equation, CPU-trained Frozen-PINNs remain hundreds of times faster than GPU-trained PINNs while delivering an order-of-magnitude better accuracy (Table 1), underscoring both their computational efficiency and high accuracy. Table 2 summarizes the advantages of our algorithm over classical mesh-based and physics-informed methods based on iterative gradient-descent-based methods.

## 4 CONCLUSION

Frozen-PINNs directly address the longstanding training and accuracy bottlenecks of PINNs by fundamentally simplifying the optimization problem and enforcing temporal causality by construction, leveraging the idea of space-time separation. Our extensive empirical analysis reveals that Frozen-PINNs consistently realize extremely fast training and high precision (often several orders of magnitude better than SOTA PINNs), and preserve temporal causality on a broad range of PDEs involving challenges such as extreme flow velocities, long-time simulation, higher-order spatial and temporal derivatives, complicated spatial domains, non-linearities, shocks, and high-dimensionality,

Table 1: **Summary of empirical results on eight PDE benchmarks**, including results from prior works: dashes denote training times not reported in prior works; Training times labeled with $^+$ were obtained using GPUs; thus, CPU-based training, as with Frozen-PINNs, would lead to substantially larger values. For each PDE, solvers above/below the horizontal line correspond to low-/high-precision regimes. Normalized training times relative to Frozen-PINNs are computed as the ratio of each method's training time to that of Frozen-PINNs, and are computed at similar accuracy.

| PDE benchmark | Method | Training time (s) | Normalized training time | Relative $L^2$ error |
|---|---|---|---|---|
| **Advection** ($\beta = 40$) | PINN (Adam) | - | - | Fail for $\beta$=40 |
| | SWIM | - | - | Fail for $\beta$=40 |
| | ELM | - | - | Fail for $\beta$=40 |
| | Causal PINN | 357.63 | 533 | 2.90e0 $\pm$ 1.2e0 |
| | PINN (L-BFGS) | 30.5 | 45.5 | 6.92e-1 $\pm$ 2.96e-2 |
| Krishnapriyan et al. (2021) | PINN (seq2seq, L-BFGS) | - | - | 2.41e-1 |
| Krishnapriyan et al. (2021) | PINN (Curriculum training, L-BFGS) | - | - | 5.33e-2 |
| | **Frozen-PINN-elm (our)** ✳ | **0.67** | 1 | **4.19e-3 $\pm$ 2.97e-3** |
| | Frozen-PINN-swim (our) ✳ | 0.7 | 1 | 8.42e-9 $\pm$ 1.12e-8 |
| | *Mesh-based method (IGA)* | **0.07** | 0.1 | **1.17e-10** |
| **Euler-Bernoulli (classical)** | PINN (Adam) | 4209.82 | 84196 | 3.95e-2 $\pm$ 1.79e-2 |
| Kapoor et al. (2023) | PINN (L-BFGS) | 2303.71 | 46074 | 4.21e-3 $\pm$ 9.56e-4 |
| | **Frozen-PINN-elm (our)** ✳ | **0.05** | 1 | **2.82e-4 $\pm$ 2.15e-4** |
| | *Mesh-based method (IGA)* | 0.94 | 0.13 | 4.21e-7 |
| | **Frozen-PINN-elm (our)** ✳ | 6.90 | 1 | **9.33e-9 $\pm$ 4.36e-9** |
| **Euler-Bernoulli (Winkler)** | PINN (L-BFGS) | $1858^+$ | $37160^+$ | 5.33e+0 |
| Kapoor et al. (2024b) | Adaptive PINN | 3807.89 | 76140 | 5.32e+0 |
| Kapoor et al. (2024b) | Self-adaptive PINN | 4042.57 | 80840 | 5.15e+0 |
| Kapoor et al. (2024b)) | Wavelet PINN | 4764.25 | 95280 | 4.38e+0 |
| Kapoor et al. (2024b) | Causal PINN | $1873^+$ | $37460^+$ | 3.00e-2 |
| | **Frozen-PINN-elm (our)** ✳ | **0.05** | 1 | **1.41e-2 $\pm$ 4.19e-3** |
| | Frozen-PINN-swim (our) ✳ | 2.41 | 1 | 1.42e-7 $\pm$ 1.20e-7 |
| | *Mesh-based method (IGA)* | **1.08** | 0.44 | **2.70e-8** |
| **Wave** Hao et al. (2024) | PINN (FBPINN) | $3090^+$ | $5517.9^+$ | 5.91e-1 $\pm$ 4.74e-2 |
| Hao et al. (2024) | PINN (L-BFGS) | $350^+$ | $625^+$ | 5.88e-1 $\pm$ 9.63e-2 |
| Hao et al. (2024) | PINN (gPINN) | $775^+$ | $1383.9^+$ | 5.56e-1 $\pm$ 1.67e-2 |
| Hao et al. (2024) | PINN (NTK) | $840^+$ | $1500^+$ | 9.79e-2 $\pm$ 7.72e-3 |
| | **Frozen-PINN-elm (our)** ✳ | **0.56** | 1 | **1.81e-6 $\pm$ 1.01e-6** |
| **Burgers** | Causal PINN | 1531.79 | 2945.75 | 1.60e-2 $\pm$ 8.97e-3 |
| | PINN (L-BFGS) | 275.2 | 529.2 | 3.88e-3 $\pm$ 2.61e-3 |
| Kiyani et al. (2025) | PINN (BFGS with trust region) | $24^+$ | $46.1^+$ | 1.1e-3 |
| | **Frozen-PINN-swim (our)** ✳ | **0.52** | 1 | **1.00e-3 $\pm$ 1.13e-3** |
| Chen et al. (2024b) | PINN (residual-based attention) | - | - | 8.22e-4 $\pm$ 2.33e-4 |
| McClenny & Braga-Neto (2023) | Self-adaptive PINN | - | - | 4.80e-4 $\pm$ 1e-4 |
| Chen et al. (2024b) | PINN (balanced residual decay rate) | - | - | 1.38e-4 $\pm$ 0.85e-4 |
| Kiyani et al. (2025) | PINN (RAdam + BFGS) | 1070 | $203^+$ | 6e-6 |
| Urbán et al. (2025) | PINN (SSBroyden) | - | - | 2.9e-6 $\pm$ 0.4e-6 |
| | **Frozen-PINN-swim (our)** ✳ | **5.25** | 1 | 2.27e-7 $\pm$ 6.89e-8 |
| | *Mesh-based method (IGA)* | 76.32 | 14.5 | 1.12e-7 |
| Kiyani et al. (2025) | **PINN (Adam + SSBroyden)** | $2812^+$ | $535^+$ | **1.62e-8** |
| **Nonlinear diffusion** | PINN (Adam) | 81.36 | 145.2 | 2.09e-2 $\pm$ 3.14e-3 |
| | PINN (L-BFGS) | 255.9 | 456.9 | 1.22e-2 $\pm$ 2.38e-4 |
| | *Mesh-based method (FEM)* | 2.71 | 4.83 | 2.68e-3 |
| | **Frozen-PINN-elm (our)** ✳ | **0.56** | 1 | **2.60e-3 $\pm$ 1.61e-3** |
| | **Frozen-PINN-swim (our)** ✳ | 423 | - | **2.00e-6 $\pm$ 1.99e-6** |
| **5-d Reaction diffusion** | PINN (Adam) | 171.43 | 3428.6 | 3.40e-1 $\pm$ 1.79e-2 |
| | PINN (L-BFGS) | 183.38 | 3667.6 | 3.33e-2 $\pm$ 1.54e-2 |
| Zang et al. (2020) | Weak Adversarial Network | - | - | 2.8e-2 |
| | **Frozen-PINN-swim (our)** ✳ | **0.05** | 1 | **1.07e-2 $\pm$ 4.52e-4** |
| | **Frozen-PINN-swim (our)** ✳ | 12.43 | - | **9.99e-5 $\pm$ 6.21e-9** |
| **10-d heat** | PINN (Adam) | 1002.49 | 3037.8 | 1.68e-1 $\pm$ 3.21e-2 |
| Wang & Dong (2024) | PINN (L-BFGS) | 189.6 | 574.5 | 6.06e-4 $\pm$ 1.00e-4 |
| benchmark extended to $d = 10$ | **Frozen-PINN-elm (our)** ✳ | **0.33** | 1 | **4.35e-4 $\pm$ 5.91e-5** |
| | **Frozen-PINN-elm (our)** ✳ | 168.6 | - | **2.28e-5 $\pm$ 2.1e-5** |
| **100-d heat** He et al. (2023) | PINN (Adam) | $141^+$ | $1084.6^+$ | 0.60e-2 |
| He et al. (2023) | PINN (no stacked-backpropagation) | $49.8^+$ | $383.1^+$ | 0.63e-2 |
| | PINN (Adam+L-BFGS) | $26.25^+$ | $201.9^+$ | 4.98e-3 $\pm$ 2.96e-4 |
| | **Frozen-PINN-elm (our)** ✳ | **0.13** | 1 | **4.12e-4 $\pm$ 1.70e-5** |

without requiring specialized hardware like GPUs. Frozen-PINNs maintain high precision over long time spans and capture high-frequency temporal dynamics where prior neural PDE solvers fail.

Within the scope of the empirical study in this work, in low dimensions, Frozen-PINNs match the accuracy of classical mesh-based solvers while retaining advantages such as mesh-free basis functions, ease of implementation, the ability to handle complex domains, spectral convergence for PDEs with smooth solutions, and scalability for high-dimensional PDEs where mesh-based approaches struggle.

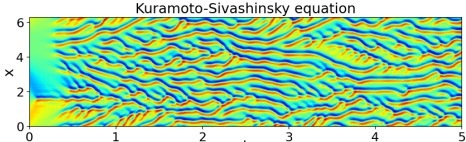

Figure 6: Frozen-PINN solution of the strongly non-linear and chaotic Kuramoto-Sivashinsky equation.

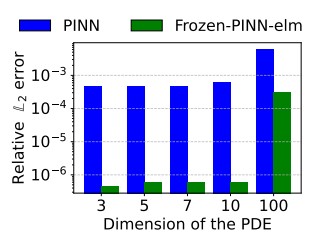

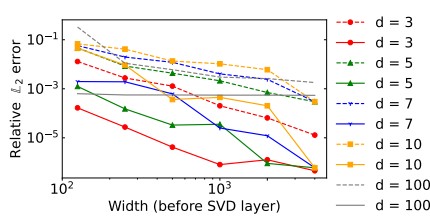

| PDE setting | IGA-FEM/ FEM | PINNs | Frozen-PINNs ❄ |
|---|---|---|---|
| Solutions with shocks | ✔ | ✔ | ✔ (SWIM) |
| Complex domains | mesh | Easy | Easy |
| High dimensionality | ✗ (CoD) | ✔ | ✔ |
| **Performance/features** | | | |
| Accuracy/Precision | High | Often low | High |
| Speed | Fast | Slow (training) | Fast |
| Temporal causality | ✔ | ✗ (soft constraint) | ✔ |

Table 2: Comparison of Frozen-PINNs with mesh-based FEM and classical PINNs in different problem settings presented in this paper: The comparison is grounded in results reported in Section 3 for the PDEs and solvers studied. ✔ denotes compatibility, and ✗ denotes either incompatibility or the need for substantial modifications. Curse of Dimensionality is abbreviated as CoD.

Figure 7: High-dimensional heat equation: (Top): comparison of test errors for varying PDE dimensions (different hatch patterns indicate different benchmarks), (Bottom): fast decay of test error with network width (dashed: Frozen-PINN-swim, solid: Frozen-PINN-elm).

**Limitations and future work:** Our method assumes knowledge of the PDE, but its speed makes it well-suited for inverse problems via fast forward solves. While Frozen-PINNs efficiently deal with extreme temporal complexity, as shown in the advection equation with extreme flow velocities, dealing with spatial complexity encountered while solving PDEs like Navier–Stokes is an exciting next step, where one could leverage domain decomposition to deal with the added complexity (Moseley et al., 2023; Howard et al., 2024). Finally, universal approximation properties concerning specific PDE settings and understanding the role of re-sampling network parameters in overcoming the Kolmogorov n-width barrier (Peherstorfer, 2022) are some of the most challenging, yet important theoretical open areas of investigation, beyond the scope of this paper.

Frozen-PINNs take a decisive step toward practical neural PDE solvers through a lightweight optimization process and extremely fast training without GPUs, promoting low-carbon AI development (Verdecchia et al., 2023), advancing state-of-the-art performance, and establishing a formidable benchmark for the community to build upon in advancing rapid and accurate neural PDE solvers.

**Reproducibility statement:** The source code used to reproduce the experimental results, along with comprehensive reproducibility instructions, is included in the supplementary material and publicly available as an open-source repository. All experiments are run with multiple seeds, and the corresponding seed values are stored in the repository to ensure reproducibility.

**Ethics statement:** Neural networks are inherently dual-use technologies, and ethical considerations are essential for any new machine learning approach. Frozen-PINNs are grounded in classical scientific computing principles, which offer well-understood behavior and interpretability. By bridging neural PDE solvers with classical numerical methods, our framework enables clearer analysis of robustness, failure modes, and reproducibility. We believe this transparency reduces the risk of misuse and enhances controllability, making Frozen-PINNs safe and interpretable. Thus, we believe that the benefits of our approach far outweigh the potential downsides of misuse because a system that is better understood can also be controlled more straightforwardly.

ACKNOWLEDGMENTS

We are grateful to Tim Bürchner for providing an initial version of the IGA code and to Jana Huhne for providing an initial version of the Frozen-PINN code. We sincerely thank Dinesh Parthasarathy, Ana Cukarska, and Adwait Datar for the detailed feedback on the manuscript. Their careful review and suggestions greatly strengthened the manuscript. We further express our appreciation to Wil Schilders for engaging discussions and valuable perspectives. F.D., I.B., and E.B. acknowledge funding by the German Research Foundation—project 468830823, and association with DFG-SPP-229. C.D. is partially funded by the Institute for Advanced Study (IAS) at the Technical University of Munich. F.D. and Q.S. are supported by the TUM Georg Nemetschek Institute - Artificial Intelligence for the Built World. F.D. and C.D. acknowledge Sølve Eidnes and Martine Mahlum for a great workshop on physics in machine learning in Oslo, 2024, which provided us with good feedback on the work.

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

APPENDIX

CONTENTS

## A    EXTENDED REVIEW OF RELATED WORK

In this section, we provide a comprehensive extended review of the literature and highlight how it relates to our work.

**Physics-informed neural networks** are widely used to solve PDEs with neural networks. In this work, we benchmark our approach against various PINN variants such as adaptive activation PINNs (Jagtap et al., 2020), self-adaptive PINNs (McClenny & Braga-Neto, 2023), wavelet PINNs (Uddin et al., 2023), and causal PINNs (Wang et al., 2024c), among others. For high-frequency temporal variations in the PDE solutions, Krishnapriyan et al. (2021) propose curriculum learning with gradually increasing advection coefficients. Compared to curriculum learning, our approach with space-time separation is much easier to implement, computationally efficient, and accurate, as we demonstrate in Section 3.1. Subramanian et al. (2023) propose using adaptive self-supervision of PINNs for sampling collocation points using the gradient of the loss function. We instead use the solution gradient to capture locally sharp features in the solution (see Section 3.4). Many specialized approaches based on PINNs (Cho et al., 2024; Meng et al., 2020; Sharma & Shankar, 2022; Chiu et al., 2022), methods based on hash-encoding (Huang & Alkhalifah, 2024; Wang et al., 2024a), and transfer learning (Kapoor et al., 2024b) have been proposed, but are still based on gradient-based iterative optimization and back-propagation, unlike ours.

Other recent advances of PINNs include methods that model the PDE system as pseudo-sequences. For instance, PINNsFormer employs a Transformer-based architecture that constructs pseudo-sequences from spatio-temporal samples and uses self-attention to model long-range temporal dependencies (Zhao et al., 2024). Another work, PINNMamba, is based on State Space Models (SSMs) and sub-sequence alignment, enabling continuous–discrete temporal modeling and improved propagation of initial-condition information (Xu et al., 2025). Although these methods model PDE systems as pseudo-sequences, these architectures often lead to more computational time and out-of-memory issues owing to their architecture, as presented by Xu et al. (2025).

**Physics-informed approaches using randomized neural networks** for solving PDEs have mostly been studied by combining Extreme Learning Machines (ELMs) with the self-supervised setting of PINNs (Chen et al., 2024a; Wang & Dong, 2024; Shang & Wang, 2024; Sun et al., 2024). For instance, Dwivedi & Srinivasan (2020) propose a physics-informed extreme learning machine (PIELM) to efficiently solve linear PDEs, while Calabrò et al. (2021); Galaris et al. (2022) employ ELMs to learn invariant manifolds as well as PDEs from data. Dong & Yang (2022) show that given a fixed computational budget, ELMs achieve substantially higher accuracy compared to classical second-order FEM and slightly higher accuracy compared to higher-order FEM. For static, nonlinear PDEs, ELMs can be used together with nonlinear optimization schemes (Fabiani et al., 2021). On larger spatiotemporal domains, Dong & Li (2021) and Dwivedi et al. (2021) propose using multiple distributed ELMs on multiple subdomains. Although the aforementioned methods simplify the optimization problem by randomly sampling hidden layer parameters and fixing them, they treat time as merely another spatial dimension. As a result, their neural basis functions span the full spatiotemporal domain, which limits their accuracy on PDEs exhibiting high-frequency temporal dynamics, unlike our approach.

While the problem setting is restricted to Hamiltonian systems, Rahma et al. (2024; 2025) discuss how to train Hamiltonian neural networks and Hamiltonian graph neural networks using ELM and SWIM approaches, and demonstrate how random sampling can be leveraged to significantly speed up training compared to gradient-based iterative optimization. In this work, we show how random sampling can speed up training and resolve optimization challenges of PINNs for time-dependent PDEs.

**Neural Galerkin schemes** (Finzi et al., 2023; Aghili et al., 2024; Berman et al., 2024; Bruna et al., 2024) offer an alternative to the full spatiotemporal approach of the randomized neural networks and PINNs. These approaches treat all or sparse subsets of network parameters, beyond just the last layer's parameters, as time-dependent. This leads to a much larger system of ODEs compared to our approach. The work on neural implicit representations (Chen et al., 2023; Yin et al., 2023) also uses neural basis functions to represent only the space component, but relies on gradient-based iterative optimization via back-propagation, unlike our approach.

**Spectral methods for solving PDEs** promise fast convergence with much fewer basis functions. Meuris et al. (2023) present a method to extract hierarchical spatial basis functions from a trained DeepONet and employ it in a spectral method to solve the given PDE. Xia et al. (2023) integrate adaptive techniques into PINN-based PDE solvers to obtain numerical solutions of unbounded domain problems that standard PINNs cannot efficiently approximate. Lange et al. (2021) propose spectral methods that fit linear and nonlinear oscillators to data and facilitate long-term forecasting of temporal signals. Dresdner et al. (2022) demonstrate spectral solvers that provide sub-grid corrections to classical spectral methods to improve their accuracy. Du et al. (2023) use fixed orthogonal bases to learn PDE solutions as a map between spectral coefficients and introduce a training strategy based on spectral loss. These methods differ from ours in problem setting, architecture, and training.

**Neural operator frameworks** (Lu et al., 2021a; Kovachki et al., 2021; Li et al., 2020; Pfaff et al., 2021) are promising but are typically trained with PDE solutions with different initial conditions, spatial domains (geometries), or parameter settings. Datar et al. (2025) have demonstrated how continuous-time neural networks can be constructed for linear operator approximation for linear and time-invariant systems. Instead, in our setting here, we solve the PDE using given coefficients, domain, and initial conditions without relying on any training data. The ease of implementation, rapid training, and high accuracy of our backpropagation-free approach can be leveraged to generate PDE solution data for training operator networks.

**Mesh-free methods** are typically based on radial basis functions (RBFs, (Powell, 1992; Chen et al., 2014)) or Moving Least Squares (MLS) (Shepard, 1968; Lancaster & Salkauskas, 1981). These often do not have user-friendly software or are only applicable in specialized settings (e.g., smoothed particle hydrodynamics, (Lucy, 1977; Gingold & Monaghan, 1977; Shadloo et al., 2016)). Moreover, despite the ease of dealing with complicated geometries, these methods typically suffer from many challenges, such as the choice of kernel, imposing boundary conditions, and convergence issues. These methods are not the focus of this work.

**Classical numerical methods** such as finite elements, finite volumes, and finite differences have been used to solve PDEs for decades. They often have a rich theoretical grounding and high accuracy. Isogeometric analysis (IGA) is one such method, in which spline-based basis functions are defined over a structured grid (Hughes et al., 2005; Cottrell et al., 2009; 2006). Mesh-based methods often entail a time-consuming setup phase, especially when mesh generation is challenging. Methods like sparse grids enable adaptivity through hierarchical bases but pose significant implementation challenges, particularly for irregular domains (Bungartz & Griebel, 2004). In this work, we benchmark our results against IGA and finite-element-based methods.

## B    Supplementary methodological details on PDE solvers

### B.1    Physics-Informed Neural Networks

This work benchmarks Frozen PINNs against many prominent variants of physics-informed neural networks. While we directly report results from other works for many PINN variants for different PDE benchmarks (see Table 1), we also implement two PINN variants for certain PDEs - classical physics-informed neural network (PINN) (Raissi et al., 2019) and causality-respecting physics-informed neural network (causal PINN) (Wang et al., 2024c). We now describe these two variants.

Classical PINNs are feedforward deep neural networks designed to approximate PDE solutions by incorporating physical laws into the learning process. The architecture of a vanilla PINN includes a deep neural network that maps inputs (e.g., space and time coordinates) to outputs (e.g., physical quantities of interest) and is trained to minimize a loss function that combines data and physics-based errors. The data term ensures that the neural network fits the provided data points, while the physics term enforces the PDE constraints with automatic differentiation. The constraints on initial and boundary conditions are satisfied via additional loss terms. The loss function for a classical PINN is:

$$L(\mu) = \lambda_1 L_{\text{PDE}}(\mu) + \lambda_2 L_{\text{IC}}(\mu) + \lambda_3 L_{\text{BC}}(\mu) + \lambda_4 L_{\text{Data}}(\mu), \tag{6}$$

where $\mu$ represents the trainable network parameters, and $\lambda_i$, for $i = 1, 2, 3, 4$ represent the weighting factors for individual loss terms, which are hyperparameters. In this work, we consider the setting of unsupervised learning and thus neglect the data loss term.

Let $N$ be the total number of training points, which is the sum of the number of interior training points $N_{\text{int}}$ (where the PDE residual is evaluated), initial condition training points $N_{\text{ic}}$ (where the initial condition is evaluated), and boundary condition training points $N_{\text{b}}$ (where the boundary condition is evaluated). We denote the neural network solution at a point $(x^{(n)}, t^{(n)})$ in the computational domain by $u^*(x^{(n)}, t^{(n)})$. We consider the generic nonlinear PDE defined by equation 1. The PDE loss term is defined by

$$L_{\text{PDE}}(\mu) = \frac{1}{N_{\text{int}}} \sum_{n=1}^{N_{\text{int}}} ||u_t^*(x^{(n)}, t^{(n)}) + Lu^*(x^{(n)}, t^{(n)}) + \lambda N(u^*)(x^{(n)}, t^{(n)}) - f(x^{(n)})||^p. \quad (7)$$

The boundary condition loss term is defined as

$$L_{\text{BC}}(\mu) = \frac{1}{N_{\text{b}}} \sum_{n=1}^{N_{\text{b}}} ||Bu^*(x^{(n)}, t^{(n)}) - g(x^{(n)})||^p. \quad (8)$$

Similarly, the initial condition loss term is defined as

$$L_{\text{IC}}(\mu) = \frac{1}{N_{\text{ic}}} \sum_{n=1}^{N_{\text{ic}}} ||u_0^*(x^{(n)}) - u_0(x^{(n)})||^p. \quad (9)$$

We now describe a Causal PINN, which modifies the PINN loss function to impose temporal causality, inherent in time-dependent PDEs, as a soft constraint. In conventional PINNs, the loss is computed without prioritizing accuracy at earlier times, which disrupts temporal causality. The Causal PINN remedies this by assigning weights at each time step based on the cumulative loss from previous steps, ensuring that the model concentrates on accurately approximating solutions at earlier times before moving forward. This tries to incorporate the causal structure of the physical problem being solved as a soft constraint. The causal PDE loss term is defined by

$$L_{\text{PDE}}(\mu) = \sum_{i=1}^{N_{\text{t}}} w_i L_{\text{PDE}}(t_i, \mu), \quad \text{where}$$

$$w_1 = 1, \quad w_i = e^{-\epsilon \sum_{k=1}^{i-1} L_{\text{PDE}}(t_k, \mu)}, \quad \text{for } i = 2, 3, \ldots N_{\text{t}}. \quad (10)$$

Here $N_{\text{t}}$ represents the number of time steps into which the computational domain is divided. The causality hyperparameter $\epsilon$ regulates the steepness of the weights and is incorporated in the loss function, similar to Kapoor et al. (2024b). This modification introduces a weighting factor $w_i$ for the loss at each time level $t_i$, with $w_i$ being dependent on the cumulative PDE loss up to time $t_i$. The network prioritizes a fully resolved solution at earlier time levels by exponentiating the negative of this accumulated loss. Consequently, the modified PDE loss term for a causal PINN is expressed as

$$L_{\text{PDE}}(\mu) = \frac{1}{N_t} \left[ w_1 L_{\text{PDE}}(t_1, \mu) + \sum_{i=2}^{N_t} e^{-\epsilon \sum_{k=1}^{i-1} L_{\text{PDE}}(t_k, \mu)} L_{\text{PDE}}(t_i, \mu) \right]. \quad (11)$$

## B.2 FROZEN-PINN-SWIM AND FROZEN-PINN-ELM

### B.2.1 EXTENDED DISCUSSION ON FROZEN-PINNS

**Difference compared to training physics-informed neural networks:** We summarize the difference between training classical physics-informed neural networks and Frozen PINNs in Figure 8.

**Comparison between Frozen-PINN-swim and Frozen-PINN-elm** One of the main factors influencing the performance of Frozen-PINN-swim and Frozen-PINN-elm is the underlying solution of the PDE. We explain, with an example of the Burgers' equation, how the SWIM sampling can be leveraged when the solution has steep gradients, as one can sample localized basis functions in the part of the domain where the solution has steep gradients. For ELM, the probability of sampling steep basis functions with the vanilla ELM is lower, as illustrated in the Figure 2. Even if one uses a different distribution to sample the network parameters such that more basis functions with steep

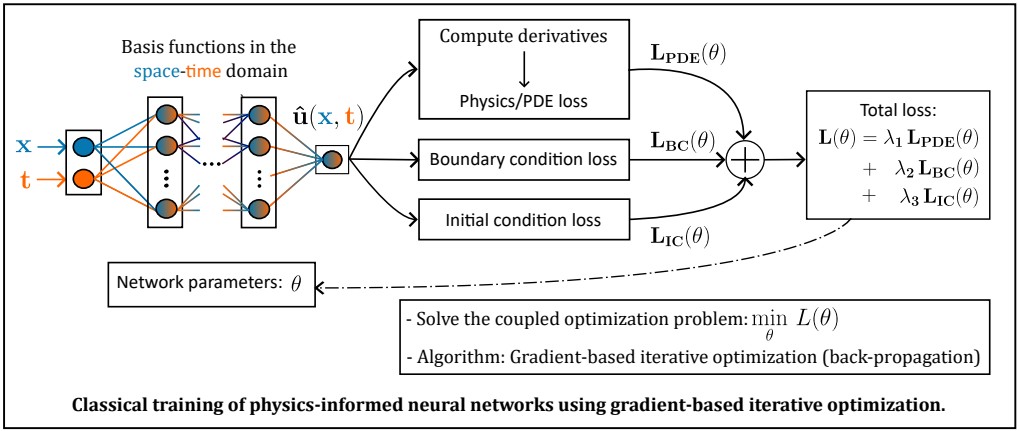

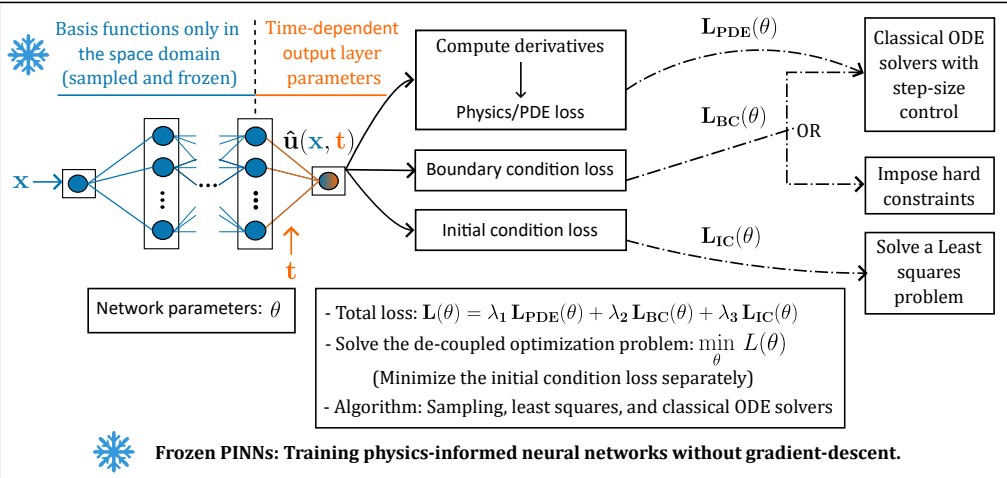

Figure 8: Comparison of Frozen-PINNs (bottom row) that leverage a gradient-descent-free training algorithm, with conventional PINNs (top row) that rely on gradient-based iterative optimization: conventional PINNs use basis functions in the entire spatio-temporal domain and solve a fully coupled optimization problem involving multiple loss terms via gradient-based iterative training. In contrast, Frozen-PINNs sample basis functions only in space, make time dependence explicit only in the output layer, decouple initial/boundary conditions, and leverage least squares and adaptive ODE solvers. Parameters dependent on space, time, and both are indicated by blue, orange, and blue-orange colors, respectively, offering a direct visual representation of the space–time separation in Frozen-PINNs. Notation: The network output $\hat{u}(x, t, \theta)$ approximates the solution to the PDE. The total loss term ($L(\theta)$) sums three loss terms - one for the initial condition ($L_{IC}(\theta)$), one for the boundary conditions ($L_{BC}(\theta)$), and one for the PDE residual ($L_{PDE}(\theta)$) that together impose physical constraints.

gradients are sampled, placing the basis functions at appropriate spatial locations is another challenge. With ELM, one cannot resample or choose basis functions using data as it is data-agnostic. Thus, especially if the solution has localized steep gradients, the performance of ELM is typically worse compared to SWIM. We additionally demonstrate with a snapshot of the Burgers' solution that SWIM basis functions exhibit a rapid exponential decay of error with increasing network width, where ELM, Fourier, and Chebyshev basis functions used in classical spectral methods suffer from the Gibbs phenomenon (see Appendix C.4.1) and lead to poor scaling and accuracy (see Figure 23, Figure 22).

If the underlying solution is sufficiently smooth and does not have steep gradients anywhere in the domain, ELM typically performs very well, as seen in the example with the Advection equation (see Section 3.1), Euler Bernoulli equation (see Section 3.2), and high-dimensional diffusion equation (Section 3.8), where Frozen-PINN-elm performs much better than Frozen-PINN-swim as shown in Table 31. While we just use the vanilla SWIM algorithm in the presented results, one can easily adapt

the algorithm and, after sampling the network parameters with SWIM, multiply the basis functions with a tunable scaling factor before applying the non-linearity to sample many more basis functions with shallow slopes.

Thus, the choice between the two strategies is particularly governed by the underlying solution of the PDE. Apart from the favorable cases for each method mentioned above, both methods have comparable performance and typically outperform PINNs by several orders of magnitude in speed and time. Thus, the rapid training of our approach could be leveraged to try out both approaches if one has no information about what the solution of the PDE could look like.

**Influence of random sampling on the method**    Similar to the question of how PINNs trained with Adam/SGD perform based on their random network initialization, understanding the influence of weights on the output is a challenge. There are two main differences between (stochastic) gradient-based optimization and our setting. First, after fixing the internal weights, we use regularized least-squares (not a stochastic method) to fit the initial condition. Second, we do not use a stochastic method to solve over time. Therefore, even though PINNs can adapt their random initialization over the gradient-based optimization, precisely that optimization also adds stochasticity. If the number of neurons for the model increases, the randomness in our case decreases because the regularized least-squares fit to the initial condition (which converges to a single solution in the limit of many neurons), while stochastic gradient descent will only converge to a distribution (because of mini-batch optimization). This has been observed for the supervised learning problems in Bolager et al. (2023), particularly in the transfer learning experiments. In Table 1, we observe that our model's performance is often orders of magnitude better, and the variance is on the same scale as the magnitude.

**"data-driven" and "data-agnostic" sampling**    In this work, we assume that we do not have access to the true solution of the PDE. The term "data-driven sampling" can be misleading for the problem setting of this paper, which concerns unsupervised learning tasks. Thus, here we clarify what we mean by data-driven sampling. Our data are random pairs of collocation points, but we do not have access to the true function values (because, at the initial time point $t = 0$, we have not solved the PDE yet). Thus, even though we do not have access to the true solution of the PDE, we call this "data-driven" sampling because we create the parameters of our basis functions (neurons) so that they are centered strictly within the domain. We achieve this by using data points sampled in the domain, thereby considering the geometry and bounds of the spatial domain. Note that with data-agnostic sampling in ELM, the neurons can easily be centered outside the spatial domain because weights and biases are chosen without considering any information about the geometry and bounds of the spatial domain. To summarize, though our algorithm proposes "data-driven" sampling, we do not start with time-series data and instead work in a self-supervised setting.

**Rationale for constructing outer basis functions**    One might reasonably ask that if one knows the outer basis functions analytically, why add another layer just to approximate them with `tanh` basis functions? When analytical basis functions are known, they should be used directly. However, in many cases, such expressions are not readily available. We argue that this idea of a boundary-compliant layer can be quite powerful for PDEs where the basis functions are not known analytically but only through boundary conditions, which we can then incorporate by constructing useful outer basis functions. For instance, to solve the diffusion equation on complex geometries, one can use the optimal bases consisting of the eigenfunctions of the Laplacian operator computed numerically at discrete points as the outer basis functions (Coifman & Lafon, 2006). Thus, representing them with `tanh` basis functions facilitates a straightforward computation of the derivatives needed for solving the PDE.

**Kolmogorov n-width barrier**    Without resampling the internal network parameters, our method faces the Kolmogorov n-width barrier Peherstorfer (2022); Du & Zaki (2021); Berman & Peherstorfer (2024); Kast & Hesthaven (2024) because our basis functions are not time-dependent. However, resampling basis functions at certain time points of the Frozen-PINN-swim (as done in the Burgers' equation in Section 3.4) results in a solution- and time-dependent basis approximation of the solution manifold and, thus, in theory, can break the barrier. PINNs can theoretically break the Kolmogorov n-width barrier as time is treated as an extra spatial dimension, and internal network parameters are time-dependent. However, for PINNs, the optimization issues pose much more severe challenges even on very simple PDEs and in low dimensions (Krishnapriyan et al., 2021; Wang et al., 2021; 2022).

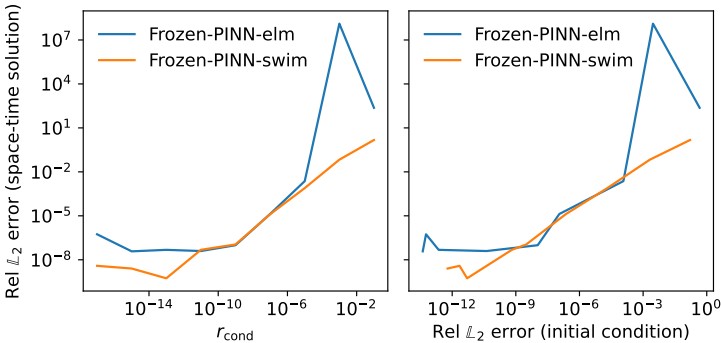

Figure 9: (Advection Equation): Empirical analysis of how initial condition loss affects the performance of Frozen-PINNs: (Left): Relative error of the full space-time solution Vs `rcond`, (Right): Relative error of the full space-time solution grows with the relative error of the initial condition (controlled via `rcond`).

So even though our vanilla Frozen-PINN-swim/Frozen-PINN-elm approach (without periodically resampling hidden layer weights) faces the Kolmogorov n-width barrier, we outperform PINNs, typically by several orders of accuracy and time in practice.

**Analysis of residual initial condition loss and its impact on model performance**  For all PDEs considered here, the initial condition is relatively easy to fit, and one can approximate it accurately by sampling enough collocation points at $t = 0$, using enough basis functions, and setting a relatively low regularization constant ($\approx 10^{-12}$). So, the initial condition loss is not the bottleneck in any of the PDEs we considered here. However, it is interesting to investigate the impact of initial condition loss on the model's performance.

Given $N_c$ collocation points $X \in \mathbb{R}^{N_c \times d}$, $M$ neurons, and hidden layer output $\Phi(X)$, the initial condition is computed via a least squares solution:

$$C(0) = u(X, 0)^\top [\Phi(X), \mathbb{1}]^+, \tag{12}$$

where $[\Phi(X), \mathbb{1}] \in \mathbb{R}^{(M+1) \times N_c}$ and the pseudo-inverse is denoted by $\cdot^+$. We compute this least squares solution $C(0)$ using the Python function `np.linalg.lstsq`, which takes as an argument `rcond` which is the cut-off ratio for small singular values of $[\Phi(X), \mathbb{1}]$. High cut-off ratios reduce the accuracy of the least-squares solution, while very low ratios lead to poorly conditioned systems that can introduce numerical errors.

We perform an experiment by progressively increasing the cut-off value to deliberately degrade the initial-condition fit and study its impact on the overall PDE residual. We solve the advection equation given in Equation (24), using the same hyperparameter settings as in Table 5, fix the advection coefficient to 40, and vary `rcond` from $10^{-1}$ to $10^{-17}$. Figure 9 shows that small `rcond` values yield highly accurate initial-condition fits and low relative error. As `rcond` increases, more dominant singular values are discarded in the least-squares solve, degrading the initial-condition representation and leading to larger errors in the full space–time solution. Thus, for Frozen-PINNs, maintaining a reasonably accurate initial condition fit is important, as inaccuracies can influence the ODE solve and increase the overall error.

**On the choice of the SVD truncation threshold:**  The SVD truncation threshold is a crucial hyperparameter for Frozen-PINNs, determining the dimensionality of the ODE solver and affecting the speed and accuracy of Frozen-PINNs. We conduct an ablation study on the SVD truncation threshold for Burgers' equation (Equation (28)), the nonlinear diffusion equation (Equation (29)), and the 10-dimensional diffusion equation (Equation (33)) with hyperparameters described in Table 14, Table 18, Table 29, respectively. We change the width to 200 for the non-linear diffusion equation, and for all PDEs, we only vary the SVD truncation thresholds. For detailed problem setups please refer to Appendix C.4, Appendix C.5, Appendix C.8.

Figure 10 highlights the well-known trade-off between accuracy and speed. Retaining fewer singular values reduces the dimensionality and stiffness of the last-layer ODE, yielding faster solutions with

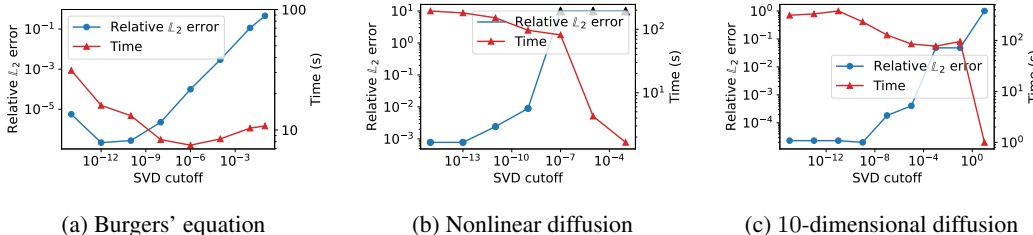

(a) Burgers' equation       (b) Nonlinear diffusion       (c) 10-dimensional diffusion

Figure 10: Impact of the SVD truncation threshold $\epsilon_{SVD}$ used for the SVD layer on the time-to-solution and accuracy of Frozen-PINN across three PDEs. Black triangles in the nonlinear diffusion plot indicate solution blow-up at large SVD cutoffs.

similar or slightly lower accuracy. Retaining more singular values increases dimensionality and stiffness, which slows the solver but improves accuracy. Importantly, the performance is robust for SVD truncation thresholds $\epsilon_{SVD} < 10^{-10}$.

The choice of the optimal SVD truncation threshold depends highly on the application constraints. Higher thresholds ($\epsilon_{SVD} \geq 10^{-8}$) are suitable for faster solutions with moderate accuracy, while very low thresholds ($\epsilon_{SVD} \leq 10^{-13}$) are preferable when high accuracy is the primary goal. The default value we choose is $10^{-12}$ as it represents a good trade-off between accuracy and speed. We always set the `rcond` (regularization constant for the initial least squares solve) to the same value as the SVD truncation threshold because: (a) it also represents the cut-off ratio for the SVD of the feature matrix for the initial least squares solve and it does not make sense to solve this with extremely high or low precision when the data has already been passed through the SVD layer, to maintain a similar level of truncation as the SVD layer. (b) Empirically, we observe very robust performance if we set the regularization constant to be equal to the SVD threshold (see Figure 9, where the relative error stays the same for the `rcond` values in the range $10^{-17} - 10^{-11}$, when the SVD truncation threshold is set at $10^{-14}$).

**On the choice between the two strategies for enforcing boundary conditions:** In practice, the choice between the boundary-compliant layer and the augmented ODE follows a simple cost-benefit tradeoff.

We recommend using a boundary-compliant layer when a problem-specific transformation $\phi_A(X)$ is easy to derive (e.g., zero Dirichlet, periodic boundary conditions on simple domains). It enforces boundary conditions (almost) exactly and does not enlarge the ODE system, so it is typically more efficient. The main limitation is that it requires deriving $\phi_A(X)$, which may be non-trivial for complex geometries or boundary conditions.

We recommend using the Augmented ODE strategy when boundary geometry or constraints make an analytic boundary-compliant mapping difficult. This is universally applicable and requires no problem-specific engineering since it soft-enforces boundary conditions by augmenting the state, at the cost of increasing system dimension and possibly worsening stiffness.

### B.2.2 COMPUTING SPATIAL AND TEMPORAL DIFFERENTIAL OPERATORS IN PDES

We use the notation described in Section 2 of the manuscript. We first discuss how to compute different spatial and temporal derivative terms appearing in the PDEs described in this manuscript using the neural network ansatz. We then use these expressions to reformulate the PDEs described in this manuscript as corresponding ODEs. We consider neural networks in the most general setting by considering the outer basis functions and the SVD layer (cf Algorithm 1).

**Computing spatial derivatives:** We list and describe how to compute the spatial derivatives of the approximate PDE solutions:

- *First-order spatial derivative* of the approximate PDE solution is computed as:

$$
\begin{aligned}
\hat{u}_x(x,t) &= C(t)[\Phi_{A_r}]_x(x) \\
&= C(t)[A_r W \odot \tilde{\sigma}_x(x), 0] \in \mathbb{R}^{1 \times d},
\end{aligned}
\tag{13}
$$

where $\odot$ is the Hadamard product, and

$$
\tilde{\sigma}_x(x) := [\sigma_z(z)|_{z=Wx^\top+b}, \sigma_z(z)|_{z=Wx^\top+b}, \ldots, \sigma_z(z)|_{z=Wx^\top+b}] \in \mathbb{R}^{M_s \times d}, \tag{14}
$$

with $\sigma_z(z) \in \mathbb{R}^{M_s}$ and $\sigma_z$ is the first derivative of the $tanh$ activation function.

- *Second-order spatial derivative* of the approximate PDE solution is computed as:

$$
\begin{aligned}
\hat{u}_{xx}(x,t) &= C(t)[\Phi_{A_r}]_{xx}(x) \\
&= C(t)[A_r W \odot W \odot \tilde{\sigma}_{xx}(x), 0] \in \mathbb{R}^{1 \times d},
\end{aligned}
\tag{15}
$$

where $\tilde{\sigma}_{xx}(x)$ is defined equivalently as $\tilde{\sigma}_x(x)$ but with $\sigma_{xx}$ being the second-order spatial derivative of the $tanh$ activation function.

- The *Laplacian* of the approximate PDE solution is computed as:

$$
\begin{aligned}
\Delta \hat{u}(x,t) &= C(t)[\Phi_{A_r}]_{xx}(x)\mathbb{1}, \quad \text{where, } \mathbb{1} \in \mathbb{R}^{d \times 1} \\
&= C(t)[A_r W \odot W \odot \tilde{\sigma}_{xx}(x), 0]\mathbb{1} \in \mathbb{R}^{1 \times 1}.
\end{aligned}
\tag{16}
$$

- *Fourth-order spatial derivative* of the approximate PDE solution is computed as:

$$
\begin{aligned}
\hat{u}_{xxxx}(x,t) &= C(t)[\Phi_{A_r}]_{xxxx}(x) \\
&= C(t)[A_r W \odot W \odot W \odot W \odot \tilde{\sigma}_{xxxx}(x), 0] \in \mathbb{R}^{1 \times d},
\end{aligned}
\tag{17}
$$

where $\sigma_{zzzz}$ is the fourth-order spatial derivative of the $tanh$ activation function.

**Computing time derivatives:** We now list and describe how to compute the time derivatives of the approximate PDE solutions:

- *First-order time derivative* of the approximate PDE solution is computed as:

$$
\hat{u}_t(x,t) = C_t(t)[\Phi_{A_r}](x). \tag{18}
$$

- *Second-order time derivative* of the approximate PDE solution is computed as:

$$
\hat{u}_{tt}(x,t) = C_{tt}(t)[\Phi_{A_r}](x). \tag{19}
$$

### B.2.3 REFORMULATING PDES AS ODES USING FROZEN-PINN ANSATZ

The partial differential equations considered in this work are recast as ordinary differential equations for evolving output layer coefficients, making use of the spatial and temporal derivatives derived in Appendix B.2.2. We denote the pseudo-inverse by $\cdot^+$.

**Advection equation:** The one-dimensional advection equation is

$$
u_t(x,t) + \beta u_x(x,t) = 0,
$$

where $\beta$ is a scalar. Approximating the solution with neural network ansatz (Equation (2)) and substituting Equation (18) and Equation (13) in the advection equation, we get,

$$
\begin{aligned}
C_t(t)[\Phi_{A_r}(X)] &= -\beta C(t)[\Phi_{A_r}(X)]_x, \\
C_t(t) &= -\beta C(t)[\Phi_{A_r}(X)]_x[\Phi_{A_r}(X)]^+.
\end{aligned}
$$

The initial condition is given by

$$
C(0) = u(X,0)^\top [\Phi_{A_r}(X)]^+.
$$

**Euler-Bernoulli equation:** The Euler-Bernoulli PDE considered in this manuscript is

$$u_{tt} + u_{xxxx} = f(x,t).$$

Approximating the solution with neural network ansatz (Equation (2)) and substituting Equation (17) and Equation (19) in the Euler-Bernoulli equation, we get,

$$C_{tt}(t)\Phi(X) = f(X,t)^\top - C(t)\Phi_{xxxx}(X)$$

We rewrite this second-order ODE as a combination of first-order ODEs given by

$$C_t(t) = D(t),$$
$$D_t(t)\Phi(X) = f(X,t)^\top - C(t)\Phi_{xxxx}(X).$$

We then reformulate the ODEs as

$$(C_t(t) \quad D_t(t)) = (C(t) \quad D(t)) \begin{pmatrix} 0 & -\Phi(X)_{xxxx}\Phi(X)^+ \\ \mathbb{1} & 0 \end{pmatrix} + (0 \quad \mathbb{1}) [f(X,t)]^\top \Phi(X)^+.$$

The initial condition is given by

$$C(0) = u(X,0)^\top \Phi(X)^+,$$
$$D(0) = u_t(X,0)^\top \Phi(X)^+.$$

The extension to the Euler-Bernoulli beam equation on a Winkler foundation is straightforward, where the reformulated ODE is written as:

$$(C_t(t) \quad D_t(t)) = (C(t) \quad D(t)) \begin{pmatrix} 0 & -(\Phi(X)_{xxxx} + \kappa\Phi(X))\Phi(X)^+ \\ \mathbb{1} & 0 \end{pmatrix} + (0 \quad \mathbb{1}) [f(X,t)]^\top \Phi(X)^+.$$

**Wave equation:** The wave equation considered in this manuscript is

$$u_{tt} - \kappa u_{xx} = f(X,t).$$

Approximating the solution with neural network ansatz (Equation (2)) and substituting Equation (15) and Equation (19) in the wave equation, we get,

$$C_{tt}(t)\Phi(X) = f(X,t)^\top - \kappa\, C(t)\Phi_{xx}(X)$$

We rewrite this second-order ODE as a combination of first-order ODEs given by

$$C_t(t) = D(t),$$
$$D_t(t)\Phi(X) = f(X,t)^\top - \kappa C(t)\Phi_{xx}(X).$$

We then reformulate the ODEs as

$$(C_t(t) \quad D_t(t)) = (C(t) \quad D(t)) \begin{pmatrix} 0 & -\kappa\Phi(X)_{xx}\Phi(X)^+ \\ \mathbb{1} & 0 \end{pmatrix} + (0 \quad \mathbb{1}) [f(X,t)]^\top \Phi(X)^+.$$

The initial condition is given by

$$C(0) = u(X,0)^\top \Phi(X)^+,$$
$$D(0) = u_t(X,0)^\top \Phi(X)^+.$$

**Burgers' equation:** The one-dimensional Burgers' PDE we consider is

$$u_t + uu_x - \alpha u_{xx} = 0,$$

where $\alpha$ is a scalar. Approximating the solution with neural network ansatz (Equation (2)) and substituting Equation (18), Equation (13) and Equation (15) in the Burgers equation, we get,

$$C_t(t)\Phi_{A_r}(X) = -(C(t)\Phi_{A_r}(X) \odot C(t)[\Phi_{A_r}]_x(X)) + \alpha\left(C(t)[\Phi_{A_r}]_{xx}(X)\right),$$
$$C_t(t) = -\left(C(t)\Phi_{A_r}(X) \odot C(t)[\Phi_{A_r}]_x(X) + \alpha\left(C(t)[\Phi_{A_r}]_{xx}(X)\right)\right)[\Phi_{A_r}(X)]^+$$

Note that the non-linearity is transferred to the right-hand side of the ODE. The initial condition is given by

$$C(0) = u(X,0)^\top \Phi(X)^+.$$

**Nonlinear diffusion equation:** The two-dimensional nonlinear diffusion equation we consider is

$$u_t - u\Delta u = f(x,t), \quad x \in \Omega \subset \mathbb{R}^2, \quad t \in [0,1]. \tag{20}$$

Approximating the solution with neural network ansatz (Equation (2)), substituting Equation (18), and Equation (15) in the nonlinear diffusion equation, we get,

$$C_t(t)\Phi(X) = (C(t)\Phi(X) \odot [C(t)\Phi_{xx}(X)]\mathbb{1}) + [f(X,t)]^\top,$$

$$C_t(t) = \left(C(t)\Phi(X) \odot [C(t)\Phi_{xx}(X)]\mathbb{1} + [f(X,t)]^\top\right)\Phi(X)^+.$$

Note that the non-linearity is transferred to the right-hand side of the ODE. The initial condition is given by

$$C(0) = u(X,0)^\top \Phi(X)^+.$$

**Nonlinear reaction-diffusion equation:** The two-dimensional nonlinear diffusion equation we consider is

$$u_t - \Delta u - u^2 = f(x,t), \quad x \in \Omega \subset \mathbb{R}^5, \quad t \in [0,1]. \tag{21}$$

Approximating the solution with neural network ansatz (Equation (2)), substituting Equation (18), and Equation (15) in the nonlinear diffusion equation, we get,

$$C_t(t)\Phi(X) = [C(t)\Phi_{xx}(X)]\mathbb{1} + (C(t)\Phi(X) \odot C(t)\Phi(X)) + [f(X,t)]^\top,$$

$$C_t(t) = \left([C(t)\Phi_{xx}(X)]\mathbb{1} + (C(t)\Phi(X) \odot C(t)\Phi(X)) + [f(X,t)]^\top\right)\Phi(X)^+.$$

The non-linearity $u^2$ is transferred to the right-hand side of the ODE. The initial condition is given by

$$C(0) = u(X,0)^\top \Phi(X)^+.$$

**High-dimensional diffusion equation:** The d-dimensional diffusion equation we consider is

$$u_t - \Delta u = f(x,t), \quad x \in \Omega \subset \mathbb{R}^d, \quad t \in [0,1]. \tag{22}$$

Approximating the solution with neural network ansatz (Equation (2)), substituting Equation (18), and Equation (15) in the diffusion equation, we get,

$$C_t(t)\Phi(X) = [C(t)\Phi_{xx}(X)]\mathbb{1} + [f(X,t)]^\top,$$

$$C_t(t) = \left([C(t)\Phi_{xx}(X)]\mathbb{1} + [f(X,t)]^\top\right)\Phi(X)^+.$$

The initial condition is given by

$$C(0) = u(X,0)^\top \Phi(X)^+.$$

**Kuramoto-Sivashinsky equation:** The governing PDE considered in this manuscript is

$$u_t + \alpha u u_x + \beta u_{xx} + \gamma u_{xxxx} = f(X,t).$$

Approximating the solution with neural network ansatz (Equation (2)) and substituting Equation (13), Equation (15), Equation (17) and Equation (18) in the Kuramoto-Sivashinsky equation, we get,

$$C_t(t)\Phi(X) = f(X,t)^\top - \alpha C(t)\Phi(X) \odot C(t)\Phi_x(X) - \beta C(t)\Phi_{xx}(X) - \gamma C(t)\Phi_{xxxx}(X).$$

We can reformulate it as:

$$C_t(t) = \left(f(X,t)^\top - \alpha C(t)\Phi(X) \odot C(t)\Phi_x(X) - \beta C(t)\Phi_{xx}(X) - \gamma C(t)\Phi_{xxxx}(X)\right)\Phi(X)^+.$$

The initial condition is given by

$$C(0) = u(X,0)^\top \Phi(X)^+.$$

**Note on ODE solvers and interpolation in time:** We use the `solve_ivp` routine of the SciPy package Virtanen et al. (2020). One can pass test points in time as an argument to the method `solve_ivp`. One can optionally set the parameter `dense_output` to true, which means that the output of the ODE is a function handle that can be evaluated by interpolation at any time point $t \in \Omega$. The method specified dictates the interpolation order. RK23 uses a cubic Hermite polynomial, while DOPRI85 uses a seventh-order polynomial.

### B.2.4 HANDLING BOUNDARY CONDITIONS VIA BOUNDARY-COMPLIANT LAYER

To enforce *periodic boundary conditions*, it is sufficient for each basis function to satisfy the periodic condition individually, as the Frozen-PINN ansatz, which is a linear combination of these functions, will inherently satisfy it as well. For instance, for a one-dimensional spatial domain, we find $A$ so that $A\Phi(x_l) = A\Phi(x_r)$, where $x_l, x_r$ are the left and right boundary points of the domain. In this paper, for certain PDEs (see Appendix C), for $x \in \Omega$ and $k = 1, 2, \ldots, M_s$, we approximate $[A\Phi]_k(x) = \sin(kx)$ (for $k$ even) and $[A\Phi]_k(x) = \cos(kx)$ (for $k$ odd) and set $c_0(t) = 1$ for all $t$. For *zero Dirichlet boundary condition* given by $u(x) = 0$, we can use the technique described above by choosing basis functions so that $A\phi(x) = 0$ for $x \in \partial\Omega$. For other boundary conditions, we propose using the augmented ODE trick to satisfy the boundary conditions.

### B.2.5 HANDLING BOUNDARY CONDITIONS VIA AUGMENTED ODE

Our approaches to satisfying the Dirichlet and periodic boundary conditions are already explained in the main text. Here, we explain how we handle time-dependent Dirichlet boundary conditions and Neumann boundary conditions.

**Time-dependent Dirichlet boundary conditions:** For handling time-dependent Dirichlet boundary conditions ($u(x, t) = g(x, t)$ for $x \in \partial\Omega$), we set $A$ to the identity map and augment the ODE (Equation (3)) with an additional equation given by

$$\hat{u}_t(x, t) = g_t(x, t) \text{ for } x \in \partial\Omega \implies C_t(t) = \underbrace{[R(X, C(t)), g_t(X_b, t)]}_{\in \mathbb{R}^{1 \times (N_c + N_b)}} \underbrace{\Phi_A([X, X_b])^+}_{\in \mathbb{R}^{(N_c + N_b) \times (M_b + 1)}}.$$

In the example in Section 3.5, we know the solution on the boundary at all time points, which is continuously differentiable. If the solution on the boundary points is not available at all time points, one can interpolate and approximate the derivative of the solution on the boundary.

**Neumann boundary conditions:** For simple spatial domains, one can choose appropriate outer basis functions as described in Section 2.4 that inherently satisfy the Neumann boundary conditions. For instance, for zero Neumann boundary conditions on a one-dimensional domain, one can choose outer basis functions consisting of cosines of different frequencies scaled to the domain (function value is 1 at the boundaries) so that their spatial derivatives, which are the sine functions, are zero on the boundary points.

On complicated domain geometries, to satisfy Neumann boundary conditions ($\nabla u(x, t) \cdot \hat{n}(x) = 0$ for $x \in \partial\Omega$), we set $A$ to the identity map and augment the ODE (Equation (3)) with an additional equation for the boundary points and solve

$$C_t(t) = \underbrace{[R(X, C(t)), 0]}_{\in \mathbb{R}^{1 \times (N_c + N_b)}} \underbrace{[\Phi_A(X), \nabla\Phi_A(X_b)[\hat{n}(X_b)]^\top]^+}_{\in \mathbb{R}^{(N_c + N_b) \times (M_b + 1)}}.$$

### B.3 IGA-FEM

First introduced in Hughes et al. (2005), Isogeometric analysis (IGA) is a numerical method developed to unify the fields of computer-aided design (CAD) and finite element analysis (FEA). The key idea is to represent the solution space for the numerical analysis using the same functions that define the geometry in CAD (Cottrell et al., 2009), which include the B-Splines and Non-Uniform Rational B-Splines (NURBS) (Piegl & Tiller, 1997).

In this paper, we use B-Splines as the basis functions. The B-Splines are defined using the Cox-de Boor recursion formula (COX, 1972; de Boor, 1972), i.e.,

$$N_{i,0}(\xi) = \begin{cases} 1 & \xi_i \leq \xi < \xi_{i+1} \\ 0 & \text{otherwise,} \end{cases}$$

$$N_{i,p}(\xi) = \frac{\xi - \xi_i}{\xi_{i+p} - \xi_i} N_{i,p-1}(\xi) + \frac{\xi_{i+p+1} - \xi}{\xi_{i+p+1} - \xi_{i+1}} N_{i+1,p-1}(\xi),$$

where $\xi_i$ is the $i$th knot, and $p$ is the polynomial degree. The vector $\Xi = [\xi_1, \xi_2, \ldots, \xi_{n+p+1}]$ is the knot vector, where $n$ is the number of B-Splines. By specifying the knot vector, we define the basis functions we use to solve the PDEs. We use a uniform open knot vector, where the first and last knots have multiplicity $p+1$, the inner knots have no multiplicity, and all knots that have different values are uniformly distributed. We refer to the knots with different values as "nodes". The intervals between two successive nodes are knot spans, which can be viewed as "elements". The elements form a "patch". A domain can be partitioned into subdomains, and each is represented by a patch. In our work, we use a single patch to represent the entire 1D domain. Figure 11a shows an example of such a patch, where the B-Splines are $C^p$-continuous within the knot spans and $C^{p-1}$ continuous at the inner knots. In order to address the boundary conditions, we adapt the B-Splines as shown in Figure 11b Figure 11c, so that the boundary conditions are directly built into the solution space.

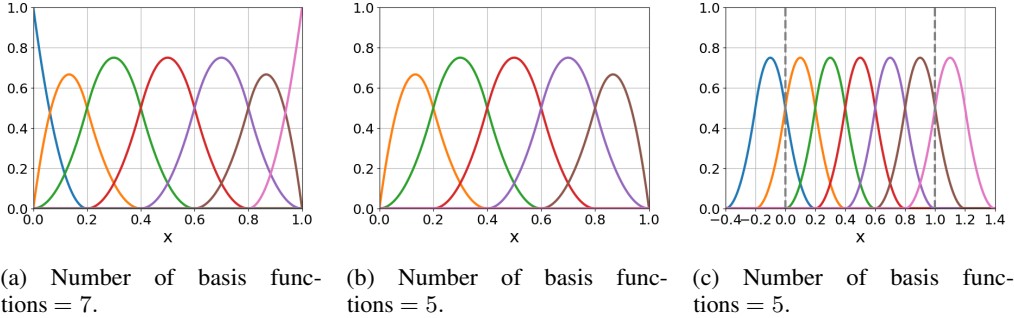

(a) Number of basis functions = 7.

(b) Number of basis functions = 5.

(c) Number of basis functions = 5.

Figure 11: Examples of B-Splines representing the 1D domain $[0,1]$. Number of nodes = 6 and degree of polynomials = 2. (Left): The original B-Splines. (Middle): Adapted B-Splines to satisfy the Dirichlet boundary condition. (Right): Adapted B-Splines to satisfy the periodic boundary condition. Note that the first (blue) spline is identical to the second last (brown) one, and the second (orange) spline is identical to the last (pink) one, as they share the same coefficient. The gray dashed lines indicate where the domain starts and ends.

In the following, we refer to the adapted B-Splines as basis functions $\phi_k(x)$. Thus, the solutions of PDEs are approximated by

$$u(x,t) = \sum_{k=1}^{K} c_k(t)\phi_k(x).$$

We solve the PDEs in the weak formulation. For the linear advection equation (see Equation (24)), the weak form of the equation is

$$\sum_{k=1}^{K} c_k'(t) \int_X \phi_k(x)v(x)dx + \beta \sum_{k=1}^{K} c_k(t) \int_X \phi_k'(x)v(x)dx = 0, \tag{23}$$

where $v(x)$ are the test functions. The test functions are chosen to be the same as the basis functions. The integral of the functions is computed using Gaussian quadrature. Then we solve the linear Ordinary differential equation (ODE)

$$\mathbf{M}\dot{\mathbf{c}} + \mathbf{K}\mathbf{c} = \mathbf{0},$$

where matrix $\mathbf{M}$ and matrix $\mathbf{K}$ contain the integral of the B-Splines and their derivatives, and the coefficient $\beta$, which are given. We solve the Euler-Bernoulli equation equation 25 and the Burgers' equation equation 28 in a similar way. The boundary condition for the Euler-Bernoulli equation is, in addition, weakly imposed, as is done in Prudhomme et al. (2001).

## C    Supplementary details on numerical experiments

Here, we discuss additional experimental details for the PDEs considered in this work. We start by listing the details on the code repository, FEM software, hardware, error metrics, and ablation studies:

- **Code repository:** The source code, along with the instructions on reproducing the results, is provided in the supplemental material (zipped file) and is made publicly available (see Section 3). The code repository provides Python scripts and notebooks that can be executed and tested readily. Moreover, we make the refactored repository publicly available and intend to actively maintain it.

- **FEM code:** In this paper, we use `DOLFINx` 0.8.0 to solve the nonlinear diffusion equation (see equation 29). DOLFINx (Baratta et al., 2023), which is part of the FEniCS project, is a C++ and Python library used for solving PDEs with the finite element method (FEM). It provides tools for defining complex geometries, formulating variational problems, and solving them efficiently on distributed architectures. We used the software Gmsh (Geuzaine & Remacle, 2009) to generate a mesh for this experiment with complicated geometry, as shown in Figure 24a.

- **Hardware details:** The computational experiments for Frozen PINNs, FEM, and IGA-FEM were performed with: `Ubuntu` 20.04.6 LTS, `NVIDIA` driver 515.105.01 and i7 CPU.

- **Metrics for computing errors:** We use the Root Mean Squared Error (RMSE) and the relative $L^2$ error to quantify errors in all experiments (see Appendix C for the definitions). We compute the test error on a uniform grid for all PDEs with 256 points in space and 100 points in time, unless otherwise specified. We use `float64` numerical precision in all the experiments.

  Let $d$ be the dimension of space and $\Omega \times [0, T] \subset \mathbb{R}^d \times \mathbb{R}$ be the spatio-temporal domain. Given $N$ points in a test set $X$, the error metrics we use to compare numerical results are Root Mean Squared Error (RMSE) and relative $L^2$ error given by

$$\text{RMSE} := \sqrt{\frac{\sum_{x \in X}(u_{true}(x) - u_{pred}(x))^2}{N}},$$

  and

$$\text{Relative } L^2 \text{ error} := \frac{\sqrt{\sum_{x \in X}(u_{true}(x) - u_{pred}(x))^2}}{\sqrt{\sum_{x \in X}(u_{true}(x))^2}}.$$

  For each experiment, the mean and standard deviation of the RMSE and the relative $L^2$ error are computed with three seeds.

- **Ablation studies for neural architecture and SVD layer:** We perform ablation studies whenever necessary for the neural architectures we considered in this work. Importantly, we also perform an ablation study on the SVD layer. To quantify the compression in width after the SVD layer, we define a compression ratio as $C_r = \frac{M_s}{r}$, where $M_s$ is the width of the (sampled) hidden layer before the SVD layer (assuming no-boundary-compliant layer), and $r$ is the width of the SVD layer (see Figure 3). We define a speed-up in computation time as $s = \frac{T_{\text{no-svd}}}{T_{\text{svd}}}$ as the ratio of computational time without the SVD layer to the time required with the SVD layer.

We now describe the detailed problem setups, ablation studies, and plots comparing the results of Frozen-PINNs with those of other approaches for all PDEs considered here (see Figure 12).

## C.1 LINEAR ADVECTION EQUATION

**Problem setup:** The linear advection equation describes the transport of a quantity and is used to model many real-life applications, such as simplified traffic flow models, transport of pollutants in rivers or the atmosphere (Rood, 1987; McGraw et al., 2024). Here, we consider the linear advection equation with periodic boundary conditions given by

$$u_t(x,t) + \beta u_x(x,t) = 0, \quad \text{for } x \in [0, 2\pi], \ t \in [0, 1], \tag{24a}$$

$$u(x, 0) = \sin(x), \quad \text{for } x \in [0, 2\pi], \tag{24b}$$

$$u(0, t) = u(2\pi, t), \quad \text{for } t \in [0, 1]. \tag{24c}$$

The analytical solution of Equation (24) is given by $u(x,t) = \sin(x - \beta t)$. We describe detailed hyperparameter settings used for the experiments on: (a) high-advection speeds (how the error grows

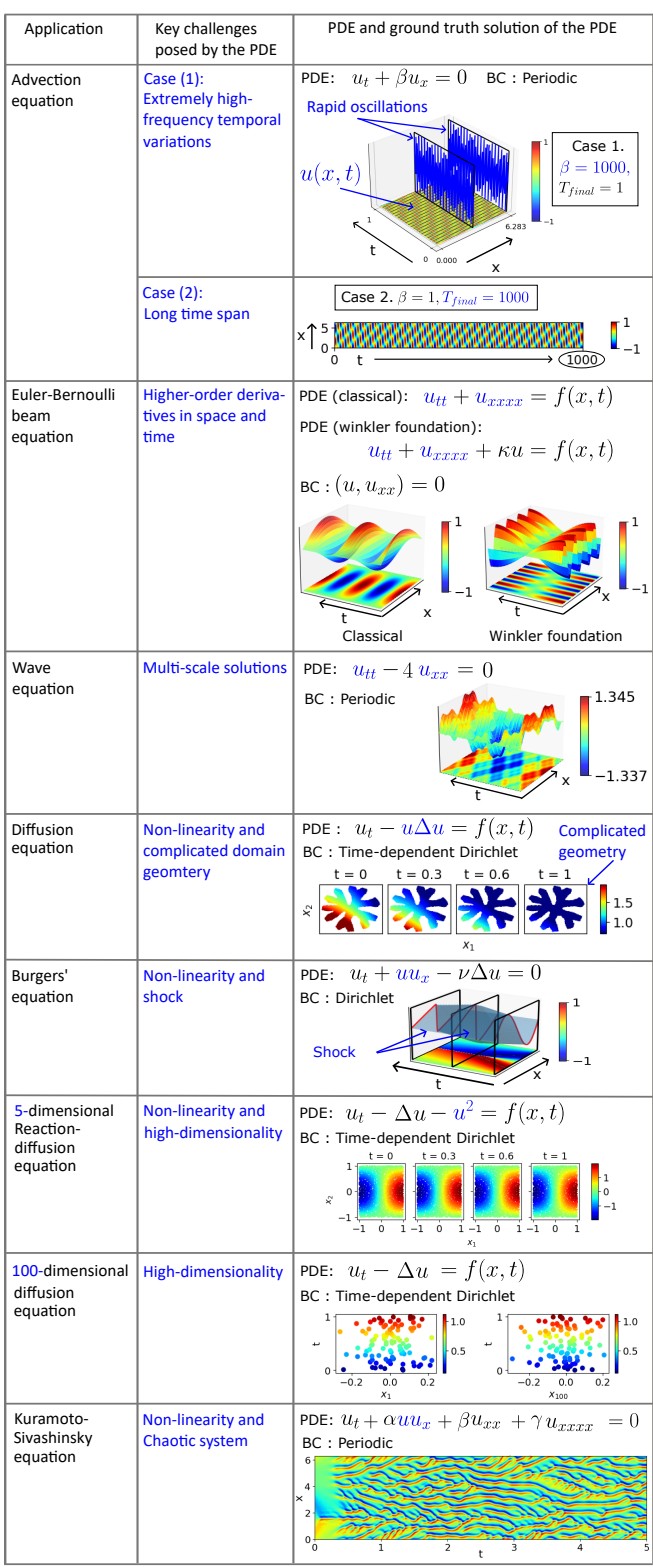

Figure 12: Overview of the PDE benchmarks considered in this study, highlighting the core challenges associated with each problem and their corresponding ground truth solutions. Boundary conditions are abbreviated as BC.

with the advection coefficient $\beta$), (b) convergence (how the error decays with the number of basis functions for a fixed advection coefficient $\beta = 10$), (c) error for advection coefficient $\beta = 40$ (for a comparison with other PINN-based variants), and (d) long-time simulation for $T = 1000$ seconds for a fixed advection coefficient $\beta = 1$ in Table 3, Table 4, Table 5, and Table 8, respectively. The hidden layer weights for ELM and Frozen-PINN-elm are sampled from the Gaussian distribution and biases from a uniform distribution in $[-4, 4]$. For SWIM and ELM, we use 1000 interior points for $\beta \in \{10^{-2}, 10^{-1}, 1, 10\}$, and we use 8000 interior points for $\beta \in \{40, 100\}$. The code repository contains all the necessary Python notebooks to reproduce results for Frozen-PINNs for all the different cases of the advection equation considered here, including the three key experiments concerning high advection speeds, convergence, and long-time simulation (see Section 3.1).

**Deeper networks and further optimization experiments:** Additional experiments are carried out for baseline PINN on deeper networks with 10 and 20 hidden layers, where each hidden layer has 30 neurons. The experiments are run for 20000 epochs using Adam and L-BFGS optimizers under multiple learning rates for the advection equation with $\beta = 10$ and $\beta = 40$. Tables 6 and 7 summarize the RMSE, relative $L^2$ errors, and training times for $\beta = 10$ and $\beta = 40$ cases, respectively. The results show consistency with the known literature of PINNs. First, deeper PINNs do not directly lead to better performance. Increasing depth from 10 to 20 layers often degrades accuracy for both optimizers, reflecting the optimization difficulty of fully-connected PINNs as the models become deeper. This behavior is also discussed previously in the literature, for instance by Wang et al. (2024b), where the authors show that PINN performance is known to degrade when larger and deeper neural network architectures are employed. Second, the $\beta = 40$ case is known to be challenging for PINNs (Krishnapriyan et al., 2021) due to high frequency features, and the results presented in Table 7 show failures across depths, learning rates, and optimizers. The results show that even with larger networks, longer training, and different optimizers, standard PINNs face challenges in achieving high accuracy, especially for $\beta = 40$, and require more computational time. Thus, matching the accuracy of standard PINNs with the proposed method is inherently challenging and computationally expensive.

**Ablation studies:** For the advection coefficient $\beta = 10$, the ablation study for Frozen-PINN-swim, Frozen-PINN-elm, and vanilla PINNs is already presented in Figure 4(Middle) for varying the number of neurons and interior points. The ablation studies for PINNs for the network width and number of interior points are presented in Table 9, and Table 10, respectively. Since the network width is already quite low for optimal parameters, the SVD layer does not further reduce the dimension of the ODE system. Hence, we do not perform ablation studies for the SVD layer in Frozen-PINNs, as it is not used in this case.

**Comparison of results:** Figure 13 shows the absolute errors obtained with the Frozen-PINN-swim, Frozen-PINN-elm, PINN, Causal PINN, and IGA methods along with the ground truth for $\beta = 40$. One can observe that all approaches considered here, besides Frozen-PINNs and IGA-FEM, fail to capture the high-frequency temporal dynamics. Figure 14 shows the true solution at $\beta = 1$ for the example with long time simulation.

## C.2 EULER-BERNOULLI EQUATION

**Problem Setup:** The time-dependent Euler–Bernoulli beam equation models the dynamic behavior of beams, including vibrations and transient loads. It is used to model loads on rail tracks, bridges, and aircraft wings, among many other applications (Beskos, 1987). We consider two different types of the Euler-Bernoulli beam equations in this work. The first is the classical Euler-Bernoulli beam equation that models a simply supported beam with varying transverse force and is described as

$$u_{\text{tt}} + u_{\text{xxxx}} = f(x,t) \quad x \in [0,\pi], t \in [0,1], \tag{25a}$$

where $f(x,t) = (1 - 16\pi^2)\sin(x)\cos(4\pi t)$, with initial and boundary conditions

$$u(x,0) = \sin(x), \quad u_t(x,0) = 0, \tag{25b}$$

$$u(0,t) = u(\pi,t) = u_{\text{xx}}(0,t) = u_{\text{xx}}(\pi,t) = 0. \tag{25c}$$

The forcing function and the analytical solution are taken from Kapoor et al. (2023).

We consider another variant of the Euler-Bernoulli beam equation, with a Winkler foundation (an elastic, deformable foundation) given by:

$$u_{\text{tt}} + u_{\text{xxxx}} + p(x,t) = f(x,t) \quad x \in [0,8\pi], t \in [0,1]. \tag{26a}$$

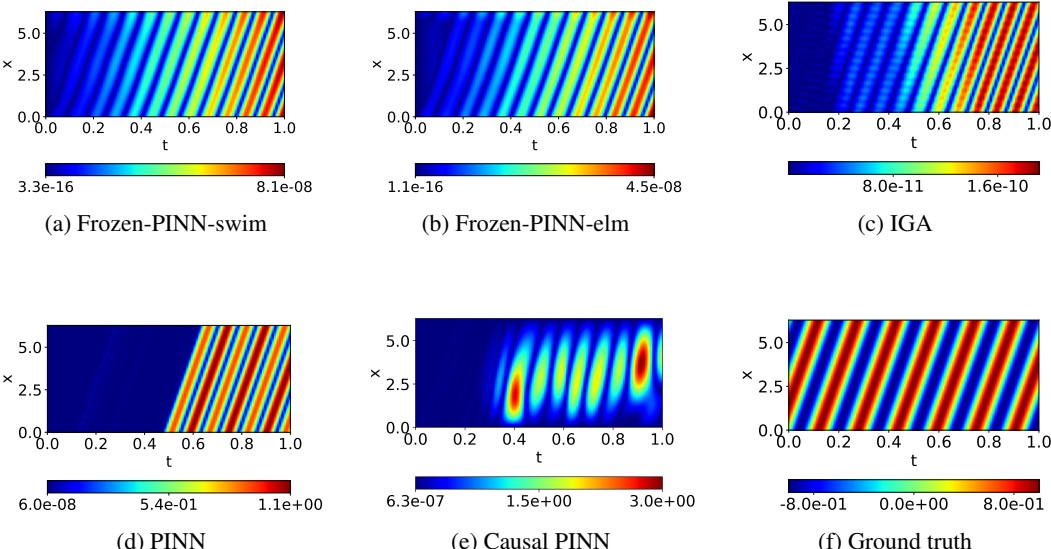

Figure 13: Advection equation ($\beta = 40$): absolute error plots and ground truth.

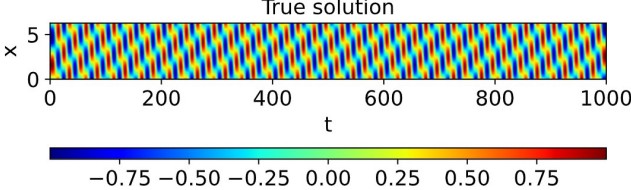

Figure 14: Advection equation $\beta = 1$, (long time simulation): Analytical solution $u(x,t) = \sin(x - \beta t)$.

The forcing term is $f(x,t) = (2 - \pi^2)\sin(x)\cos(\pi t)$, with the initial and boundary conditions

$$u(x,0) = \sin(x), \quad u_t(x,0) = 0, \tag{26b}$$

$$u(0,t) = u(8\pi,t) = u_{xx}(0,t) = u_{xx}(8\pi,t) = 0. \tag{26c}$$

The foundation reaction force $p(x,t)$ is assumed to be proportional to the displacement of the beam and modeled as $p(x,t) = \kappa u(x,t)$, where $\kappa$ is the spring constant and is set to 1 in this case. The forcing function and the analytical solution for the Euler-Bernoulli beam equation with a Winkler foundation are taken from Kapoor et al. (2024b).

**Ablation studies:** The ablation studies for the PINN-based variants for classical Euler Bernoulli and the one with the Winkler foundation could be found in Kapoor et al. (2023; 2024b). The hyperparameters for various neural PDE solvers used for solving the classical Euler-Bernoulli PDE and the one with the Winkler foundation are described in Table 11, and Table 12, respectively. The hidden layer weights for Frozen-PINN-elm are sampled from the Gaussian distribution and biases from a uniform distribution in $[-2, 2]$.

For this example, we have employed the augmented ODE strategy to satisfy boundary conditions and obtain the results presented in Table 1. We empirically investigate the effect of the penalty term in the augmented ODE on the performance of Frozen-PINNs, considering both accuracy and computation time. For this experiment, we use the hyperparameters described in Table 11 for Frozen-PINN-elm (high-precision regime) and vary the value of $\kappa$. Figure 15 shows that: (a) for $\kappa > 10^5$, the boundary loss is negligible (RMSE $< 10^{-10}$) and the total loss is very low (RMSE $\sim 10^{-5}$–$10^{-9}$), and (b) for extremely large $\kappa \geq 10^6$, the augmented ODE becomes slightly stiffer, resulting in increased solution time.

Table 3: Advection equation (high-advection speeds): Network hyper-parameters used for $\beta \in \{10^{-2}, 10^{-1}, 1, 10, 40, 100, 1000, 10000\}$ to study the influence of the advection coefficient on the errors (optimal hyper-parameters in bold) (see Figure 4(Left).

| | Parameter | Value |
|---|---|---|
| Frozen-PINN-swim, | Number of hidden layers | 2 |
| Frozen-PINN-elm | Hidden layer width | $[140, \mathbf{380}, 560]$ |
| | Outer basis functions | $[10, \mathbf{14}, 20, 40]$ |
| | Activation | `tanh` |
| | $L^2$-regularization | $[10^{-8}, \mathbf{10^{-10}}, 10^{-12}, 10^{-14}]$ |
| | Loss | mean-squared error |
| | boundary condition strategy | boundary-compliant layer |
| SWIM, ELM | Number of hidden layers | 2 |
| | SVD cutoff | $10^{-12}$ |
| | Hidden layer width | $[140, \mathbf{380}, 560]$ |
| | Activation | `tanh` |
| | $L^2$-regularization | $[10^{-8}, \mathbf{10^{-10}}, 10^{-12}]$ |
| | Loss | mean-squared error |
| | # Initial and boundary points | 400 |
| IGA | Number of nodes | 16 |
| | Degree of polynomials | 8 |
| | Number of basis functions | 15 |
| PINN | Number of hidden layers | 4 |
| | Layer width | $[10, 20, \mathbf{30}, 40]$ |
| | Activation | `tanh` |
| | Optimizer | LBFGS |
| | Epochs | 5000 |
| | Loss | mean-squared error |
| | Learning rate | 0.1 |
| | Batch size | 200 |
| | Parameter initialization | Xavier (Glorot & Bengio, 2010) |
| | Loss weights, $\lambda_1, \lambda_2$ | 1, 1 |
| | # Interior points | $[500, 1000, 1500, \mathbf{2000}]$ |
| | # Initial and boundary points | 600 |
| Causal PINN | Number of hidden layers | 4 |
| | Layer width | 30 |
| | Activation | `tanh` |
| | Optimizer | ADAM followed by LBFGS |
| | ADAM Epochs | 2000 |
| | LBFGS Epochs | 5000 |
| | Loss | mean-squared error |
| | Learning rate | 0.1 |
| | Batch size | 2000 |
| | Parameter initialization | Xavier (Glorot & Bengio, 2010) |
| | Loss weights, $\lambda_1, \lambda_2$ | 1, 1 |
| | # Interior points | 40000 |
| | # Initial and boundary points | 6000 |
| | Causality parameter, $\epsilon$ | 10 |

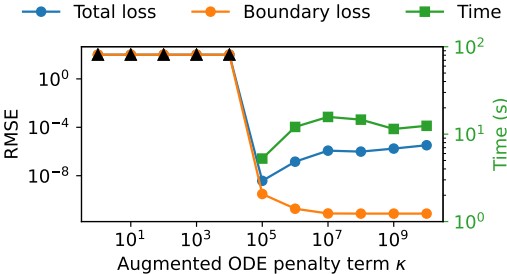

Figure 15: The effect of penalty term $\kappa$ in the augmented ODE (see Equation (5)) on the losses and time to solution.

**Comparison of results:** Figure 16 and Figure 17 present the absolute errors for the classical Euler-Bernoulli PDE and its variant with a Winkler foundation, respectively, using Frozen-PINN-swim, Frozen-PINN-elm, PINN, and IGA methods, along with the true solution. The error plots for the Euler-Bernoulli beam equation with a Winkler foundation for other variants of PINNs, such as

Table 4: Advection equation (convergence for $\beta = 10$): Optimal hyper-parameters in the experiment designed to study how the error decays with the number of basis functions in the neural network (see Figure 4(Middle).

| | Parameter | Value |
|---|---|---|
| Frozen-PINNs (both variants) | Number of hidden layers | 2 |
| | Hidden layer width | $[2, ..., 30]$ |
| | Activation | `tanh` |
| | $L^2$-regularization | $[10^{-7}, 10^{-8}, 10^{-9}, \mathbf{10^{-10}}, 10^{-11}, 10^{-12}]$ |
| | Loss | mean-squared error |
| | boundary condition strategy | boundary-compliant layer |
| PINN | Number of hidden layers | 4 |
| | Layer width | $[10, 20, \mathbf{30}, 40]$ |
| | Activation | `tanh` |
| | Optimizer | LBFGS |
| | Epochs | 5000 |
| | Loss | mean-squared error |
| | Learning rate | 0.1 |
| | Batch size | 200 |
| | Parameter initialization | Xavier (Glorot & Bengio, 2010) |
| | Loss weights, $\lambda_1, \lambda_2$ | 1, 1 |

Table 5: Advection equation (for $\beta = 40$): Hyper-parameters for the results in Table 1.

| | Parameter | Value |
|---|---|---|
| Frozen-PINN-elm (low-precision) | Number of hidden layers | 2 |
| | Hidden layer width | 50 |
| | Outer basis functions | $[14]$ |
| | svd cutoff | $[10^{-12}]$ |
| | Activation | `tanh` |
| | $L^2$-regularization | $10^{-10}$ |
| | ODE solver tolerance | $10^{-4}$ |
| | Loss | mean-squared error |
| | boundary condition strategy | boundary-compliant layer |
| Frozen-PINN-swim (high-precision) | Number of hidden layers | 2 |
| | Hidden layer width | 380 |
| | Outer basis functions | $[14]$ |
| | svd cutoff | $[10^{-12}]$ |
| | Activation | `tanh` |
| | $L^2$-regularization | $10^{-14}$ |
| | ODE solver tolerance | $10^{-8}$ |
| | Loss | mean-squared error |
| | boundary condition strategy | boundary-compliant layer |
| PINN | Number of hidden layers | 4 |
| | Layer width | $[10, 20, \mathbf{30}, 40]$ |
| | Activation | `tanh` |
| | Optimizer | LBFGS |
| | Epochs | 5000 |
| | Loss | mean-squared error |
| | Learning rate | 0.1 |
| | Batch size | 200 |
| | Parameter initialization | Xavier (Glorot & Bengio, 2010) |
| | Loss weights, $\lambda_1, \lambda_2$ | 1, 1 |

Wavelet PINN, causal PINN, adaptive PINN, and self-adaptive PINNs, can be found in Kapoor et al. (2024b). Table 1 shows the summary of results for the classical Euler-Bernoulli beam equation and the variant considering the Winkler foundation for different methods.

## C.3 Wave equation

**Problem Setup:** The acoustic wave equation models the propagation of sound waves through a medium. It describes how pressure or velocity evolve over time. We consider the wave equation on $\Omega = [0, 1]$ for time $t \in [0, 1]$ from Hao et al. (2024), given by:

$$u_{tt} - 4u_{xx} = 0, \quad x \in \Omega, \quad t \in [0, 1], \tag{27a}$$

Table 6: Results of PINNs for advection equation ($\beta = 10$) for dense networks (number of hidden layers = 10 and 20, with each hidden layer having 30 neurons). The experiment studies the performance of PINNs with L-BFGS and ADAM optimizers under different learning rates. Each case is run for 20000 epochs.

| Optimizer | Learning rate | RMSE | | Relative $L^2$ error | | Training time (s) | |
| | | Hidden layers | | Hidden layers | | Hidden layers | |
| | | 10 | 20 | 10 | 20 | 10 | 20 |
|---|---|---|---|---|---|---|---|
| L-BFGS | 0.1 | 6.02e-4 | 1.23e-2 | 8.51e-4 | 1.74e-2 | 204.25 | 336.32 |
| | 0.01 | 1.57e-3 | 1.30e-2 | 2.22e-3 | 1.84e-2 | 213.26 | 327.41 |
| Adam | 0.001 | 2.26e-2 | 1.64e-2 | 3.19e-2 | 2.32e-2 | 174.84 | 301.12 |
| | 0.0001 | 6.72e-3 | 2.07e-2 | 9.5e-3 | 2.92e-2 | 190.24 | 317.97 |

Table 7: Results of PINNs for advection equation ($\beta = 40$) for dense networks (number of hidden layers = 10 and 20, with each hidden layer having 30 neurons). The experiment studies the performance of PINNs with ADAM and L-BFGS optimizers under different learning rates. Each case is run for 20000 epochs.

| Optimizer | Learning rate | RMSE | | Relative $L^2$ error | | Training time (s) | |
| | | Hidden layers | | Hidden layers | | Hidden layers | |
| | | 10 | 20 | 10 | 20 | 10 | 20 |
|---|---|---|---|---|---|---|---|
| L-BFGS | 0.1 | 3.09e-3 | 1.53e+1 | 4.37e-3 | 2.17e+1 | 207.25 | 320.43 |
| | 0.01 | 1.46e-2 | 7.07e-1 | 2.07e-2 | 1.00e+0 | 205.08 | 323.43 |
| Adam | 0.001 | 3.66e-1 | 7.07e-1 | 5.18e-1 | 1.00e+0 | 168.14 | 331.03 |
| | 0.0001 | 6.67e-1 | 6.82e-1 | 9.44e-1 | 9.65e-1 | 181.89 | 316.90 |

Table 8: Advection equation (long-time simulation for $\beta = 1$): Optimal hyper-parameters for Frozen-PINNs in the experiment used to demonstrate that the errors with Frozen-PINNs stay low for simulations up to 1000 seconds (see Figure 4(Right)).

| | Parameter | Value |
|---|---|---|
| Frozen-PINN | Number of hidden layers | 2 |
| (both variants) | Hidden layer width | 250 |
| | Outer basis functions | 25 |
| | Activation | tanh |
| | $L^2$-regularization | $[\mathbf{10^{-10}}]$ |
| | Loss | mean-squared error |
| | boundary condition strategy | boundary-compliant layer |

Table 9: Advection equation ($\beta = 10$): Ablation study for PINN (LBFGS) with respect to the network width. The mean is computed over 3 seeds.

| Layer width | Training time (s) | RMSE | Relative $L^2$ error |
|---|---|---|---|
| 10 | $\mathbf{24.47 \pm 0.19}$ | 1.24e-3 $\pm$ 2.38e-4 | 1.76e-3 $\pm$ 3.37e-4 |
| 20 | 27.46 $\pm$ 0.08 | 6.52e-4 $\pm$ 2.59e-4 | 9.22e-4 $\pm$ 3.66e-4 |
| 30 | 30.43 $\pm$ 0.50 | $\mathbf{3.69e\text{-}4 \pm 4.33e\text{-}5}$ | $\mathbf{5.23e\text{-}4 \pm 6.13e\text{-}5}$ |
| 40 | 33.64 $\pm$ 0.41 | 3.86e-4 $\pm$ 9.37e-5 | 5.46e-4 $\pm$ 1.32e-4 |

with the initial condition

$$u(x,0) = \sin(\pi x) + \frac{1}{2}\sin(4\pi x), \quad x \in \Omega, \tag{27b}$$

$$u_t(x,0) = 0, \quad x \in \Omega, \tag{27c}$$

and the boundary condition

$$u(0,t) = u(1,t) = 0 \quad t \in [0,1]. \tag{27d}$$

Table 10: Advection equation ($\beta = 10$): hyperparameter optimization for PINN (LBFGS) varying the number of interior points. The mean is computed over 3 seeds.

| Interior points | Training time (s) | RMSE | Relative $L^2$ error |
|---|---|---|---|
| 500 | **25.76 ± 0.29** | 4.10e-4 ± 7.20e-5 | 5.80e-4 ± 1.01e-4 |
| 1000 | 27.44 ± 0.25 | 3.72e-4 ± 4.06e-5 | 5.27e-4 ± 5.74e-5 |
| 1500 | 29.61 ± 0.16 | 5.68e-4 ± 1.97e-4 | 8.03e-4 ± 2.79e-4 |
| 2000 | 30.43 ± 0.50 | **3.69e-4 ± 4.33e-5** | **5.23e-4 ± 6.13e-5** |

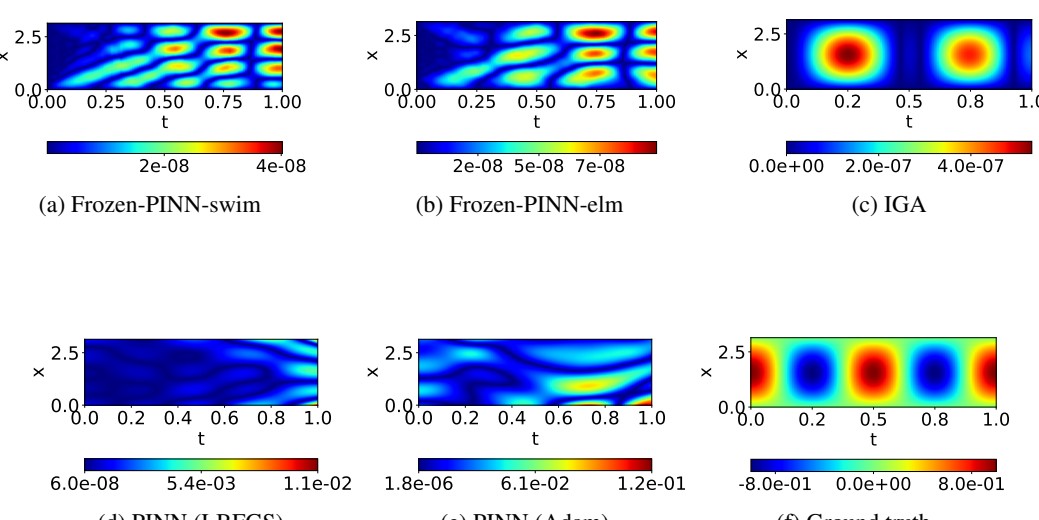

Figure 16: The classical Euler-Bernoulli beam equation: absolute error plots and ground truth.

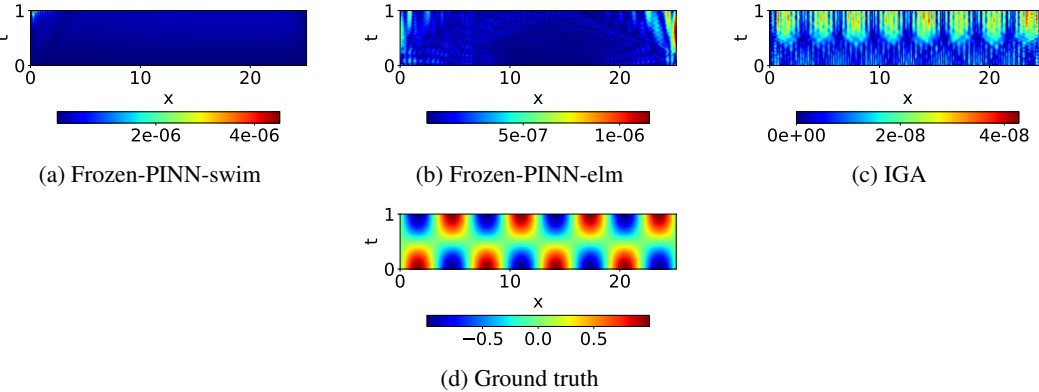

Figure 17: The Euler-Bernoulli beam equation on Winkler foundation: absolute error plots and ground truth.

The analytical solution of the problem is

$$u(x, t) = \sin(\pi x)\cos(2\pi t) + \frac{1}{2}\sin(4\pi x)\cos(8\pi t). \tag{27e}$$

Table 11: The classical Euler-Bernoulli beam equation for the results in Table 1: summary of all hyperparameters.

|  | Parameter | Value |
|---|---|---|
| Frozen-PINN-elm (low-precision) | Number of hidden layers | 2 |
|  | Hidden layer width | 50 |
|  | SVD-cutoff | $10^{-6}$ |
|  | Activation | `tanh` |
|  | $L^2$-regularization | $10^{-6}$ |
|  | Loss | mean-squared error |
|  | boundary condition strategy | augmented ODE |
| Frozen-PINN-elm (high-precision) | Number of hidden layers | 2 |
|  | Hidden layer width | 100 |
|  | SVD-cutoff | $10^{-12}$ |
|  | Activation | `tanh` |
|  | $L^2$-regularization | $10^{-10}$ |
|  | Loss | mean-squared error |
|  | boundary condition strategy | augmented ODE |
| IGA | Number of nodes | 27 |
|  | Degree of polynomials | 9 |
|  | Number of basis functions | 33 |
| PINN | Number of hidden layers | 4 |
|  | Layer width | 20 |
|  | Activation | `tanh` |
|  | Optimizer | LBFGS (ADAM) |
|  | Epochs | 15000 (30000) |
|  | Loss | mean-squared error |
|  | Learning rate | 0.1 |
|  | Batch size | 2000 |
|  | Parameter initialization | Xavier (Glorot & Bengio, 2010) |
|  | Loss weights, $\lambda_1, \lambda_2$ | 0.1, 1 |
|  | # Interior points | 10000 |
|  | # Initial and boundary points | 6000 |

Table 12: The Euler-Bernoulli beam equation on Winkler foundation for the results in Table 1: summary of all hyperparameters.

|  | Parameter | Value |
|---|---|---|
| Frozen-PINN-elm (low-precision) | Number of hidden layers | 2 |
|  | Hidden layer width | 200 |
|  | SVD-cutoff | $10^{-6}$ |
|  | Activation | `tanh` |
|  | $L^2$-regularization | $10^{-6}$ |
|  | Loss | mean-squared error |
|  | boundary condition strategy | augmented ODE |
| Frozen-PINN-swim (high-precision) | Number of hidden layers | 2 |
|  | Hidden layer width | 400 |
|  | SVD-cutoff | $10^{-10}$ |
|  | Activation | `tanh` |
|  | $L^2$-regularization | $10^{-10}$ |
|  | Loss | mean-squared error |
|  | boundary condition strategy | augmented ODE |
| IGA | Number of nodes | 60 |
|  | Degree of polynomials | 6 |
|  | Number of basis functions | 63 |
| PINN | Number of hidden layers | 4 |
|  | Layer width | 200 |
|  | Activation | `tanh` |
|  | Optimizer | LBFGS |
|  | Epochs | 10000 |
|  | Loss | mean-squared error |
|  | Learning rate | 0.1 |
|  | Batch size | 500 |
|  | Parameter initialization | Xavier (Glorot & Bengio, 2010) |
|  | Loss weights, $\lambda_1, \lambda_2$ | 1, 1 |
|  | # Interior points | 10000 |
|  | # Initial and boundary points | 1500 |

In addition, we present a significantly challenging scenario involving a multi-scale solution by employing the initial condition

$$u(x, 0) = \sin(\pi x) + \frac{1}{2}\sin(4\pi x) + \frac{1}{4}\sin(9\pi x), \quad x \in \Omega, \tag{27f}$$

$$u_t(x, 0) = 0, \quad x \in \Omega, \tag{27g}$$

for which the corresponding analytical solution is given by

$$u(x, t) = \sin(\pi x)\cos(2\pi t) + \frac{1}{2}\sin(4\pi x)\cos(8\pi t) + \frac{1}{4}\sin(9\pi x)\cos(18\pi t). \tag{27h}$$

**Comparison of results:** In Table 1, we compare the Frozen-PINN-swim result of Equation (27e) with other PINN methods from Hao et al. (2024), see Figure 18. The hyperparameters used for this experiment can be found in Table 13. Additionally, we demonstrate the capability of our proposed method by solving the multi-scale problem as in Equation (27h) using the same neural network architecture, and the relative $L^2$ error is less than $10^{-5}$. The results are shown in Figure 19.

Table 13: Wave equation: Hyper-parameters for the result in Table 1.

| | Parameter | Value |
|---|---|---|
| Frozen-PINN-swim | Number of hidden layers | 2 |
| | Hidden layer width | 400 |
| | Outer basis functions | 30 |
| | Activation | `tanh` |
| | $L^2$-regularization | $10^{-12}$ |
| | Loss | mean-squared error |
| | boundary condition strategy | boundary-compliant layer |

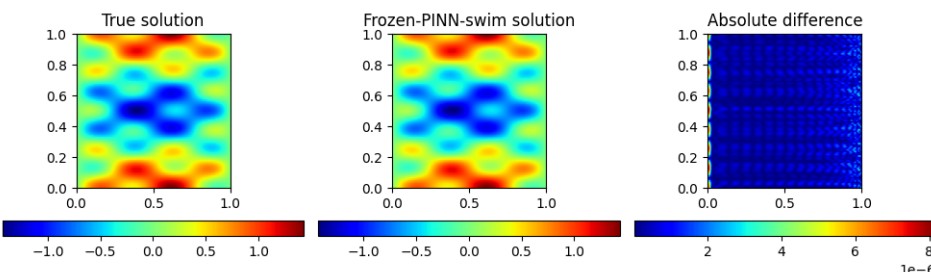

Figure 18: Wave equation Equation (27e): Ground truth, Frozen-PINN-swim solution, absolute error.

### C.4 BURGERS

**Problem Setup:** The Burgers' equation in different settings is used to model traffic flows, large-scale structure formation in cosmology, and shock formation in inviscid flows, among other applications (Bonkile et al., 2018). The inviscid Burgers' equation is a nonlinear PDE, which can form shock waves. We consider Burgers' equation on $\Omega = [-1, 1]$ for time $t \in (0, 1]$ from Raissi et al. (2019), given by:

$$u_t + uu_x - (0.01/\pi)u_{xx} = 0, \quad x \in \Omega, \quad t \in [0, 1], \tag{28a}$$

with initial and boundary conditions

$$u(0, x) = -\sin(\pi x), \quad x \in \Omega, \tag{28b}$$

$$u(t, -1) = u(t, 1) = 0 \quad t \in [0, 1]. \tag{28c}$$

We consider the analytical solution provided by Basdevant et al. (1986).

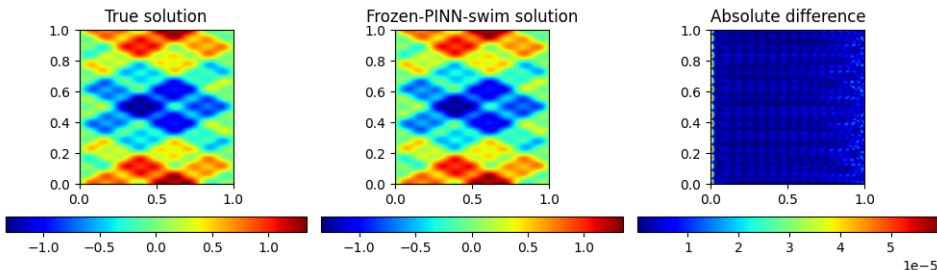

(a) (Left): Ground truth, (Middle): Frozen-PINN-swim solution, (Right): point-wise absolute error.

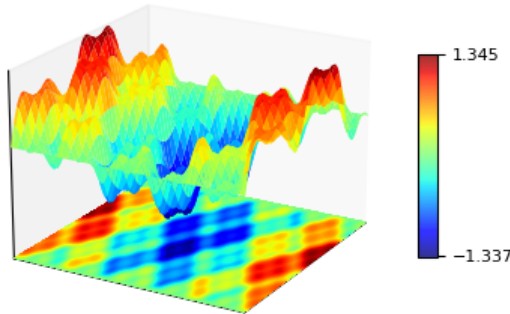

(b) 3D visualization of the multi-scale wave solution.

Figure 19: Wave equation Equation (27h): Ground truth, Frozen-PINN-swim solution, absolute error, and multi-scale solution visualization.

**Why Frozen-PINN-elm can't resolve shocks in PDE solutions?** To accurately resolve PDE solutions with sharp gradients, it is essential to: (a) construct basis functions with steep gradients, and (b) position them particularly near the shock regions within the domain. Figure 20a (Right) illustrates why solution- or data-agnostic ELM basis functions make it very difficult for Frozen-PINN-elm to capture the sharp features in the solution, particularly at the center of the domain, due to the exponentially small probability of sampling steep basis functions (Huang et al., 2006). While sampling weights from a wider uniform distribution, as discussed by Calabrò et al. (2021) for linear PDEs, can increase the probability of sampling steeper basis functions, it offers no spatial control over their placement.

**Ablation studies:** We describe additional details in solving the Burgers' equation with various neural PDE solvers in Table 14 and Table 15. The results of the ablation study with the number of neurons in the hidden layer for Frozen-PINN-swim are presented in Table 16. We observe that starting with a width of 1200, the error decreases for a width up to 600 and increases again below 600. We believe that for widths lower than 600, the network capacity seems to be the reason for the loss of accuracy. For very high widths, the regularization constant has to be kept to a higher value to avoid overfitting. Otherwise, the ODE system becomes highly stiff. With this high regularization constant, the training becomes stable, but it affects the training accuracy. We do not include results for Frozen-PINN-elm as it fails on all widths, as it is not able to capture the sharp shocks and exhibits Gibbs phenomenon (Gottlieb & Shu, 1997), which is explained in detail in Appendix C.4.1.

We also perform an ablation study for the SVD layer for Frozen-PINN-swim. Please refer to Table 17. The ablation study reveals that the SVD layer compresses the number of neurons by a factor of $1.58$, which reduces the output computation time by a factor of 7 for almost the same accuracy. This highlights the utility of the SVD layer.

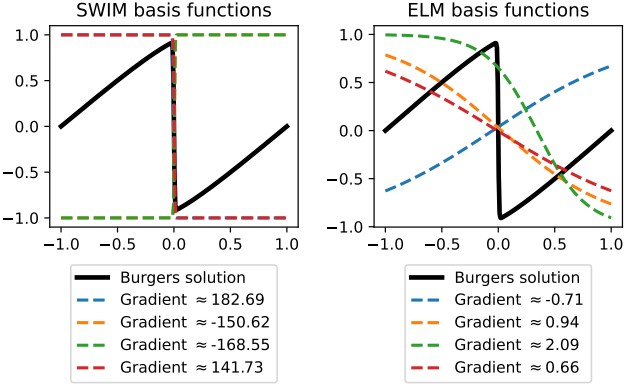

(a) (Left): Re-sampled SWIM basis functions (with steep gradients centered around the shock) at $t = 0.66$, (Right): data-agnostic ELM basis functions.

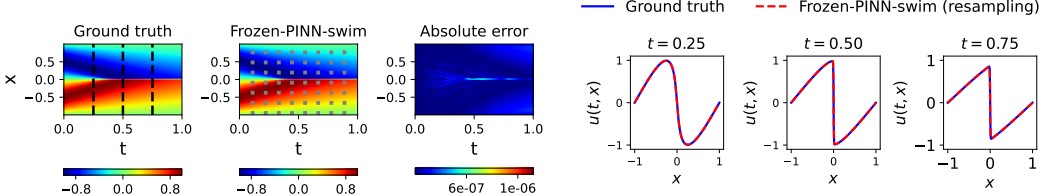

(b) (Left): Ground truth, (Middle): Frozen-PINN-swim solution, where black and gray dashed lines mark time snapshots selected for a comparison (in (d) on the right) and the collocation points resampling times, respectively, (Right): point-wise absolute error.

(c) Comparison between Frozen-PINN-swim and numerical solutions at three time instances.

Figure 20: Illustration of experimental results for the Burgers' equation.

**Comparison of results:** Figures 20b and 20c present a comparison between the Frozen-PINN-swim solution and the numerical solution from Raissi et al. (2019), validating the ability of Frozen-PINN-swim to resolve shocks with high accuracy. We demonstrate with a snapshot of the Burgers' solution that SWIM basis functions exhibit a rapid exponential decay of error with increasing network width, where Fourier and Chebyshev basis functions suffer from the Gibbs phenomenon Gottlieb & Shu (1997) (See Figure 22, Figure 23, Appendix C.4.1). Figure 21 shows the absolute errors obtained with the PINN, Causal PINN, and IGA methods.

### C.4.1 COMPARISON WITH CLASSICAL SPECTRAL METHODS

In this section, we study how the basis functions sampled with SWIM and ELM approaches perform in comparison to the basis functions typically employed in traditional spectral methods. We try to approximate a single snapshot of the solution to the Burgers' equation, which has a locally steep gradient. If a method fails to even approximate a single snapshot well enough, it is highly unlikely to achieve better results in approximating the entire space-time solution of the PDE.

Figure 22 shows the approximation of the Burgers' equation solution at $t = 0.99$, using SWIM basis functions, ELM basis functions, Fourier series, and Chebyshev polynomials, respectively. The number of basis functions is 102 for all methods. Figure 23 shows the approximation error using a different number of basis functions. We can see that for ELM basis functions, Fourier basis functions, and Chebyshev polynomials, there are oscillations near the shock, and the error is large compared to the SWIM basis functions, where we are able to take advantage of resampling data points and sampling appropriate basis functions in order to adapt to the target function well. Note that in this experiment, the weights for the ELM basis functions are sampled from a Gaussian distribution

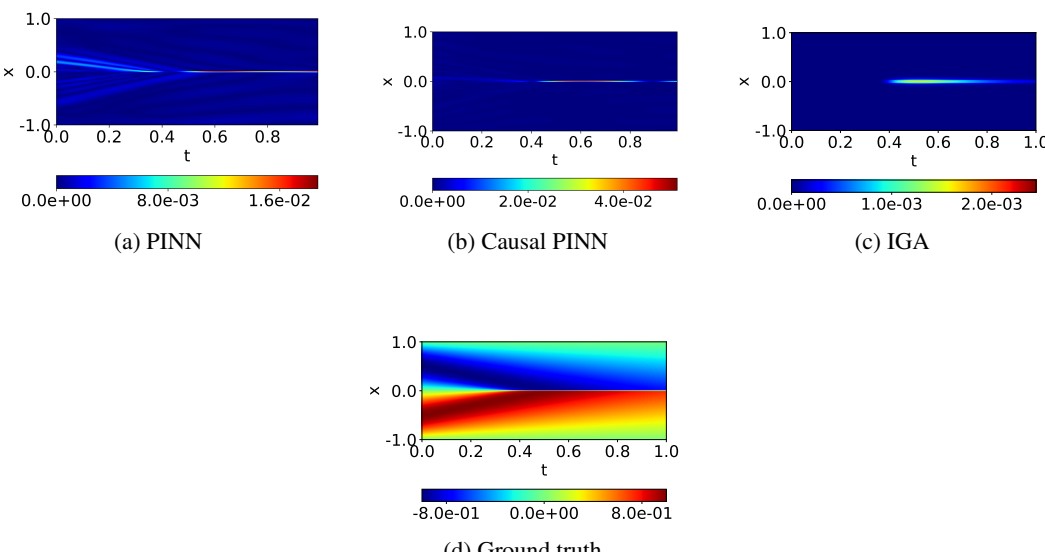

(a) PINN        (b) Causal PINN        (c) IGA

(d) Ground truth

Figure 21: Burgers' equation: absolute error plots and ground truth.

Table 14: Burgers' equation: Summary of hyper-parameters for Frozen-PINNs (see Table 1).

| | Parameter | Value |
|---|---|---|
| Frozen-PINN-swim (low-precision) | Number of hidden layers | 2 |
| | Hidden layer width | [300] |
| | Activation | `tanh` |
| | $L^2$-regularization | $[10^{-6}, 10^{-7}, \mathbf{10^{-8}}, 10^{-10}, 10^{-12}]$ |
| | svd cutoff | $10^{-8}$ |
| | Loss | mean-squared error |
| | # collocation points (space) | [600] |
| | # sampling points | [1000] |
| | ODE solver tolerance | $10^{-3}$ |
| | # time windows for resampling | 9 |
| | boundary condition strategy | augmented ODE |
| Frozen-PINN-swim (high-precision) | Number of hidden layers | 2 |
| | Hidden layer width | [450] |
| | Activation | `tanh` |
| | $L^2$-regularization | $[10^{-6}, 10^{-7}, 10^{-8}, 10^{-10}, 10^{-12}, \mathbf{10^{-13}}]$ |
| | svd cutoff | $5 \times 10^{-11}$ |
| | Loss | mean-squared error |
| | # collocation points (space) | [1000] |
| | # sampling points | [6000] |
| | ODE solver tolerance | $10^{-6}$ |
| | # time windows for resampling | 9 |
| | boundary condition strategy | augmented ODE |
| Frozen-PINN-elm | Number of hidden layers | 2 |
| | Hidden layer width | [2000] |
| | Activation | `tanh` |
| | $L^2$-regularization | $[10^{-6}, \mathbf{10^{-7}}, 10^{-8}, 10^{-10}, 10^{-12}]$ |
| | Loss | mean-squared error |
| | # collocation points (space) | [3000] |
| | # sampling points | [6000] |
| | boundary condition strategy | augmented ODE |

with a standard deviation of 10 in order to increase the number of basis functions. The biases are sampled from a uniform distribution in $[-10, 10]$. For the Fourier basis functions and Chebyshev polynomials, we use equispaced grid points. We also experimented with quadrature points and placed more points near the steep gradient in an attempt to mitigate the oscillations associated with the Gibbs phenomenon and the Runge phenomenon, but it did not lead to any significant improvement in the

Table 15: Burgers' equation (see Table 1): Network hyper-parameters used for PINN, Causal PINN, and IGA.

|  | Parameter | Value |
|---|---|---|
| PINN | Number of hidden layers | 9 |
|  | Layer width | 20 |
|  | Activation | `tanh` |
|  | Optimizer | LBFGS |
|  | Epochs | 10000 |
|  | Loss | mean-squared error |
|  | Learning rate | 0.1 |
|  | Batch size | 200 |
|  | Parameter initialization | Xavier (Glorot & Bengio, 2010) |
|  | Loss weights, $\lambda_1, \lambda_2$ | 1, 1 |
|  | # Interior points | 10000 |
|  | # Initial and boundary points | 600 |
| Causal PINN | Number of hidden layers | 9 |
|  | Layer width | 20 |
|  | Activation | `tanh` |
|  | Optimizer | ADAM followed by LBFGS |
|  | ADAM Epochs | 5000 |
|  | LBFGS Epochs | 10000 |
|  | Loss | mean-squared error |
|  | Learning rate | 0.1 |
|  | Batch size | 200 |
|  | Parameter initialization | Xavier (Glorot & Bengio, 2010) |
|  | Loss weights, $\lambda_1, \lambda_2$ | 1, 1 |
|  | # Interior points | 40000 |
|  | # Initial and boundary points | 600 |
|  | Causality parameter, $\epsilon$ | 5 |
| IGA | Number of nodes | 750 |
|  | Degree of polynomials | 9 |
|  | Number of basis functions | 756 |

Table 16: Burgers' equation: ablation study for the network width for Frozen-PINN-swim.

| Width | Relative $L^2$ error |
|---|---|
| 240 | 4.27e-4 |
| **550** | **2.27e-7** |
| 800 | 2.78e-6 |
| 1200 | 1.54e-6 |

Table 17: Burgers' Equation: Ablation Study for the SVD layer with Frozen-PINN-swim.

|  | With SVD layer | Without SVD layer | Ratio |
|---|---|---|---|
| Number of neurons | 500 | 316 | Width Compression $\approx 1.58$x |
| Time (s) | 141.5 | 989.84 | Speed-up $\approx 7$x |
| Rel. $L_2$ error | 3.34e-4 | 3.28e-4 | - |

results. This conclusively demonstrates that SWIM basis functions perform better than traditional bases used in spectral methods in accurately resolving shocks.

## C.5 NONLINEAR DIFFUSION EQUATION

**Problem Setup:** The non-linear diffusion equation in different forms is used to model the spread of populations, bacterial colonies, and forest fires, as well as to model groundwater and ice-sheet flow in glaciers, and mass diffusion in reactive flows (Li et al., 2001). We consider a two-dimensional nonlinear diffusion equation given by

$$u_t - u\Delta u = f(x, y, t), \quad (x, y) \in \Omega, \quad t \in [0, 1], \tag{29a}$$

with a forcing function

$$f(x, y, t) = 5e^{-t} \sin(\pi x) y^{-3} \left(-1 + e^{-t} \sin(\pi x) y^{-5} \left(-12 + \pi^2 y^2\right)\right) \tag{29b}$$

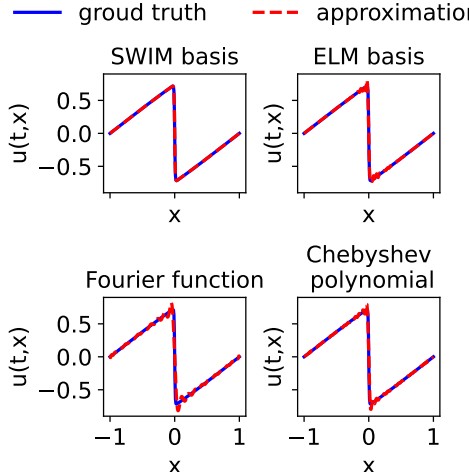 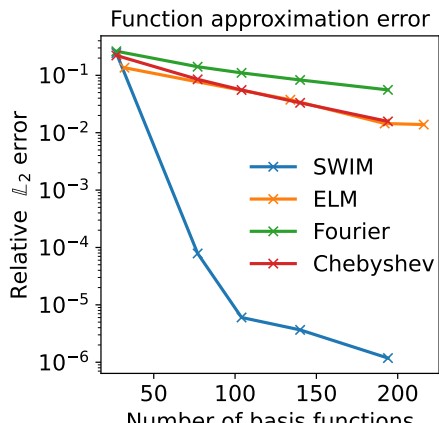

Figure 22: Approximation of Burgers' equation solution at $t = 0.99$ with four types of basis functions. The number of basis functions in all cases is 102. Oscillations can be seen near the steep gradient for the methods using ELM basis functions, Fourier functions, and Chebyshev polynomials.

Figure 23: Approximation error for four types of basis functions. Here, we directly fit the Burgers' equation solution at $t = 0.99$. The approximation error decreases as we increase the number of basis functions, and the SWIM basis functions yield the best result among all methods.

on a complicated geometry inspired by a tree-like pattern occurring during the controlled shaping of fluids Islam & Gandhi (2017). The initial condition and time-dependent Dirichlet boundary conditions are obtained from the constructed solution of the PDE

$$u(x, y, t) = 5e^{-t}\sin(\pi x)y^{-3}, \quad (x, y) \in \Omega, \quad t \in [0, 1]. \tag{29c}$$

The training is performed on 1500 data points in the interior and boundary. We test the neural-PDE solvers with 5000 data points in the interior and on the boundary. The weights of the hidden layer for the Frozen-PINN-elm are sampled from the Gaussian distribution and biases from a uniform distribution in $[-1, 1]$. For our approach to handling time-dependent Dirichlet boundary conditions, please refer to Appendix B.2.5. The hyperparameters for various neural PDE solvers are outlined in Table 18. Figure 24 shows the mesh generated for the FEM and the sampled collocation points for the neural PDE solvers. For the mesh we consider for this problem (see Figure 24), we could not improve the accuracy further with FEM by using higher-order polynomial basis functions. While mesh refinement is possible, it's time-consuming, and our method avoids this by working directly with point clouds.

**Ablation studies:** The ablation study for the number of neurons in the hidden layer of the network for Frozen-PINN-elm and Frozen-PINN-swim is presented in Table 19. For PINN, the results for the ablation studies for the width of the network and the number of data points are included in Table 20, Table 21. Additionally, we perform an ablation study for the SVD layer to demonstrate its impact on the computation time saved in Table 22. Particularly, we observe that with the SVD layer, the number of basis functions (width after the SVD layer) is reduced by up to 22x for Frozen-PINN-elm and up to 1.5x for Frozen-PINN-swim, and we obtain substantial speed-ups (more than a factor of 50) in the computation time.

**Comparison of results:** The comparison of training times and errors is presented in Table 1. Figure 25 shows the ground truth and Figure 26 shows the error plots with all approaches.

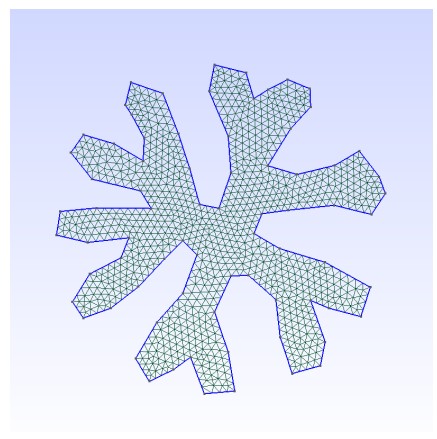
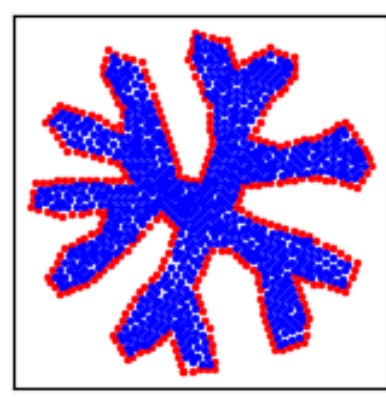

(a) Generated Mesh: FEM

(b) Sampled collocation points: Neural PDE solvers

Figure 24: Advantages of mesh-free methods: (a) For mesh-based methods, a complicated mesh must be constructed, whereas (b) for neural PDE solvers, one can easily sample arbitrary points in the interior (blue) and on the boundary (red) of the domain and work directly with point clouds.

Table 18: Non-linear diffusion equation (see Table 1): Summary of hyper-parameters.

|  | Parameter | Value |
|---|---|---|
| Frozen-PINN-elm (low-precision) | Number of hidden layers | 2 (nonlinear and SVD layer) |
|  | Hidden layer width | 350 |
|  | Activation | `tanh` |
|  | $L^2$-regularization | $5 \times 10^{-11}$ |
|  | SVD cutoff | $5 \times 10^{-11}$ |
|  | ODE solver tolerance | $10^{-6}$ |
|  | Loss | mean-squared error |
|  | boundary condition strategy | augmented ODE |
| Frozen-PINN-swim (high-precision) | Number of hidden layers | 2 (nonlinear and SVD layer) |
|  | Hidden layer width | 500 |
|  | Activation | `tanh` |
|  | $L^2$-regularization | $10^{-15}$ |
|  | SVD cutoff | $10^{-15}$ |
|  | ODE solver tolerance | $10^{-6}$ |
|  | Loss | mean-squared error |
|  | boundary condition strategy | augmented ODE |
| FEM | Number of entities | 154 |
|  | Number of nodes | 1193 |
|  | Number of elements | 2070 |
|  | Type of elements | Lagrange |
|  | Shape of elements | triangle |
|  | Degree of polynomials | 1 |
|  | Number of basis functions | 1193 |
|  | Solver | Newton solver |
|  | Timestep size | 0.001 |
| PINN | Number of hidden layers | 4 |
|  | Layer width | $[10, 20, \mathbf{30}, 40]$ |
|  | Activation | `tanh` |
|  | Optimizer | LBFGS & ADAM |
|  | Epochs | 10000 |
|  | Loss | mean-squared error |
|  | Learning rate | 0.01 |
|  | Batch size | 1000 |
|  | Parameter initialization | Xavier (Glorot & Bengio, 2010) |
|  | Loss weights, $\lambda_1, \lambda_2$ | 0.01, 1 |
|  | # Interior points | $[8790, 1760, \mathbf{880}, 440]$ |
|  | # Initial and boundary points | $[3140, 630, \mathbf{320}, 160]$ |

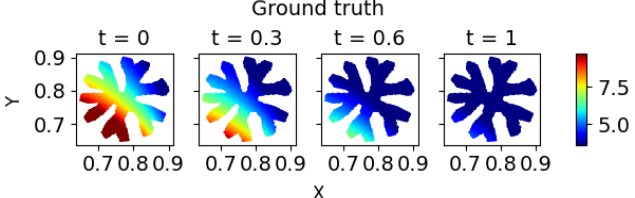

Figure 25: Non-linear diffusion equation: ground truth.

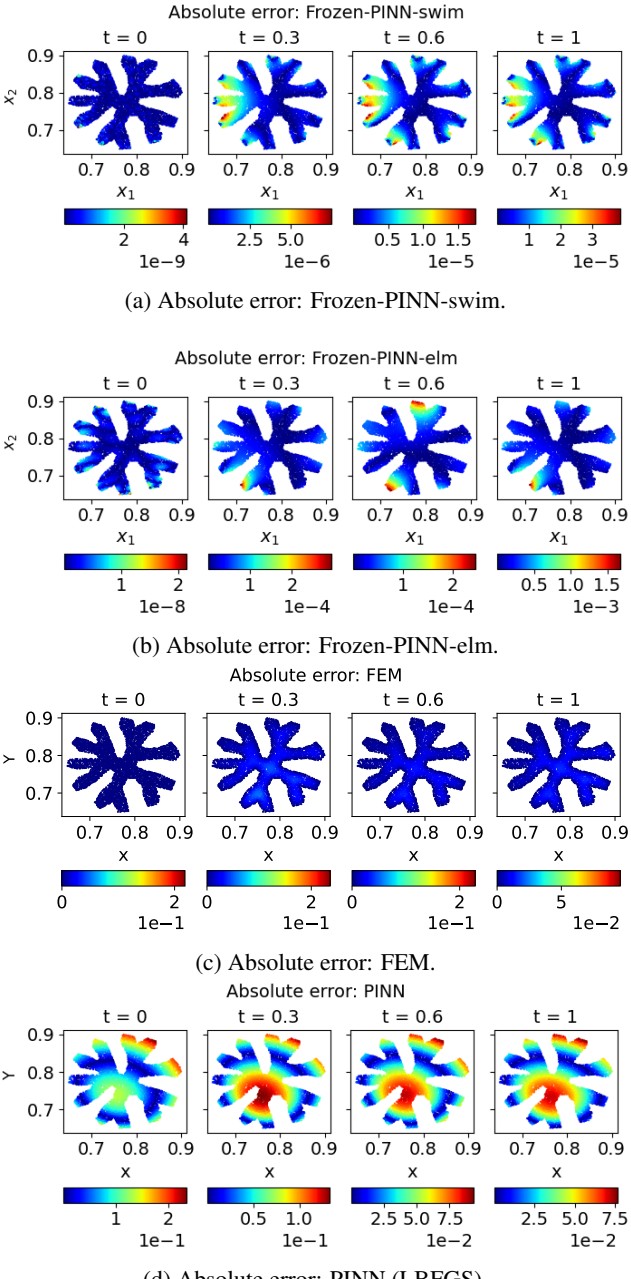

(a) Absolute error: Frozen-PINN-swim.

(b) Absolute error: Frozen-PINN-elm.

(c) Absolute error: FEM.

(d) Absolute error: PINN (LBFGS).

Figure 26: Non-linear diffusion equation: absolute error plots and ground truth at four-time instants.

Table 19: Non-linear diffusion equation: ablation study for the network width for Frozen-PINN-swim and Frozen-PINN-elm. The mean is computed over 3 seeds.

| Width | Relative $L^2$ error (Frozen-PINN-swim) | Relative $L^2$ error (Frozen-PINN-elm) |
|-------|------------------------------------------|----------------------------------------|
| 200 | 1.34e-4 | 4.92e-3 |
| 300 | 5.07e-6 | 3.13e-5 |
| 400 | 2.88e-6 | **1.02e-5** |
| 500 | **3.02e-7** | 1.52e-5 |

Table 20: Non-linear diffusion equation: hyperparameter optimization for PINN varying layer width. The mean is computed over 3 seeds.

| Layer width | Training time (s) | RMSE | Relative $L^2$ error |
|-------------|-------------------|------|----------------------|
| 10 | **61.09 ± 1.62** | 4.11e-2 ± 2.04e-3 | 1.50e-2 ± 7.48e-4 |
| 20 | 68.05 ± 1.56 | 3.74e-2 ± 1.04e-3 | 1.37e-2 ± 3.82e-4 |
| 30 | 76.01 ± 0.57 | **3.67e-2 ± 1.03e-3** | **1.34e-2 ± 3.78e-4** |
| 40 | 82.43 ± 0.45 | 3.76e-2 ± 1.69e-3 | 1.37e-2 ± 6.21e-4 |

Table 21: Non-linear diffusion equation: hyperparameter optimization for PINN varying interior points.

| Interior points | Training time (s) | RMSE | Relative $L^2$ error |
|-----------------|-------------------|------|----------------------|
| 600 | **65.08 ± 4.23** | 3.74e-2 ± 1.04e-3 | 1.37e-2 ± 3.82e-4 |
| 1200 | 98.48 ± 3.78 | 3.51e-2 ± 6.67e-4 | 1.28e-2 ± 2.44e-4 |
| 2390 | 143.31 ± 5.50 | **3.34e-2 ± 6.53e-4** | **1.22e-2 ± 2.38e-4** |

Table 22: Non-linear diffusion equation: Ablation study of the SVD layer in Frozen-PINN-swim and Frozen-PINN-elm. We report $\infty$ for runtimes exceeding 3 hours. Two variants of Frozen-PINN-elm are shown: Frozen-PINN-elm-accurate (higher accuracy, longer runtime) and Frozen-PINN-elm-fast (lower runtime, with error comparable to or better than PINNs, enabling fair comparison). The ratio of the hidden layer width to the SVD layer width is denoted by $C_r$.

| Method | Quantity | With SVD layer | Without SVD layer | Ratio |
|--------|----------|----------------|-------------------|-------|
| Frozen-PINN-elm-accurate | Width | 62 | 300 | $C_r \approx 22.8$x |
| | Time (s) | 60.98 | 7087.38 | Speed-up $\approx 52$x |
| | Rel. $L_2$ error | 6.49e-8 | 1.02e-6 | - |
| Frozen-PINN-elm-fast | Width | 35 | 300 | $C_r \approx 8.5$x |
| | Time (s) | 30.57 | $\infty$ | Speed-up $\infty$ |
| | Rel. $L_2$ error | 5.12e-5 | - | - |
| Frozen-PINN-swim | Width | 316 | 500 | $C_r \approx 1.5$x |
| | Time (s) | 328.03 | $\infty$ | Speed-up $\infty$ |
| | Rel. $L_2$ error | 2e-6 | - | - |

## C.6 NONLINEAR REACTION-DIFFUSION EQUATION

**Problem Setup:** The non-linear reaction–diffusion equation models biological pattern formation, such as Zebra stripes, fish spots, in myriad chemical reactions, and flame propagation during combustion (Britton, 1986; Lam & Lou, 2022).

In this benchmark from Zang et al. (2020), we consider a five-dimensional nonlinear diffusion equation given by

$$u_t - \Delta u - u^2 = f(x,t), \quad x \in \Omega \subset \mathbb{R}^d, \quad t \in [0,1], \tag{30a}$$

$$f(x,t) = (\pi^2 - 2)\sin\left(\frac{\pi}{2}x_1\right)\cos\left(\frac{\pi}{2}x_2\right)e^{-t} - 4\sin^2\left(\frac{\pi}{2}x_1\right)\cos^2\left(\frac{\pi}{2}x_2\right)e^{-2t}, \tag{30b}$$

on the domain $\Omega = [-1,1]^d$. The initial condition and time-dependent Dirichlet boundary conditions are obtained from the constructed solution of the PDE

$$u(x,t) = 2\sin\left(\frac{\pi}{2}x_1\right)\cos\left(\frac{\pi}{2}x_2\right)e^{-t}. \tag{31}$$

Note that the solution is independent of three out of five dimensions. The training is performed on 1000 data points in the interior and 1000 data points on the boundary. The test data set is generated the same way as in Zang et al. (2020) to evaluate the weak adversarial networks. In particular, to compute the error in the 5-dimensional domain, we use a mesh of size $100 \times 100$ for the two coordinate directions in which the solution changes (here, $(x_1, x_2)$) and uniformly randomly sample the other coordinates (here, $(x_3, x_4, x_5)$) in the domain. The hidden layer weights for the Frozen-PINN-elm are sampled from the standard Gaussian distribution and biases from a uniform distribution in $[-1, 1]$. Please refer to Appendix B.2.5 for our approach to handling time-dependent Dirichlet boundary conditions.

The sampling strategy for basis functions described in Section 3.7 using projected pairs of data points substantially improves efficiency and accuracy. Our approach requires 20 times fewer training points in the interior compared to Zang et al. (2020) while simultaneously achieving a relative $L^2$ error more than two orders of magnitude lower.

**Details on the sampling well-oriented basis functions:** For each pair of collocation points in the spatial domain $x^{(1)}, x^{(2)} \in \Omega$, we project the vector $x^{(2)} - x^{(1)}$ onto the two-dimensional hyper-plane spanned by the gradient of the initial solution at $x^{(1)}, x^{(2)}$ and use the projected points as the new pair of points $\hat{x}^{(1)}, \hat{x}^{(2)} \in \Omega$. Since $\hat{x}^{(2)} - \hat{x}^{(1)}$ always points in the direction of the gradient of the initial solution, this allows the SWIM algorithm to embed directional information into basis functions, unlike PINNs and ELMs, which lack this control. This idea is illustrated in Figure 5b.

**Ablation studies:** The ablation study for the number of neurons in the hidden layer of the network for Frozen-PINN-elm and Frozen-PINN-swim is presented in Table 23. We further validate the efficiency of sampling basis functions using projected pairs of data points with the Frozen-PINN-swim approach by performing an ablation study varying the number of internal collocation points in Table 24. Our results show that using just 1,000 data points achieves training errors that are nearly identical to those with 20,000 points. This highlights the effectiveness of the projection trick in reducing the need for excessive collocation points, thereby significantly lowering computational cost without compromising accuracy. Additionally, we perform an ablation study for the SVD layer to demonstrate its impact on the computation time saved in Table 25. We observe that with the SVD layer, the number of basis functions (width after the SVD layer) is reduced by up to 1.57x for Frozen-PINN-swim, and we obtain substantial speed-ups by a factor of 4.1x in the computation time.

Table 23: Non-linear reaction diffusion equation: ablation study for the network width for Frozen-PINN-swim and Frozen-PINN-elm. The mean is computed over 3 seeds.

| Width | Frozen-PINN-swim (with projection) | Frozen-PINN-swim | Frozen-PINN-elm |
|-------|-----------------------------------|------------------|-----------------|
| 100   | 1.44e-4                           | 7.65e-2          | 2.08e-1         |
| 400   | 9.99e-5                           | 1.75e-2          | 6.37e-2         |
| 700   | 9.92e-5                           | 8.72e-3          | 3.65e-2         |
| 1000  | 9.87e-5                           | **5.70e-3**      | 2.58e-2         |
| 2000  | 9.86e-5                           | 8.62e-3          | **1.67e-2**     |
| 4000  | **9.86e-5**                       | 9.98e-3          | 3.68e-2         |

Table 24: Non-linear reaction diffusion equation: ablation study for the number of interior collocation points for Frozen-PINN-swim. The mean is computed over 3 seeds, and the network width is 400.

| Interior points | Training time (s) | RMSE | Relative $L^2$ error |
|-----------------|-------------------|------|----------------------|
| 1000  | **12.43** | 3.68e-5 +- 6.78e-12 | 9.99e-5 +- 1.84e-11 |
| 2000  | 102.74    | 3.67e-5 +- 3.00e-10 | 9.90e-5 +- 8.16e-10 |
| 20000 | 689.81    | **3.63e-5 +- 2.02e-9** | **9.87e-5 +- 5.49e-09** |

Table 25: Non-linear reaction diffusion equation: Ablation Study for the SVD layer with Frozen-PINN-swim.

|  | With SVD layer | Without SVD layer | Ratio |
|---|---|---|---|
| Number of neurons | 254 | 400 | Width Compression $\approx 1.57$x |
| Time (s) | 12.86 | 53.51 | Speed-up $\approx 4.1$x |
| Rel. $L_2$ error | 9.99e-5 | 9.99e-5 | - |

**Comparison of results:** The exact architectures and comparison of training times and errors are presented in Table 26 and Table 27. We observe that Frozen-PINN-swim with the projected pairs of points (Frozen-PINN-swim-p) far outperforms all the other approaches by around 2 orders of magnitude, while simultaneously being $9 - 50$ times faster. Figure 27 shows the errors with all approaches and the ground truth.

Table 26: Non-linear reaction diffusion equation: Summary of hyper-parameters.

|  | Parameter | Value |
|---|---|---|
| PINN | Number of hidden layers | 4 |
|  | Activation | `tanh` |
|  | Optimizer | LBFGS & ADAM |
|  | Epochs | 10000 |
|  | Loss | mean-squared error |
|  | Learning rate | 0.001 |
|  | Batch size | 1000 |
|  | Parameter initialization | Xavier (Glorot & Bengio, 2010) |
|  | Loss weights, $\lambda_1, \lambda_2$ | 0.01, 1 |
|  | # Interior points | 1000 |
|  | # Initial and boundary points | 1000 |
| Frozen-PINN-swim | Number of hidden layers | 2 (nonlinear and SVD layer) |
|  | Hidden layer width | 1000 |
|  | Activation | `tanh` |
|  | $L^2$-regularization | $10^{-10}$ |
|  | SVD cutoff | $10^{-10}$ |
|  | ODE solver tolerance | $10^{-4}$ |
|  | Loss | mean-squared error |
|  | boundary condition strategy | augmented ODE |
| Frozen-PINN-elm | Number of hidden layers | 2 (nonlinear and SVD layer) |
|  | Hidden layer width | 2000 |
|  | Activation | `tanh` |
|  | $L^2$-regularization | $10^{-10}$ |
|  | SVD cutoff | $10^{-10}$ |
|  | ODE solver tolerance | $10^{-4}$ |
|  | Loss | mean-squared error |
|  | boundary condition strategy | augmented ODE |
| Frozen-PINN-swim (with projection) | Number of hidden layers | 2 (nonlinear and SVD layer) |
|  | Hidden layer width | 700 |
|  | Activation | `tanh` |
|  | $L^2$-regularization | $10^{-10}$ |
|  | SVD cutoff | $10^{-10}$ |
|  | ODE solver tolerance | $10^{-4}$ |
|  | Loss | mean-squared error |
|  | boundary condition strategy | augmented ODE |

Table 27: Non-linear reaction diffusion equation: Summary of results.

| Method | Train time (s) | RMSE | Relative $L^2$ error | architecture |
|---|---|---|---|---|
| PINN (ADAM) | 171.43 | 1.25e-1 $\pm$ 6.60e-3 | 3.40e-1 $\pm$ 1.79e-2 | (6, 4×20, 1) |
| PINN (LBFGS) | 183.38 | 3.33e-2 $\pm$ 1.54e-2 | 3.33e-2 $\pm$ 1.54e-2 | (6, 4×20, 1) |
| Frozen-PINN-elm | 621.2 | 6.17e-3 $\pm$ 2.02e-4 | 1.67e-2 $\pm$ 5.49e-4 | (5, 2000, 1) |
| Frozen-PINN-swim | 117.24 | 2.09e-3 $\pm$ 1.91e-5 | 5.70e-3 $\pm$ 5.19e-5 | (5, 1000, 1) |
| **Frozen-PINN-swim (projection)** | **12.43** | **3.67e-5 $\pm$ 2.28e-9** | **9.99e-5 $\pm$ 6.21e-9** | (5, 700, 1) |

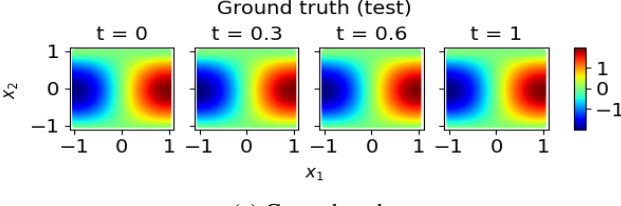

(a) Ground truth.

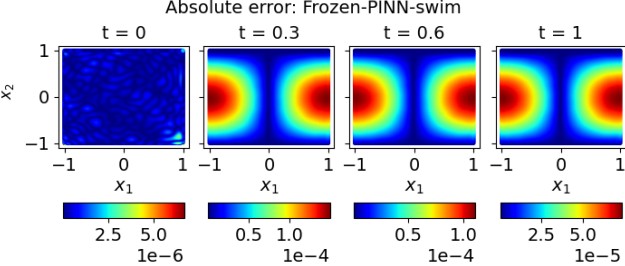

(b) Absolute error: Frozen-PINN-swim (with projected pairs of data points).

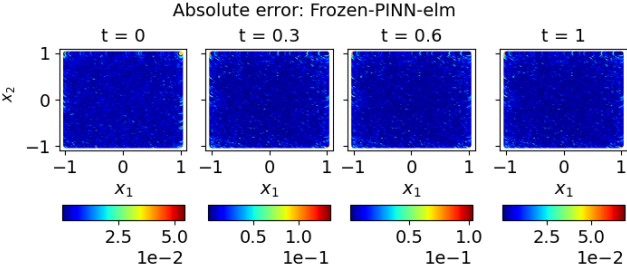

(c) Absolute error: Frozen-PINN-elm.

Figure 27: Non-linear reaction diffusion equation: absolute error plots and ground truth at four time instants.

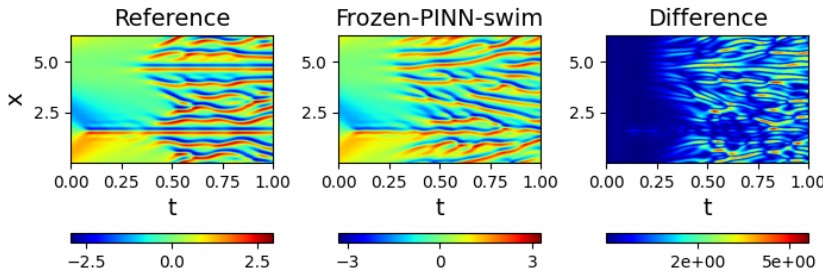

Figure 28: The Kuramoto-Sivashinsky equation: Reference solution, Frozen-PINN solution, and difference between two solutions.

### C.7 Kuramoto-Sivashinsky equation

**Problem Setup:** The Kuramoto-Sivashinsky equation is a fourth-order nonlinear PDE, which models the instabilities in flames and exhibits chaos. We consider the equation from Hao et al. (2024) of the form:

$$u_t + + \alpha u u_x + \beta u_{xx} + \gamma u_{xxxx} = 0, \quad x \in \Omega, \quad t \in T, \tag{32a}$$

with the parameters

$$\alpha = \frac{100}{16}, \beta = \frac{100}{16^2}, \gamma = \frac{100}{16^4}. \tag{32b}$$

The domain is $\Omega \times T = [0, 2\pi] \times [0, 5]$ for the experiment in Figure 6, or $\Omega \times T = [0, 2\pi] \times [0, 1]$ for the experiment in Figure 28. The initial condition is

$$u(x, 0) = \cos(x)(1 + \sin(x)), \quad x \in \Omega, \tag{32c}$$

and we apply the periodic boundary condition.

**Comparison of results:** In Figure 28, we compare our Frozen-PINN solution with the reference solution in Hao et al. (2024). The training time of Frozen-PINN is 2.7 seconds on CPU (averaged over 5 seeds). We also solve the PDE for a longer time span, as shown in Figure 6. The architecture of the model can be found in Table 28.

Table 28: Kuramoto-Sivashinsky equation: Hyper-parameters for the result in Figure 6 and Figure 28.

|  | Parameter | Value |
| --- | --- | --- |
| Frozen-PINN-swim | Number of hidden layers | 2 |
|  | Hidden layer width | 2000 |
|  | Outer basis functions | 100 |
|  | Activation | `tanh` |
|  | $L^2$-regularization | $10^{-12}$ |
|  | Loss | mean-squared error |
|  | boundary condition strategy | boundary-compliant layer |

### C.8 High-dimensional diffusion equation

**Problem setup:** High-dimensional diffusion plays an important role in various fields, including image processing, finance, and quantum mechanics (Sapiro, 2001; Janssen et al., 2013; Nagasawa, 2012).

We consider two benchmarks for the high-dimensional diffusion equation. In the first case, following (Wang & Dong, 2024), we solve the diffusion equation defined over the domain $\Omega = [-1, 1]^d$ and time interval $t \in (0, 1)$, for dimension $d \in \{3, 5, 7, 10\}$ given by

$$u_t - \Delta u = \left(\frac{1}{d} - 1\right) \cos\left(\frac{1}{d}\sum_{i=1}^{d} x_i\right) \exp\left(-t\right), \quad x \in \Omega, \quad t \in [0, 1], \tag{33}$$

with the exact solution given by

$$u(x, t) = \cos\left(\frac{1}{d}\sum_{i=1}^{d} x_i\right)\exp\left(-t\right).$$ (34)

The initial and boundary conditions are derived from Equation (34). For this high-dimensional diffusion equation, we use $16000$ training points in the interior and $4000$ points on the boundary randomly sampled using the Latin hypercube strategy. The test data contains $8000$ points in the interior and $2000$ points on the domain's boundary, which were also sampled with a Latin hypercube strategy.

For the second benchmark from He et al. (2023), we consider a 100-dimensional heat equation for $x \in B(0, 1)$, $t \in (0, 1)$ given by

$$u_t = \Delta u, \quad x \in B(0, 1), \quad t \in [0, 1],$$ (35a)

$$u(x, 0) = \frac{||x||^2}{2N}, \quad x \in B(0, 1)$$ (35b)

$$u(x, t) = t + \frac{1}{2N}, \quad x \in \partial B(0, 1), \quad t \in [0, 1],$$ (35c)

where the true solution is

$$u(x, t) = t + \frac{||x||^2}{2d}.$$ (36)

The value of $d$ is 100 and represents the dimension of the PDE. To solve the 100-dimensional heat equation, we generate $1000$ interior and $1000$ boundary training samples using Latin hypercube sampling. The test dataset comprises $8000$ interior points and $2000$ boundary points, also selected via Latin hypercube sampling.

**Extended discussion on results:** Note that, in general, it is extremely hard to accurately represent arbitrary 100-dimensional functions with a few hundred basis functions unless the solution is already in their span (e.g, approximating a linear solution with linear bases). The 100-dimensional heat equation benchmark from He et al. (2023), indeed, admits a true solution with very shallow gradients in space that varies linearly in time (see Equation (36)). Although this benchmark technically admits a quadratic analytical solution, rendering second-order polynomials a natural fit for this particular example, higher-order approximations, in general, quickly become infeasible due to the curse of dimensionality. For instance, a cubic approximation already requires millions of basis functions. A natural alternative could be to use lower-order approximations like linear regression. However, if one uses linear bases to solve the diffusion equation, they cannot capture temporal dynamics because linear basis functions are harmonic (the Laplacian is zero). By contrast, our approach provides mildly non-linear bases that have non-zero Laplacians, facilitating "almost linear" approximation at each point in time.

The fact that Frozen-PINN-elm with a single hidden layer yields a significantly accurate and faster approximation compared to PINNs with multiple hidden layers trained with classical back-propagation reveals an interesting observation that one does not necessarily benefit from using deeper neural networks. While stochastic and iterative training methods might eventually identify suitable parameters, the highly non-linear, non-convex loss landscape makes such optimization particularly challenging.

Due to the smoothness and lack of steep gradients in the solution of the PDE, Frozen-PINN-elm is more suitable for approximating the solution of the chosen PDE and is one to three orders of magnitude more accurate than vanilla Frozen-PINN-swim, as one would expect (see Section 2.2).

**Ablation studies:** The ablation study with respect to the network width for Frozen-PINN-elm and Frozen-PINN-swim is already presented in Figure 7, where we observe a rapid exponential decay of error with respect to increasing width of the network (even exponential convergence for the high-dimensional diffusion equation in 3 and 5 dimensions).

The hyperparameters for all neural PDE solvers considered in this work for the 10-d heat equation and the 100-d heat equation are presented in Table 29 and Table 30, respectively. The results for up

to 100-dimensional diffusion equations are summarized in Table 31. Please refer to He et al. (2023) for details on hyperparameters for PINNs for the 100-dimensional heat equation.

The results of the ablation study for the SVD layer with the high-dimensional diffusion equations for different dimensions are presented in Tables 32 and 33. We observe that for Frozen-PINN-elm, the SVD layer results in substantial speed-ups for 3, 5, and 7 dimensional heat equations - by factors of 52, 77, and 21, respectively. We observe that the compression ratios achieved with the SVD layer are also substantial 22.8, 5, and 1.2, for dimensions 3, 5, and 7, respectively. For the 10-dimensional diffusion equation, to cover the high-dimensional space, we observe a (relatively lower compared to other dimensions) compression ratio of 1.4, as more basis functions are required to represent functions in high dimensions accurately. Thus, the time required with the SVD layer is around 6 percent less than the time required without the SVD layer. In all the cases, the loss is always in the same order as the one without the SVD layer.

Note that in all cases, the extra cost of computing the SVD easily pays off by substantially saving time in the ODE solver for Frozen-PINN-elm. This is because of the improved conditioning of the feature matrix and the reduction in the size of the ODE system to be solved. With Frozen-PINN-swim, the observations are similar but with lower compression ratios and speed-ups. But, for this problem, Frozen-PINN-swim is not the preferred method, as the underlying solution is smooth, has low-frequency spatial variations, and does not have steep gradients anywhere in the domain. Thus, SWIM basis functions are not optimal in the vanilla setting. See Appendix B.2.1 for details on this.

**Comparison of results:** We demonstrate that Frozen-PINN-elm accurately solves the 10-dimensional and 100-dimensional heat equation by visualizing the time evolution of the solution at some sampled points in space in Figure 29 and Figure 30 in certain dimensions.

Table 29: Summary of hyperparameters for the 10-dimensional diffusion equation.

|  | Parameter | Value |
|---|---|---|
| Frozen-PINN-swim | Number of hidden layers | 2 (nonlinear and SVD layer) |
|  | Hidden layer width | 4000 |
|  | Activation | `tanh` |
|  | $L^2$-regularization | $10^{-10}$ |
|  | SVD cutoff | $10^{-10}$ |
|  | ODE solver tolerance | $10^{-6}$ |
|  | Loss | mean-squared error |
|  | boundary condition strategy | augmented ODE |
| Frozen-PINN-elm (low-precision) | Number of hidden layers | 2 (nonlinear and SVD layer) |
|  | Hidden layer width | 400 |
|  | Activation | `tanh` |
|  | $L^2$-regularization | $10^{-5}$ |
|  | SVD cutoff | $10^{-5}$ |
|  | ODE solver tolerance | $10^{-4}$ |
|  | parameter range $[-r_m, r_m]$ | $r_m = 0.05$ |
|  | Loss | mean-squared error |
|  | boundary condition strategy | augmented ODE |
| Frozen-PINN-elm (high-precision) | Number of hidden layers | 2 (nonlinear and SVD layer) |
|  | Hidden layer width | 4000 |
|  | Activation | `tanh` |
|  | $L^2$-regularization | $10^{-10}$ |
|  | SVD cutoff | $10^{-10}$ |
|  | ODE solver tolerance | $10^{-6}$ |
|  | parameter range $[-r_m, r_m]$ | $r_m = 0.05$ |
|  | Loss | mean-squared error |
|  | boundary condition strategy | augmented ODE |
| PINN | Number of hidden layers | 4 |
|  | Layer width | 20 |
|  | Activation | `tanh` |
|  | Optimizer | LBFGS (ADAM) |
|  | Epochs | 1000 (5000) |
|  | Loss | mean-squared error |
|  | Learning rate | 0.1 |
|  | Batch size | 4000 |
|  | Parameter initialization | Xavier (Glorot & Bengio, 2010) |
|  | Loss weights, $\lambda_1, \lambda_2$ | 1, 1 |
|  | # Interior points | 16000 |
|  | # Initial and boundary points | 4000 |

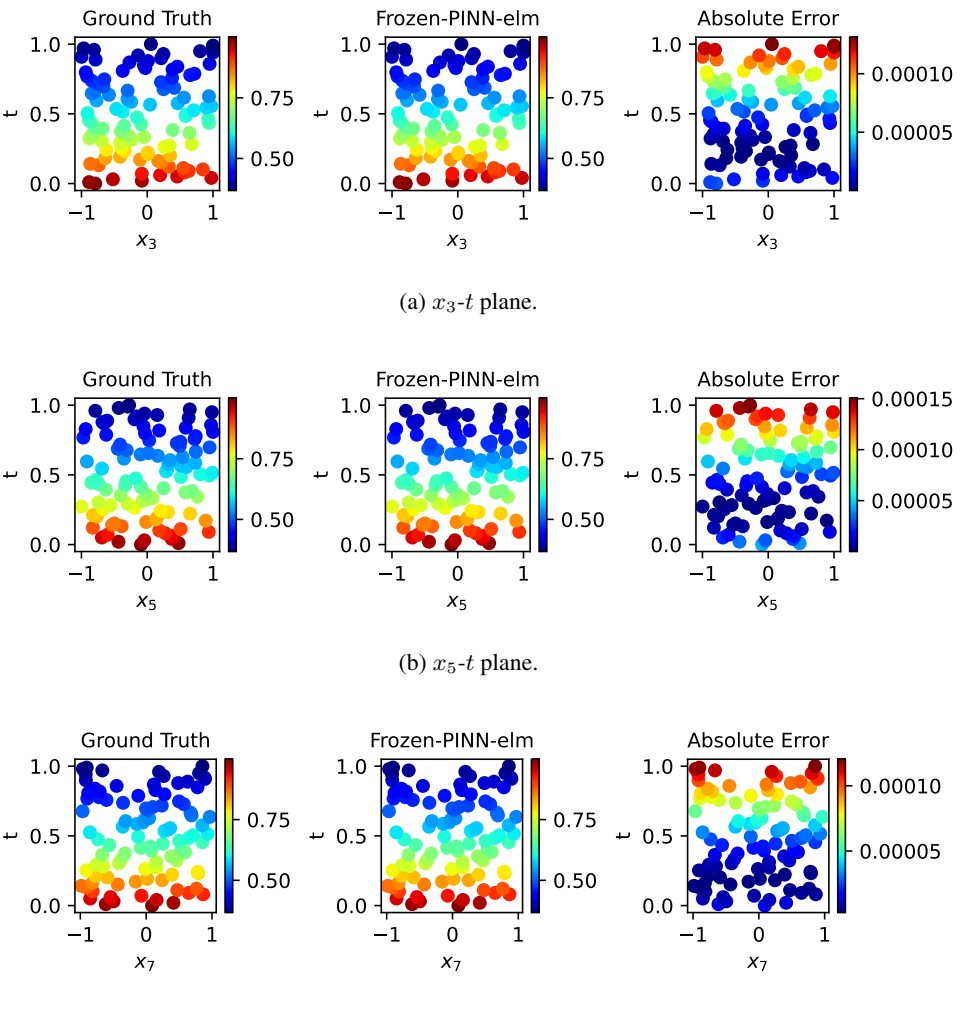

(a) $x_3$-$t$ plane.

(b) $x_5$-$t$ plane.

(c) $x_7$-$t$ plane.

Figure 29: 10-dimensional diffusion equation: Ground truth, Frozen-PINN-elm solution, and point-wise absolute error at various planes at different time points. The rest of the spatial coordinates are set to the center of the spatial-temporal domain.

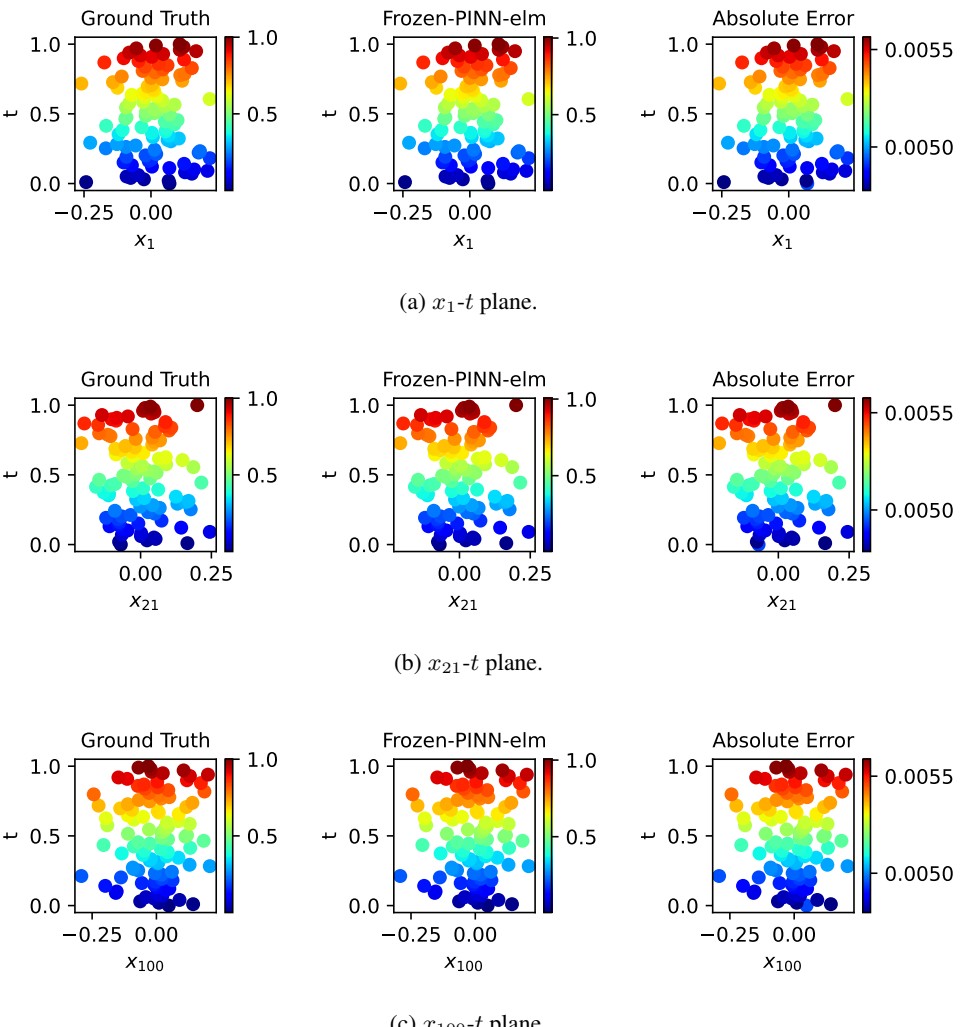

(a) $x_1$-$t$ plane.

(b) $x_{21}$-$t$ plane.

(c) $x_{100}$-$t$ plane.

Figure 30: 100-dimensional diffusion equation: Ground truth, Frozen-PINN-elm solution, and point-wise absolute error at various planes at different time points. The rest of the spatial coordinates are set to the center of the spatial-temporal domain.

Table 30: Summary of hyper-parameters for the 100-dimensional diffusion equation.

| | Parameter | Value |
|---|---|---|
| Frozen-PINN-swim | Number of hidden layers | 2 (nonlinear and SVD layer) |
| | Hidden layer width | 200 |
| | Activation | `tanh` |
| | $L^2$-regularization | $10^{-8}$ |
| | SVD cutoff | $10^{-8}$ |
| | ODE solver tolerance | $10^{-4}$ |
| | Loss | mean-squared error |
| | boundary condition strategy | augmented ODE |
| Frozen-PINN-elm | Number of hidden layers | 2 (nonlinear and SVD layer) |
| | Hidden layer width | 125 |
| | Activation | `tanh` |
| | $L^2$-regularization | $10^{-4}$ |
| | SVD cutoff | $10^{-4}$ |
| | ODE solver tolerance | $10^{-2}$ |
| | parameter range $[-r_m, r_m]$ | $r_m = 0.05$ |
| | Loss | mean-squared error |
| | boundary condition strategy | augmented ODE |

Table 31: Summary of results for high-dimensional diffusion equation. We denote the Frozen-PINN-elm results in the low-precision and high-precision regimes with Frozen-PINN-elm-fast and Frozen-PINN-elm-accurate, respectively.

| Dimension | Method | Time (s) | RMSE | Relative $L^2$ error |
|---|---|---|---|---|
| 3-d | PINN (LBFGS) | 102.32 | 2.84e-4 $\pm$ 3.73e-5 | 4.54e-4 $\pm$ 5.97e-5 |
| | Frozen-PINN-swim (our) | 95.73 | 2.18e-6 $\pm$ 1.93e-6 | 5.37e-6 $\pm$ 4.27e-7 |
| | **Frozen-PINN-elm-fast (our)** | **0.9** | 2.42e-6 $\pm$ 1.37e-6 | 3.90e-6 $\pm$ 2.98e-6 |
| | **Frozen-PINN-elm-accurate (our)** | 60.98 | **3.48e-8 $\pm$ 2.17e-6** | **6.49e-8 $\pm$ 4.31e-8** |
| 5-d | PINN (LBFGS) | 133.95 | 2.91e-4 $\pm$ 5.34e-5 | 4.52e-4 $\pm$ 8.30e-5 |
| | Frozen-PINN-swim (our) | 129.65 | 1.03e-4 $\pm$ 5.94e-5 | 2.39e-4 $\pm$ 8.69e-5 |
| | **Frozen-PINN-elm-fast (our)** | **1.2** | 1.25e-4 $\pm$ 2.42e-5 | 3.74e-4 $\pm$ 5.37e-5 |
| | **Frozen-PINN-elm-accurate (our)** | 102.95 | **4.71e-7 $\pm$ 3.56e-7** | **7.5e-7 $\pm$ 3.92e-7** |
| 7-d | PINN (LBFGS) | 163.89 | 3.05e-4 $\pm$ 2.94e-5 | 4.69e-4 $\pm$ 4.51e-5 |
| | Frozen-PINN-swim (our) | 198.20 | 3.96e-4 $\pm$ 1.03e-4 | 7.8e-4 $\pm$ 2.50e-4 |
| | **Frozen-PINN-elm-fast (our)** | **5.95** | 1.05e-5 $\pm$ 8.76e-6 | 2.21e-5 $\pm$ 1.01e-5 |
| | **Frozen-PINN-elm-accurate (our)** | 176.95 | **1.19e-6 $\pm$ 2.93e-7** | **2.54e-6 $\pm$ 5.10e-7** |
| 10-d | PINN (LBFGS) | 189.67 | 3.98e-4 $\pm$ 6.59e-5 | 6.06e-4 $\pm$ 1.00e-4 |
| | Frozen-PINN-swim (our) | 61.07 | 1.01e-3 $\pm$ 3.09e-4 | 2.31e-3 $\pm$ 1.03e-3 |
| | **Frozen-PINN-elm-fast (our)** | **2.07** | 2.89e-4 $\pm$ 5.91e-5 | 4.46e-4 $\pm$ 9.61e-5 |
| | **Frozen-PINN-elm-accurate (our)** | 182.91 | **1.04e-5 $\pm$ 3.32e-6** | **2.28e-5 $\pm$ 5.91e-6** |
| 100-d | Vanilla PINN ((He et al., 2023)) | 141 | - | 6.00e-3 |
| | PINN ((He et al., 2023)) | 49.8 | - | 6.30e-3 |
| | Frozen-PINN-swim (our) | 68.39 | 1.00e-3 $\pm$ 1.75e-5 | 1.71e-3 $\pm$ 3.01e-5 |
| | **Frozen-PINN-elm (our)** | **5.24** | **2.40e-4 $\pm$ 9.92e-6** | **4.12e-4 $\pm$ 1.70e-5** |

Table 32: High-dimensional diffusion equation: Ablation Study for the SVD layer with Frozen-PINN-swim.

| Dimension | Quantity | With SVD layer | Without SVD layer | Ratio |
|---|---|---|---|---|
| 3-d | Width | 1391 | 4000 | Compression $\approx 2.9$x |
| | Time (s) | 95.73 | 388.12 | Speed-up $\approx 4$x |
| | Rel. $L_2$ error | 5.29e-6 | 4.77e-6 | - |
| 5-d | Width | 1437 | 4000 | Compression $\approx 2.8$x |
| | Time (s) | 129.65 | 199.92 | Speed-up $\approx 1.5$x |
| | Rel. $L_2$ error | 2.39e-4 | 2.18e-4 | - |
| 7-d | Width | 3114 | 4000 | Compression $\approx 1.3$x |
| | Time (s) | 120.32 | 198.31 | Speed-up $\approx 1.6$x |
| | Rel. $L_2$ error | 7.83e-4 | 7.83e-4 | - |
| 10-d | Width | 3100 | 4000 | Compression $\approx 1.3$x |
| | Time (s) | 121.93 | 111.8 | Speed-up $\approx 0.91$x |
| | Rel. $L_2$ error | 2.30e-3 | 2.30e-3 | - |
| 100-d | Width | 200 | 200 | Compression $\approx 1$x |
| | Time (s) | 5.24 | 5.13 | Speed-up $\approx 0.97$x |
| | Rel. $L_2$ error | 3.82e-3 | 3.82e-3 | - |

Table 33: High-dimensional diffusion equation: Ablation Study for the SVD layer with Frozen-PINN-elm.

| Dimension | Quantity | With SVD layer | Without SVD layer | Ratio |
|---|---|---|---|---|
| 3-d | Width | 175 | 4000 | Compression $\approx 22.8$x |
| | Time (s) | 60.98 | 7087.38 | Speed-up $\approx 52$x |
| | Rel. $L_2$ error | 6.49e-8 | 1.02e-6 | - |
| 5-d | Width | 794 | 4000 | Compression $\approx 5$x |
| | Time (s) | 89.27 | 6873.8 | **Speed-up $\approx$ 77x** |
| | Rel. $L_2$ error | 7.30e-7 | 2.19e-6 | - |
| 7-d | Width | 3336 | 4000 | Compression $\approx 1.2$x |
| | Time (s) | 176.95 | 3770.09 | Speed-up $\approx 21$x |
| | Rel. $L_2$ error | 2.54e-6 | 4.06e-6 | - |
| 10-d | Width | 2856 | 4000 | Compression $\approx 1.4$x |
| | Time (s) | 119 | 127 | Speed-up $\approx 1.06$x |
| | Rel. $L_2$ error | 5.57e-5 | 4.36e-5 | - |
| 100-d | Width | 552 | 600 | Compression $\approx 1.3$x |
| | Time (s) | 68.39 | 71.38 | Speed-up $\approx 0.96$x |
| | Rel. $L_2$ error | 1.71e-3 | 1.71e-3 | - |

