# OpenReview forum: "Fast training of accurate physics-informed neural networks without gradient descent"
_ICLR.cc/2026/Conference — ICLR 2026 Oral_

### Official Review · Reviewer_Ukoa · 2025-10-29

**Soundness:** 2
**Presentation:** 3
**Contribution:** 2
**Rating:** 4
**Confidence:** 4

**Summary:**

This paper presents Frozen-PINNs, a gradient-free framework for solving time-dependent partial differential equations. The method separates spatial and temporal components of the solution: spatial bases are randomly sampled and fixed, while the temporal coefficients evolve through a least-squares initialization and an adaptive ODE solver, ensuring temporal causality. By avoiding gradient descent, the approach simplifies optimization and achieves remarkable computational speedups and accuracy across various PDE benchmarks, including advection, diffusion, Burgers’, and high-dimensional heat equations.

**Strengths:**

1. The paper is clearly written and well organized, with informative figures and tables that effectively convey key ideas.
2. The method delivers strong empirical performance, achieving large speedups and accuracy gains across diverse PDE benchmarks.
3. The experiments are thorough and well documented, with detailed ablations and transparent reporting that make reproduction straightforward.

**Weaknesses:**

1. The proposed approach essentially reformulates the PDE into an ODE and integrates it with a standard solver, while the neural network merely provides a fixed spatial basis. As a result, the method’s “learning” component is limited, and the main computational work is performed by the numerical ODE solver. This makes the conceptual novelty questionable and leaves unclear why such a neural formulation is preferable to directly applying classical ODE or PDE solvers.

2. The evaluation focuses mainly on relatively simple or smooth problems. To demonstrate robustness and generality, the method should be tested on more strongly nonlinear or coupled systems—such as the Navier–Stokes equations (e.g., lid-driven cavity flow)—where the dynamics are more complex and multi-scale behavior is prominent.

3. From the hyperparameter tables (Tables 4 and 5, etc.), the proposed model appears considerably larger than the PINN and causal PINN baselines. Moreover, the baseline settings deviate from common practice—the PINN models are quite small, trained for fewer than 10,000 iterations, and in some cases use an unusually large initial learning rate of 0.1 with Adam. While this setup shortens their reported training time, it likely underrepresents their attainable accuracy, making the comparison of efficiency and performance less conclusive.

**Questions:**

1. If I understand correctly, the authors use neural bases to represent the spatial domain and then integrate the resulting ODE system using a numerical ODE solver. If that is the case, why not simply apply a conventional ODE solver directly? The authors may argue that their approach improves upon standard PINNs, which I acknowledge. However, this direction seems to diverge from the central objective of PINN research in machine learning—to understand how well neural networks can approximate physical dynamics without relying on traditional numerical solvers. Such investigations are valuable because they provide insights transferable to other frameworks such as physics-informed neural ODEs, physics-informed neural operators, and physics-informed diffusion models. How do the authors justify this shift in focus? Do Frozen-PINNs have the potential to achieve higher accuracy than classical ODE solvers? What additional benefits or conceptual insights beyond accuracy does this approach provide?

2. What numerical precision is used during training—float32 or float64? Please clarify, as this can significantly affect both runtime and accuracy.

3. Is the method scalable in practice? It seems to rely on full-batch computation. How many collocation points are used for the 100-dimensional diffusion equation, and how does the cost scale with dimensionality?


4. How were the SVD truncation threshold and L2 regularization chosen? Was any ablation study conducted to assess their sensitivity and impact on performance?

---

> ### Author Response · Authors · 2025-11-21
>
> We appreciate your valuable time and thorough feedback. We thank you for highlighting many strengths of our work, including its clear and concise presentation style, detailed ablation studies, strong empirical performance, and transparent reporting. We have carefully implemented your suggestions through new experiments and attempted to address your concerns and questions to the best of our ability. Our point-by-point responses to the identified weaknesses and questions are provided below.
>
> > **W1.1** The proposed approach essentially reformulates the PDE into an ODE and integrates it with a standard solver, while the neural network merely provides a fixed spatial basis. As a result, the method’s “learning” component is limited, and the main computational work is performed by the numerical ODE solver.
>
> \--> We believe that, although the network parameters are not learned iteratively via gradient-based approaches, the *learning component* is not necessarily limited. One can utilize sampling algorithms in useful ways that **offer advantages compared to implicitly learning basis functions via iterative gradient descent-based optimization**, as demonstrated in the following two examples.
>
> 1. **Burgers' equation (Section 3.4):** our method enables us to modify the neural basis over simulated time, based on the required complexity to efficiently capture shocks by sampling step basis functions near the shock, resulting in orders of magnitude faster training for a comparable accuracy.
> 2. **Non-linear diffusion equation (Section 3.5):** explicitly embedding directional information to orient Frozen-PINN-swim basis functions along the gradient of the initial condition proves to be much more effective (faster and more accurate) than implicitly learning the basis functions.
>
> > **W1.2** This makes the conceptual novelty questionable
>
> \--> The concepts of space-time separation and ODE solvers are, of course, **classical techniques in scientific computing**, as correctly pointed out by the reviewer. We argue that the main contribution here lies in how these tools **are** **repurposed to construct parameters of the neural network**, which is used as an ansatz for the PDE solution.
>
> Thus, the novelty of our work lies in the **knowledge transfer from random feature methods and scientific computing** **to alleviate some of the long-standing and key optimization (training) issues** that have severely restricted the performance of PINNs. To be precise, in our view, some of the key components that add to the conceptual novelty of our work are:
>
> 1. **Fundamentally simplifying the PINN optimization problem via an elegant reformulation** using the Frozen-PINN ansatz via space-time separation (as opposed to the focus of the PINN research community on developing advanced optimizers (\[a,b\]) that try to solve an inherently difficult optimization problem).
> 2. **Enforcing temporal causality in PINNs by construction** (which was mostly incorporated as soft constraints before).
> 3. **Use of SVD layer** (to reduce dimensionality of the ODE solver and accelerate training).
> 4. **New strategies for enforcing boundary conditions** (boundary-compliant layer, augmented ODE)
> 5. **Embedding directional and temporal information** (solutions from previous time-steps) **to construct "efficient" neural basis functions**, as shown in **Section 3.4** when solving the Burgers' equation and **Section 3.5** in solving the non-linear reaction-diffusion equation (directly extending the work on random feature methods by Bolager et al.)
>
>
>
> > **W1.3** leaves unclear why such a neural formulation is preferable to directly applying classical ODE or PDE solvers.
>
> \--> Please refer to our answer to your Question 1.

---

> ### Author Response · Authors · 2025-11-21
>
> > **W2.** The evaluation focuses mainly on relatively simple or smooth problems. To demonstrate robustness and generality, the method should be tested on more strongly nonlinear or coupled systems—such as the Navier–Stokes equations (e.g., lid-driven cavity flow)—where the dynamics are more complex and multi-scale behavior is prominent.
>
> \--> We thank the reviewer for this comment. Our point-wise response is given below.
>
> - **Need for Domain Decomposition for Extreme Multiscale Problems:** We agree that testing on highly nonlinear and multiscale systems (e.g., the Navier–Stokes equations) would further validate robustness. However, for such problems, **combining Frozen-PINNs with domain-decomposition approaches** is essential, especially given the strong performance of finite-basis PINNs and multi-level FB-PINNs \[c, d\].
> - **Ongoing Work with Multilevel Schwarz–Inspired Methods:** We are already pursuing this direction using **multilevel Schwarz–inspired domain decomposition** \[d\]. Preliminary results on *static* PDEs show that Frozen-PINNs can match the accuracy of multi-level FB-PINNs **while training significantly faster**. These results are not included here, as the paper focuses on time-dependent PDEs.
> - **Future Extensions:** Extending Frozen-PINNs to **time-dependent 2D/3D fluid flows** **is a natural next step**, aligning with recent progress in \[e\], but **requires effort beyond the scope of this rebuttal.** The same holds for coupled PDE systems, which represent another promising and natural extension.
> - **Existing Nonlinear Benchmark:** Within the present scope, our experiments already include the **viscous Burgers equation**, which **features strong nonlinearity, steep gradients, and shock-like structures**, providing a meaningful stress test within the intended scope.
> - ***New Benchmark on Wave Equation (Hao et al. \[f\]):*** Motivated by the reviewer’s request, we have **added a new benchmark from Hao et al. \[f\]** for the wave equation (***Sec C.3***). This benchmark contains slightly complex and multi-scale solution behavior. We demonstrate that CPU-trained Frozen-PINNs achieve **$625$ to $5500$ times faster training** than GPU-trained competing PINN variants (vanilla PINN, PINN-NTK, FB-PINN, gPINN) while simultaneously being **four to five** **orders of magnitude higher in accuracy** on the wave equation (***See Table 1 and Figure 18***).
> - ***Additional New Benchmark on Multi-Scale Wave Equation*:** To further stress-test the robustness and generality of Frozen-PINNs, we introduce a **more challenging multi-scale wave-equation benchmark** (see ***Sec. C.3, Fig. 19***). Frozen-PINNs solve this problem **extremely quickly and with high precision**, reinforcing their potential for PDEs with complex, multiscale dynamics.
> - **Additional New Benchmark: Kuramoto-Sivashinsky (KS) Equation:** We tackle the **highly nonlinear KS equation exhibiting spatiotemporal chaos and significant spatial complexity**, and show that Frozen-PINNs capture the characteristic chaotic pattern over a long-time horizon (See **Fig. 6**).

---

> ### Author Response · Authors · 2025-11-21
>
> > **W3.** From the hyperparameter tables (Tables 4 and 5, etc.), the proposed model appears considerably larger than the PINN and causal PINN baselines. Moreover, the baseline settings deviate from common practice—the PINN models are quite small, trained for fewer than 10,000 iterations, and in some cases use an unusually large initial learning rate of 0.1 with Adam. While this setup shortens their reported training time, it likely underrepresents their attainable accuracy, making the comparison of efficiency and performance less conclusive.
>
> \--> We appreciate the reviewer's sharp observation. The reviewer's comment prompted us to conduct a more comprehensive evaluation of the PINN baselines and validate the results known in the PINN literature. Please find our point-wise answer below.
>
> - ***New ablations for deeper networks added (See Table 6, Table 7)***: To further analyze the performance of PINN on deeper networks under different optimization scenarios, additional experiments have been conducted for the baseline PINN. The deeper networks consist of 10 and 20 hidden layers, where each hidden layer has 30 neurons. The experiments are run for 20000 epochs using Adam and L-BFGS optimizers under multiple learning rates for the advection equation with $\\beta=10$ and $\\beta=40$. ***Tables 6 and 7*** summarize the RMSE, relative $L^2$ errors, and training times for $\\beta=10$ and $\\beta=40$ cases, respectively.
> - **Summary of results:** The **results are consistent with the known literature of PINNs \[g,h\]**. First, deeper PINNs do not directly lead to better performance. Increasing the depth from 10 to 20 layers often degrades accuracy for both optimizers, reflecting the optimization difficulty of fully connected PINNs as the models become deeper. This behavior has also been discussed previously in the literature, for instance, in \[g\], where the **authors show that PINN performance degrades when larger and deeper neural network architectures** are employed. Second, **the $\\beta=40$ case is known to be challenging for PINNs \[h\] due to the presence of high-frequency features**, and the results presented in Table 7 demonstrate failures across various depths, learning rates, and optimizers. **The results show that even with larger networks, longer training, and different optimizers, standard PINNs face challenges in achieving high accuracy, especially for $\\beta=40$, and require more computational time**. Thus, the previous comparison favored other PINN-based methods by shortening the training time, as correctly pointed out by the reviewer, over Frozen-PINNs, since we were unable to match the high accuracy of Frozen-PINNs for this benchmark anyway.
>
> ***Thanks to the reviewer's comments, this discussion is now supplemented in the revised manuscript's Appendix C.1 and Tables 6 and 7.***
>
> > **Q1.** If I understand correctly, the authors use neural bases to represent the spatial domain and then integrate the resulting ODE system using a numerical ODE solver. If that is the case, why not simply apply a conventional ODE solver directly? The authors may argue that their approach improves upon standard PINNs, which I acknowledge. However, this direction seems to diverge from the central objective of PINN research in machine learning—to understand how well neural networks can approximate physical dynamics without relying on traditional numerical solvers. Such investigations are valuable because they provide insights transferable to other frameworks such as physics-informed neural ODEs, physics-informed neural operators, and physics-informed diffusion models. How do the authors justify this shift in focus? Do Frozen-PINNs have the potential to achieve higher accuracy than classical ODE solvers? What additional benefits or conceptual insights beyond accuracy does this approach provide?
>
> \--> We thank the reviewer for these important questions regarding the objectives of PINN research, the shift in focus of our training algorithm, comparison to classical solvers, additional benefits of Frozen-PINNs, and the broader conceptual insights of our approach. Below, we provide detailed, point-by-point responses to all comments.
>
>  **(continued in the next comment)...**

---

> ### Author Response · Authors · 2025-11-22
>
> > **Q1.1** However, this direction seems to diverge from the central objective of PINN research in machine learning—to understand how well neural networks can approximate physical dynamics without relying on traditional numerical solvers.
>
> \--> While we appreciate the reviewer’s perspective, we would like to offer an alternative view on the reviewer’s point. We argue that the **core idea of PINNs is to use neural networks as an ansatz for the solution**, **rather than**, for example, **classical Fourier bases, radial bases, or mesh-based finite elements**. The training method for PINNs includes the **PDE in the loss**, which is **very similar to the traditional methods** already (e.g., a Galerkin approach that applies the PDE operator to the basis). What we address in our work is how the ansatz of PINNs (neural networks) can be trained differently, to improve both training speed and accuracy. **Our approach still includes the PDE in the loss, and we also end up with a trained neural network that represents the PDE solution.**
>
> > **Q1.3** Do Frozen-PINNs have the potential to achieve higher accuracy than classical ODE solvers?
>
> We thank the reviewer for this question. We answer it point-wise below.
>
> - On the **Burgers’ equation**, we demonstrate (**see Figure 22, Figure 23, and Appendix C.4.1 for details**) that SWIM basis functions employed in Frozen-PINNs accurately resolve shocks, **outperforming traditional bases (Fourier, Chebychev, ELM) in both convergence rate and accuracy**. Moreover, for the same accuracy, SWIM requires far fewer basis functions, which is a significant benefit. Thus, for certain problems, SWIM bases are quite useful and better than traditional analytical bases used in classical PDE solvers.
> - For low-dimensional PDEs (advection, Euler-Bernoulli, and Burgers' equation), we actually compare our method to the classical method - Iso Geometric Analysis (IGA), and the **performance is similar in low dimensions (See Table 1)**. Importantly, **for high-dimensional PDEs, mesh-based methods (IGA) and spectral methods suffer from the curse of dimensionality** (number of mesh points/basis functions blow up with dimension), whereas **neural networks can solve certain high-dimensional PDEs with smooth solutions quite efficiently**, as demonstrated with two benchmarks in **Section 3.8** for up to a 100-dimensional heat equation.
> - We view **Frozen-PINNs as an important computational method complementing the classical PDE solvers** based on mesh-based, spectral, and particle-based methods. We believe that each method has its pros and cons, and **no method is universally better than the others.**
>
> > **Q1.4** What additional benefits or conceptual insights beyond accuracy does this approach provide?
>
> \--> We thank the reviewer for this question. The key conceptual insights on Frozen-PINNs beyond accuracy are as follows:
>
> - **A simple reformulation of the PINN optimization problem via the Frozen-PINN neural ansatz addresses some of the major optimization issues for classical PINNs** that create bottlenecks of accuracy and training time.
> - **Explicitly using solution-aware/direction-aware basis functions can be orders of magnitude faster** (for Burgers and non-linear reaction diffusion benchmarks) **and more accurate** (for non-linear reaction diffusion benchmark) than implicitly learning the basis functions either with steep gradients (for Burgers') or appropriately oriented (for non-linear reaction diffusion).
> - Conceptually, solution-driven neural bases **behave like adaptive meshing** in finite elements, **without requiring explicit meshes**, offering significant computational and representational advantages.

---

> ### Author Response · Authors · 2025-11-22
>
> > **Q1.2** Such investigations are valuable because they provide insights transferable to other frameworks such as physics-informed neural ODEs, physics-informed neural operators, and physics-informed diffusion models. How do the authors justify this shift in focus?
>
> We thank the reviewer for this question, and we answer it with the following two points:
>
> - **Concerning the shift in focus:**
>   - One of the primary focus areas of PINN research is improving gradient-descent-based training and optimizers. Yet, resolving training difficulties in PINNs remains one of the core challenges in PINN research. While the prior attempts at designing efficient optimizers are valuable, they address the symptoms of an inherently difficult optimization problem. **Our work intentionally shifts focus toward reformulating this problem to provide solutions for mitigating two root causes (instead of symptoms)**, limiting the performance of PINNs in terms of accuracy and training time: (1) Inherent challenges posed by the PINN optimization problem, and (2) non-causal treatment of time as an extra spatial dimension. Thus, **we believe that, despite the change in focus, the goal of our work remains central to PINN research.**
>   - The value of this **change in focus is further justified by the substantial benefits of our approach**, which are **supported by empirical evidence on nine PDE benchmarks**. To be precise, some of the key benefits are: (1) **substantial speed-ups**, (2) **substantial accuracy gains**, (3) **respecting temporal causality by construction**, and (4) **less reliance on specialized hardware** like GPUs and iterative gradient-descent-based methods. As a consequence, we believe that **these conceptual and empirical advances**, along with their ramifications, **are** **crucial for the PINN community to build upon** in advancing and understanding fast and accurate neural PDE solvers.
> - **Concerning the point on insights transferable to other frameworks:**
>   - **Random feature methods** (such as SWIM, ELM), which are central to Frozen-PINNs, **have already been successfully used in synergy with other frameworks to accelerate their training**. Bolager et al. \[i\] demonstrate how to use random sampling **to speed up training of neural operator frameworks like POD-DeepONet \[m\] and Fourier Neural Operator \[n\]**. Rahma et al. \[j\], \[k\] show how random features can be used **to accelerate the training of Hamiltonian neural networks and Hamiltonian graph neural networks**.
>   - Similarly, the idea of combining data-driven bases with classical ODE solvers has been explored in the context of operator learning. For instance, **Meuris et al.** **\[l\]** propose using the spatial basis functions from a trained DeepONet and employing it in a traditional spectral method with classical ODE solvers, **synergistically integrating the data-driven bases and classical ODE solvers, in the same spirit as the core idea of Frozen-PINNs**. Thus, **the key ideas of Frozen PINNs have already shown significant promise and can certainly be transferred to many other useful frameworks.**
>   - Physics-informed machine learning frameworks are increasingly used within broader scientific machine learning pipelines. For example, (causal) **physics-informed neural networks are often combined with neural ordinary differential equations (neural ODEs) in a two-stage procedure** to extrapolate the behavior of physical systems beyond the original computational domain. In approaches of this form (e.g., \[o\], \[p\]), the first stage uses a PINN to generate solution data, and the second stage trains a neural ODE on that data to produce an extrapolative model. **Replacing standard PINNs with the proposed Frozen PINNs in the first stage of such pipelines would both accelerate training and improve predictive accuracy.** As a result, the entire framework would benefit from faster and more reliable predictions, including in regimes outside the training domain.
>   - Additionally, we are **actively exploring applying these ideas within the framework of FB-PINNs \[c\] and multi-level FB-PINNs \[d\]** to solve PDEs with multi-scale solutions, and we have already obtained **promising results for static PDEs** (which are excluded from this work, as the focus is on space-time separation).
>
>
> > **Q2.** What numerical precision is used during training—float32 or float64? Please clarify, as this can significantly affect both runtime and accuracy.
>
> \--> We thank the reviewer for this question. We used float64 during training for all approaches, and thanks to your remark, we have now mentioned this in the updated manuscript as well.

---

> ### Author Response · Authors · 2025-11-22
>
> > **Q3.1** How many collocation points are used for the 100-dimensional diffusion equation
>
> \--> For the 100-dimensional diffusion equation, we used 2000 training collocation points (1000 in the interior and 1000 on the boundary) and 8000 test collocation points (8000 in the interior and 2000 on the boundary). See Section C.8 for details.
>
> > **Q3.2** Is the method scalable in practice? It seems to rely on full-batch computation. and how does the cost scale with dimensionality?
>
> \--> We thank the reviewer for this question. The amount of data needed may increase quickly as the dimension increases. The primary concern from a computational standpoint is how many data points are needed; more importantly, how does increasing the number of data points significantly impact training time? For our method, this boils down to the linear solver complexity, which is cubic (in computation time) with respect to the number of neurons. We know from several results that the number of neurons needed for many functions is much smaller than $N$. In the literature for random features, it is supported that the number of neurons needed is on the scale of $N^{1/2}log N^{1/2} << N$ \[q\]. This means that **our method scales cubically with the number of neurons** and **linearly with the number of data points,** and that **the number of neurons required is much less than the number of data points.** Hence, there are good reasons to believe that **the scaling is relatively good when the dimension increases, something we observed in Figure 7.** This is not to say our method will not suffer from the curse of dimensionality at all, but we can expect it to do comparatively well.
>
> The ODE system to be solved has the dimension equal to the number of neurons, as $C(t)$ at time $t$ is a vector of size equal to the number of neurons. Based on the discussion above, this would also imply that **we can solve in a much lower-dimensional space than the number of data points.** The exact scaling properties would, of course, depend on the particular solver, the time horizon, and the specified accuracy.
>
> > **Q4.1** How were the SVD truncation threshold and L2 regularization chosen?
>
> \--> We thank the reviewer for this question. We performed a **hyperparameter sweep** to **select the optimal SVD truncation threshold**. **We always set the L2 regularization constant** (***rcond*** parameter passed to NumPy’s least-squares solver **numpy.linalg.lstsq)** for the initial least squares solve **to the same value as the SVD truncation threshold** because it also represents the cut-off ratio for the SVD of the feature matrix for the initial least squares solve. This is because it does not make sense to solve the least squares with extremely high or low precision when the data has already been passed through the SVD layer using the SVD truncation threshold, to maintain a similar level of truncation/regularization as the SVD layer.
>
> > **Q4.2** Was any ablation study conducted to assess their sensitivity and impact on performance?
>
> \--> We thank the reviewer for this question. Motivated by the reviewer's question, **we have now also added ablation studies for the SVD truncation threshold (See Figure 10) and the L2 regularization constant (see Figure 9) to assess their sensitivity and impact on performance.** For details, **please refer to the new experiments in Section B.2.1** concerning the extended discussion on Frozen-PINNs, with the bullet point titled **\`\`On the choice of SVD truncation threshold''**. We summarize the key findings below:
>
> - **SVD truncation threshold ablation:** Figure 10 illustrates the well-known **trade-off between accuracy and speed, as determined by the SVD truncation threshold.** Retaining fewer singular values reduces the dimensionality and stiffness of the last-layer ODE, yielding faster solutions with similar or slightly lower accuracy. Retaining more singular values increases dimensionality and stiffness, which slows the solver but improves accuracy. Importantly, **the performance is robust for SVD truncation thresholds ($\epsilon\_{SVD} < 10^{-10}$) in all cases.**
> - **L2 regularization ablation:** We observe very robust performance when the regularization constant is set equal to the SVD threshold. In this newly added ablation study (see **Figure 9**), the relative error stays the same for the regularization constant in the range $10^{-17} - 10^{-11}$, when the SVD truncation threshold is set at $10^{-14}$. This **empirically validates: (a) setting the SVD truncation threshold and L2 regularization to be the same results in robust performance, and (b) the performance is not too sensitive to values of L2 regularization in a broad range around the SVD truncation threshold.**

---

> ### Author Response · Authors · 2025-11-22
>
> To conclude, we sincerely appreciate your thoughtful review and constructive comments. Your suggestions led directly to several meaningful improvements in the manuscript, including the addition of multiple new experiments and clarifications.
>
> We sincerely hope we satisfactorily answered your questions and improved our work to the standard you expect for the ICLR community. We eagerly look forward to your response to our rebuttal and would be more than happy to answer any further questions as soon as possible.
>
> **References:**
>
> \[a\] Liu, Qiang, Mengyu Chu, and Nils Thuerey. "ConFIG: Towards Conflict-free Training of Physics Informed Neural Networks." The Thirteenth International Conference on Learning Representations.
>
> \[b\] Kiyani, Elham, et al. "Optimizing the optimizer for physics-informed neural networks and Kolmogorov-Arnold networks." Computer Methods in Applied Mechanics and Engineering 446 (2025): 118308.
>
> \[c\] Moseley, Ben, Andrew Markham, and Tarje Nissen-Meyer. "Finite basis physics-informed neural networks (FBPINNs): a scalable domain decomposition approach for solving differential equations." Advances in Computational Mathematics 49.4 (2023): 62.
>
> \[d\] Dolean, Victorita, et al. "Multilevel domain decomposition-based architectures for physics-informed neural networks." Computer Methods in Applied Mechanics and Engineering 429 (2024): 117116.
>
> \[e\] Wang, Sifan, et al. "Simulating three-dimensional turbulence with physics-informed neural networks." *arXiv preprint arXiv:2507.08972* (2025).
>
> \[f\] Zhongkai, Hao, et al. "Pinnacle: A comprehensive benchmark of physics-informed neural networks for solving pdes." *Advances in Neural Information Processing Systems* 37 (2024): 76721-76774.
>
> \[g\] Wang, Sifan, et al. "Piratenets: Physics-informed deep learning with residual adaptive networks." *Journal of Machine Learning Research* 25.402 (2024): 1-51.
>
> \[h\] Krishnapriyan, Aditi, et al. "Characterizing possible failure modes in physics-informed neural networks." *Advances in neural information processing systems* 34 (2021): 26548-26560.
>
> \[i\] Bolager, Erik L., et al. "Sampling weights of deep neural networks." Advances in Neural Information Processing Systems 36 (2023): 63075-63116.
>
> \[j\] Rahma, Atamert, Chinmay Datar, and Felix Dietrich. "Training Hamiltonian neural networks without backpropagation." arXiv preprint arXiv:2411.17511 (2024).
>
> \[k\] Rahma, Atamert, et al. "Rapid training of Hamiltonian graph networks without gradient descent." arXiv preprint arXiv:2506.06558 (2025).
>
> \[l\] Meuris, Brek, Saad Qadeer, and Panos Stinis. "Machine-learning-based spectral methods for partial differential equations." Scientific Reports 13.1 (2023): 1739.
>
> \[m\] Lu, Lu, et al. "A comprehensive and fair comparison of two neural operators (with practical extensions) based on fair data." Computer Methods in Applied Mechanics and Engineering 393 (2022): 114778.
>
> \[n\] Li, Zongyi, et al. "Fourier neural operator for parametric partial differential equations." arXiv preprint arXiv:2010.08895 (2020).
>
> \[o\] Kapoor, Taniya, et al. "Neural oscillators for generalization of physics-informed machine learning." *Proceedings of the AAAI Conference on Artificial Intelligence*. Vol. 38. No. 12. 2024.
>
> \[p\] Kapoor, Taniya, et al. "Neural differential equation-based two-stage approach for generalization of beam dynamics." *IEEE Transactions on Industrial Informatics* (2024).
>
> \[q\] Generalization Properties of Learning with Random Features, Rudi et.al., 2021

---

### Official Review · Reviewer_iLK9 · 2025-10-31

**Soundness:** 4
**Presentation:** 3
**Contribution:** 4
**Rating:** 8
**Confidence:** 3

**Summary:**

The paper proposes Frozen-PINN, a gradient-free Physics-Informed Neural Network that replaces iterative optimization with a space–time separation strategy. Spatial features are randomly sampled and fixed, while time-dependent coefficients are solved using ODE solvers, ensuring temporal causality by design. This approach greatly simplifies training and removes dependence on backpropagation. Across multiple PDE benchmarks, Frozen-PINN achieves much faster convergence and higher accuracy than existing PINNs and even matches mesh-based solvers, establishing a fast, causal, and interpretable alternative to gradient-descent-based PINN training.

**Strengths:**

>S1.

This paper presents a methodology that fundamentally differs from conventional PINN training approaches. Through a space-time separation framework, the proposed method circumvents the multi-objective optimization problem inherent in PINNs and naturally incorporates time causality, thereby avoiding several fundamental challenges that have long limited PINN performance.

>S2.

The paper is well-organized with clear motivation. The proposed approach offers a reasonable alternative framework with solid justification for why it sidesteps the challenges faced by traditional PINNs.

>S3.

Frozen PINN demonstrates substantially faster training times and higher accuracy compared to the conventional PINN approach. The experiments are conducted on benchmark problems known to be challenging for PINNs, including cases with extreme advection speeds, shocks, and high-dimensionality. Across these benchmark, Frozen PINN consistently exhibits superior performance.

**Weaknesses:**

> W1.

Frozen-PINN satisfies the initial condition by solving a least squares problem. While this decouples the initial condition from the PDE and boundary conditions, thereby simplifying the optimization, the least squares solution may not always be perfect. An analysis of the residual initial condition loss and its impact on model performance would be valuable. More broadly, investigating the sources of different loss components and their effects on accuracy would enhance our understanding of Frozen-PINN's behavior.

>W2.

Despite the various challenges in PINN training, increasing the number of collocation points (combined with appropriate sampling strategies and architectures) generally improves performance when sufficient training is applied. In my understanding, Frozen-PINN could also reduce PDE residual loss by investing more computation to solve the ODEs more accurately. However, since the initial condition relies on a predetermined least squares solution, it is unclear whether additional computation can improve the initial condition fitting. Is there a mechanism in Frozen-PINN to leverage increased computational resources for better initial condition satisfaction?

**Questions:**

>Q1. See W1.


>Q2. See W2.

---

> ### Author Response · Authors · 2025-11-21
>
> We appreciate your valuable time and thorough feedback. We thank you for highlighting many strengths of our work, including clear motivation, solid justification for why it sidesteps the challenges faced by traditional PINNs, and strong empirical performance. We have now carefully addressed your concerns and questions. Our point-by-point responses to the identified weaknesses and questions are provided below.
>
>
> > **W1.1** Frozen-PINN satisfies the initial condition by solving a least squares problem. While this decouples the initial condition from the PDE and boundary conditions, thereby simplifying the optimization, the least squares solution may not always be perfect. An analysis of the residual initial condition loss and its impact on model performance would be valuable.
>
> \--> We appreciate the reviewer's thoughtful question and answer it with the following points.
>
> - **On the challenges with a rigorous theoretical analysis of the residual initial condition loss and its impact on model performance:** Although the least squares solution may not be perfect, by scaling the number of neurons appropriately, we can achieve an arbitrarily small error, provided the initial condition is somewhat well-behaved. More specifically, the sampling followed by least square solve with L2 regularization has probabilistic convergence with bounds described in \[d\]. Nevertheless, there will naturally be an error between the true function and ours. Based on this, there are then two different questions one may ask about the ODE solution. Firstly, consider two ansatz $f_1,f_2$, both being neural networks of the same hidden layer but different outer layers, where the difference between the two functions is $\\delta >0$. Applying an ODE solver to these two functions will propagate the error in a predictable way, as we are applying the same continuous transformation to both of them. Secondly, consider the difference between the ODE solver and the ODE solution itself on a function $f$, where $f$ is a weighted sum of neurons. This is a much more complex question, and we can illustrate this slightly by assuming a linear ODE $L$. To produce bounds on $f$, we need to understand the bound between the two operators $e^{tPL}$ and $Pe^{tL}$, where $P$ is the projection from $L_2$ to the function space spanned by the neurons. That is, when applying the ODE solver, we project down to the function space at each small increment, while the true ODE solution evolves to the specified time $t$ and then projects down. We have attempted to gain a better understanding of this without making too many assumptions about $L$. This is quite challenging, and our efforts have not yet yielded success, which is why we have **included it in the 'Limitations and Future Work' section.**
> - ***New Ablation Study: Empirical analysis of the residual initial condition loss and its impact on model performance:*** Motivated by the reviewer's question, we have ***added a new empirical ablation study*** to investigate this further. For details, please refer to ***Section B.2.1 - Analysis of Residual Initial Condition Loss and Its Impact on Model Performance.***
> - **Summary of Findings:** We present a concise summary of the key findings (excluding details) below for the reviewer's convenience. The least squares solution $C(0)$ is computed using the Python function *np.linalg.lstsq*, which takes as an argument *rcond* representing the cut-off ratio for small singular values of $[\Phi(X),1]$. **We perform an experiment by progressively increasing the cut-off value *rcond* to deliberately degrade the initial-condition fit and study its impact on the overall PDE residual.** We solve the advection equation with an advection coefficient of 40, and vary *rcond* from $10^{-1}$ to $10^{-17}$. ***Figure 9*** shows that small *rcond* values yield highly accurate initial-condition fits and low relative error. As *rcond* increases, more dominant singular values are discarded in the least-squares solve, degrading the initial-condition representation and leading to larger errors in the full space–time solution. Thus, for Frozen-PINNs, **maintaining a reasonably accurate initial condition fit is important, as inaccuracies can influence the ODE solve and increase the overall error.**
> - **Remark:** For all PDEs considered here, the initial condition is relatively easy to fit, and one can approximate it accurately by sampling enough collocation points at t = 0, using enough basis functions, and setting a low regularization constant ($\approx 10^{-12}$). So, **the initial condition loss is not the bottleneck in any of the PDEs we considered here.**

---

> ### Author Response · Authors · 2025-11-21
>
> > **W1.2** More broadly, investigating the sources of different loss components and their effects on accuracy would enhance our understanding of Frozen-PINN's behavior.
>
> \--> We thank the reviewer for this interesting and insightful suggestion, which motivated us to conduct additional experiments to investigate the sources of different loss terms and their impact on the performance of Frozen-PINNs.
>
> - **The conventional sources of error:** Most of the sources that produce errors in satisfying initial conditions, boundary conditions, and PDE residual are quite similar to any neural PDE solver. The important factors influencing the losses are: the number of collocation points, the distribution of collocation points (train and test), the complexity of the PDE solution (smoothness, multi-scale features, and steep gradients), and the number and distribution of basis functions.
> - **Sources of error specific to Frozen-PINNs:**  Frozen-PINNs have **three novel mechanisms/features that differ from traditional PINNs**: (1) **least squares solve** for enforcing initial condition, (2) **augmented ODE** for enforcing boundary conditions, (3) **SVD layer** for reducing the dimensionality and stiffness of the ODE to be solved. Based on the reviewer's suggestion, we have ***added new experiments to quantify the contribution of each of these factors*** to the initial condition, boundary condition, and PDE residual losses, respectively, **to aid practitioners in using Frozen-PINNs and enhance the understanding of our approach.**
>   - **Initial condition loss:** The effect of parameter *rcond* used in the least squares solution is discussed previously in the answer to W1 (See ***Section B.2.1 - Analysis of Residual Initial Condition Loss and Its Impact on Model Performance)***
>   - **Boundary condition loss:**
>             **- Boundary-compliant layer:** For certain boundary conditions, when one can use a boundary-compliant layer (zero Dirichlet boundary condition, periodic boundary condition), the boundary condition loss is zero by construction. The basic intuition is that we construct individual basis functions that satisfy the boundary condition exactly, and thus, the network prediction, which is a linear combination (defined by the output layer parameters) of these basis functions, also satisfies the boundary condition.
>             **- Augmented ODE: (*See Section C.2, ablation studies and Figure 15*): *We have added new experiments to empirically investigate the effect of the penalty term $\\kappa$ in the augmented ODE*** (see Equation 5) on the performance of Frozen-PINNs, considering both accuracy and computation time. Figure 15 shows that: (a) for $\kappa > 10^5$, the boundary loss is negligible (RMSE $< 10^{-10}$) and the total loss is very low (RMSE $\sim 10^{-5} - 10^{-9}$), and (b) for extremely large $\kappa \ge 10^6$, the augmented ODE becomes slightly stiffer, resulting in increased solution time.
>   - **PDE residual loss:** The SVD truncation threshold is a crucial hyperparameter for Frozen-PINNs, determining the dimensionality of the ODE solver and affecting the speed and accuracy of Frozen-PINNs. (See ***Section B.2.1 - On the choice of the SVD truncation threshold and Figure 10***).
> - **Key takeaways for practitioners for Frozen-PINNs based on the empirical findings:**
>   - **Set the default SVD truncation threshold to $10^{-12}$**, which empirically results in a good trade-off between speed and accuracy. Lower the SVD truncation threshold to increase accuracy at the cost of reduced speed and vice versa.
>   - **Set the regularization constant (*rcond*) equal to the SVD truncation threshold** (see Figure 9).
>   - **Set the $\\kappa$ parameter to $10^5$** as a default value. Increase it further to reduce the boundary condition loss at the cost of more computational time in the ODE solver, and decrease it for a faster solution as necessary.

---

> ### Author Response · Authors · 2025-11-21
>
> > **W2.** Despite the various challenges in PINN training, increasing the number of collocation points (combined with appropriate sampling strategies and architectures) generally improves performance when sufficient training is applied. In my understanding, Frozen-PINN could also reduce PDE residual loss by investing more computation to solve the ODEs more accurately. However, since the initial condition relies on a predetermined least squares solution, it is unclear whether additional computation can improve the initial condition fitting. Is there a mechanism in Frozen-PINN to leverage increased computational resources for better initial condition satisfaction?
>
> We thank the reviewer for their thoughtful comments. We address each point below in a point-wise manner.
>
> - **Effect of extra computation on PDE residuals.** We agree with the reviewer that Frozen-PINNs can reduce the PDE residual by allocating more computation to the adaptive ODE solver (e.g., using lower tolerances for the adaptive ODE solvers with step-size control like RK45 [a], LSODA [b] using the *solve_ivp* routine of the *SciPy* package [c]). This improves time-integration accuracy but does not modify how well the initial condition is satisfied.
> - **The ODE solver cannot influence the initial-condition fitting.** The ODE solver only propagates a given initial state forward in time. The accuracy of the ODE solver has no impact on the quality of the initial-condition fit, which is determined solely by the least-squares projection step.
> - **Can additional computation improve IC satisfaction?** In the current Frozen-PINN formulation, the initial condition projection is a closed-form or efficiently solvable least-squares problem. Extra computation helps exactly as the reviewer points out if one enlarges the representation (e.g., increasing basis or network capacity) or samples the IC more densely, in which case the LS fit naturally improves.
> - **Why is this not a practical limitation?** The initial condition is typically known and can be sampled densely. With sufficient samples, low regularization, and enough representation capacity, the least-squares fit is very accurate and not a bottleneck compared to enforcing the PDE.
>
> To conclude, we thank you very much for your time and highly valuable feedback. We sincerely hope that we have satisfactorily answered your questions and further improved our work. We eagerly look forward to your response to our rebuttal and would be more than happy to answer any further questions as soon as possible.
>
> **References:**
>
> [a] Dormand, John R., and Peter J. Prince. "A family of embedded Runge-Kutta formulae." Journal of computational and applied mathematics 6.1 (1980): 19-26.
>
> [b] Petzold, Linda. "Automatic selection of methods for solving stiff and nonstiff systems of ordinary differential equations." SIAM journal on scientific and statistical computing 4.1 (1983): 136-148.
>
> [c] Virtanen, Pauli, et al. "SciPy 1.0: fundamental algorithms for scientific computing in Python." Nature methods 17.3 (2020): 261-272.
>
> \[d\] Generalization Properties of Learning with Random Features, Rudi [et.al](http://et.al)., 2021

---

> ### Comment · Reviewer_iLK9 · 2025-11-25
>
> Thank you for the detailed response. I have no further questions.

---

### Official Review · Reviewer_6QHM · 2025-11-01

**Soundness:** 3
**Presentation:** 3
**Contribution:** 3
**Rating:** 8
**Confidence:** 4

**Summary:**

The paper proposes Frozen-PINN, a physics-informed neural network framework that avoids gradient-descent altogether by freezing space-dependent features and evolving only time-dependent coefficients through ODE solvers, thereby enforcing temporal causality by design. It argues that two root causes of slow/inaccurate PINNs are (i) hard, multi-objective, nonconvex optimization over many parameters and (ii) treating time as just another spatial dimension, and shows that space–time separation plus loss decoupling sidestep both. Frozen-PINNs train orders of magnitude faster, while reaching accuracies comparable to or better than SOTA PINNs and even close to mesh-based solvers in low dimensions. The method also introduces an SVD layer to reduce stiffness and further accelerate solving.

**Strengths:**

1. The paper removes SGD/backprop from the loop by freezing spatial features and evolving only time-dependent coefficients with classical ODE solvers and least squares, which is a non-standard but very clean way to “solve” the training bottleneck of PINNs rather than tuning optimizers.

2. By constructing the solution as frozen spatial bases plus time-evolving coefficients, the method enforces the Markov/causal structure of time-dependent PDEs “by construction,” something many domain-decomposition / curriculum PINNs only approximate with scheduling. That’s an insightful modeling decision with high originality.

3. The 10D–100D heat-equation results and the comparison plot (Frozen-PINN-elm vs. PINN) show the method keeps errors small while increasing width, a case where many neural PDE solvers and all classical meshes struggle; this is important significance-wise for scientific ML.

4. Because training is reduced to classical ODE solves and least squares on frozen bases, the method does not lean on big GPUs or specialized accelerators, which broadens the audience and strengthens the paper’s practical significance.

**Weaknesses:**

1. Missing literatures regarding model-architecture-based temporal PINN methods. In the introduction section, it is claimed that Non-causal treatment of time as an extra spatial dimension. But there are indeed such researches like PINNsFormer[1] and PINNMamba[2] which consider and model the PDE system as a (pseudo-)sequence. Although this doesn't hurt the novelty and significance of this paper, it should be discussed.

[1] Zhao Z, Ding X, Prakash B A. Pinnsformer: A transformer-based framework for physics-informed neural networks. ICLR 2024.

[2] Xu C, Liu D, Hu Y, et al. Sub-sequential physics-informed learning with state space model. ICML 2025.

**Questions:**

You propose two strategies: boundary-compliant layer vs. augmented ODE. How should a practitioner decide between the two for a new PDE/domain? Is there a cost/accuracy tradeoff (e.g. augmented ODE enlarges the system and can worsen stiffness; boundary layer needs a problem-specific A)? A decision table or rule-of-thumb would make this much more usable.

---

> ### Author Response · Authors · 2025-11-21
>
> We appreciate your valuable time and thorough feedback. We thank you for highlighting many strengths of our work, including its clean way to “solve” the training bottleneck of PINNs, insightful modeling decisions, strong convergence in high-dimensional PDEs, and broad practical significance. We have now carefully addressed your concerns and questions. Our point-by-point responses to the identified weaknesses and questions are provided below.
>
> > **W1.** Missing literatures regarding model-architecture-based temporal PINN methods. In the introduction section, it is claimed that Non-causal treatment of time as an extra spatial dimension. But there are indeed such researches like PINNsFormer[1] and PINNMamba[2] which consider and model the PDE system as a (pseudo-)sequence. Although this doesn't hurt the novelty and significance of this paper, it should be discussed.
>
> \--> We thank the reviewer for pointing out recent model-architecture-based temporal PINN approaches such as PINNsFormer \[a\] and PINNMamba \[b\]. Both works introduce temporal inductive biases by treating PDE evolution as a (pseudo-)sequence. ***We have added the following discussion about these works in the updated draft in the Extended Review of Related Work Section in Appendix A.***
>
> Other recent advances of PINNs include methods that model the PDE system as pseudo-sequences. For instance, PINNsFormer employs a Transformer-based architecture that constructs pseudo-sequences from spatio-temporal samples and uses self-attention to model long-range temporal dependencies \[a\]. Another work, PINNMamba, is based on State Space Models (SSMs) and sub-sequence alignment, enabling continuous–discrete temporal modeling and improved propagation of initial-condition information \[b\]. Although these methods model PDE systems as pseudo-sequences, these architectures often lead to increased computational time and out-of-memory issues owing to their architecture, as presented in \[b\].
>
> > **Q1.** You propose two strategies: boundary-compliant layer vs. augmented ODE. How should a practitioner decide between the two for a new PDE/domain? Is there a cost/accuracy tradeoff (e.g. augmented ODE enlarges the system and can worsen stiffness; boundary layer needs a problem-specific A)? A decision table or rule-of-thumb would make this much more usable.
>
> \--> We appreciate the reviewer's suggestion to discuss the rule of thumb for selecting a strategy that satisfies the boundary conditions. Thanks to the reviewers' question, ***we have now added this in Section B.2.1 under the heading "On the choice between the two strategies for enforcing boundary conditions,"*** in the revised version to aid practitioners.
>
> In practice, the choice between the **boundary-compliant layer** and the **augmented ODE** follows a simple cost-benefit tradeoff.
>
> **Boundary-compliant layer (preferred when available):** We recommend using it when a problem-specific transformation $\\phi_A(X)$ (see Section 2.4, B.2.5) is easy to derive (e.g., zero Dirichlet, periodic boundary conditions on simple domains). It **enforces boundary conditions (almost) exactly** and **does not enlarge the ODE system**, so it is typically **more efficient**. The **main limitation is that it requires deriving $\phi_A(X)$**, which may be non-trivial for complex geometries or boundary conditions.
>
> **Augmented ODE (when $\\phi_A(X)$ is hard to construct):** We recommend using this when boundary geometry or constraints make an analytic boundary-compliant mapping difficult. This is **universally applicable** and **requires no problem-specific engineering** since it soft-enforces boundary conditions by augmenting the state, **at the cost of increasing system dimension and possibly worsening stiffness.**
>
> **Rule-of-thumb:**  *Use the boundary-compliant layer whenever a simple mapping $\phi_A$ can be constructed; otherwise, use the augmented ODE, as it is universal and requires no problem-specific engineering.*
>
>
> To conclude, we thank you very much for your time and highly valuable feedback. We sincerely hope that we have satisfactorily answered your questions and further improved our work. We eagerly look forward to your response to our rebuttal and would be more than happy to answer any further questions as soon as possible.
>
> **References:**
>
> [a] Zhao Z, Ding X, Prakash B A. Pinnsformer: A transformer-based framework for physics-informed neural networks. ICLR 2024.
>
> [b] Xu C, Liu D, Hu Y, et al. Sub-sequential physics-informed learning with a state space model. ICML 2025.

---

> > ### Comment · Reviewer_6QHM · 2025-11-22
> >
> > Thank you for answering the question. All my concerns have been addressed. I increased my rating to 10.

---

> > > ### Author Response · Authors · 2025-11-22
> > >
> > > We are pleased to have addressed all of your concerns. Thank you for recognizing the contributions of our work and for increasing your score. We sincerely appreciate your time and thoughtful feedback.

---

### Official Review · Reviewer_19cn · 2025-11-07

**Soundness:** 4
**Presentation:** 4
**Contribution:** 4
**Rating:** 8
**Confidence:** 3

**Summary:**

The paper proposes Frozen-PINN, a gradient-descent-free approach for time-dependent PDEs that enforces temporal causality by separating space and time. Hidden-layer spatial bases (weights and biases) are sampled and then frozen (via either data-agnostic ELM sampling or data-dependent SWIM sampling), while the time-dependent output coefficients are evolved by solving an ODE obtained by inserting the frozen spatial basis into the PDE. Initial coefficients are set by least squares and boundary conditions are handled via either a boundary-compliant layer or an augmented ODE. The method further compresses the neural bases with a truncated SVD layer to reduce ODE dimension and accelerate computation. Empirically, the authors evaluate Frozen-PINN on eight PDE benchmarks and show improved accuracy compared to many PINN variants and classical mesh solvers in the studied regimes. The manuscript additionally provides ablation studies (e.g., on the SVD layer and SWIM vs ELM sampling), implementation details, and a reproducibility statement.

**Strengths:**

1. The paper introduces a clear and intuitively appealing space–time separation idea: freezing spatial basis functions and evolving only the time-dependent output layer via ODE solvers, which enforces temporal causality by construction and directly addresses a known PINN weakness (treating time as an extra spatial dimension).

2. The manuscript presents two pragmatic sampling strategies for spatial bases: (i) data-agnostic Extreme Learning Machine (ELM) sampling and (ii) data-dependent SWIM sampling (Sample-Where-It-Matters), where SWIM uses pairs of collocation points (and optional projection onto solution gradients) to place basis functions near shocks or informative regions. This design is well motivated and demonstrated to be useful for handling shocks and directional features.

3. The authors implement an SVD compression layer to orthogonalize and truncate the neural basis, substantially reducing the dimension of the resulting ODE system and lowering computational cost; the paper supplies ablation evidence that the SVD layer can yield very large runtime improvements in some settings (e.g., tens of speedups in several ablations).

4. The experimental evaluation is broad and systematic within the chosen set of PDEs: eight benchmarks spanning advection extremes, shocks, higher-order spatial derivatives, complex geometries, and high dimensionality are considered, and the authors report means and standard deviations over multiple seeds with numerous ablations (resampling, basis widths, SVD thresholds). This breadth strengthens the empirical narrative.

5. The method removes reliance on iterative gradient-descent training for the bulk of parameters, enabling very fast CPU training in many reported cases and promising a low-resource alternative to GPU-centric PINN training; the authors emphasize carbon-efficiency and implementation simplicity as benefits.

6. The paper is reasonably reproducible in spirit: many methodological details, hyperparameter settings, and appendices are provided, and the authors pledge to release code and seeds to reproduce experiments.

**Weaknesses:**

1. Regarding the experimental claims of extreme speedups and accuracy (e.g., “up to 100,000× faster training” and multiple orders-of-magnitude accuracy gains), it should be carefully noted that these comparisons are across different CPU vs GPU settings (CPUs for Frozen PINNs vs GPUs for PINNs), and different solver regimes (low-precision vs high-precision). The paper does acknowledge benchmarking rules (matching accuracy regimes and noting GPU timings), but the mixed hardware/configurations needs to be specified while making broad claims of speedups and accuracy (which could be attributed to the differences in configurations too).

2. The scope of theoretical guarantees and assumptions is limited in interpretability for practitioners. While the paper references and partially justifies why random-feature style bases and ODE evolution reduce optimization complexity, it lacks clear, practically checkable conditions that explain when SWIM/ELM sampling will produce sufficiently expressive bases (e.g., how many bases are needed as a function of Kolmogorov n-width or solution regularity). The theoretical framing would benefit from tighter bounds or guidance connecting basis counts, SVD truncation thresholds, and expected approximation errors in realistic regimes.

3. The overhead and robustness of repeated resampling in high dimensions or with sparse data should be quantified and discussed more explicitly. Some ablations show SVD reduces runtime dramatically, but the cost of SWIM itself (and its scaling w.r.t. dimension and collocation count) needs clearer accounting.

4. The generalization to fully spatially complex PDEs (e.g., Navier–Stokes turbulence or domains requiring domain decomposition) is asserted but not demonstrated. Claims about matching mesh solvers in low-dimensions and scaling to high dimensions are promising but would be stronger with at least one medium-scale 2D/3D fluid example or a domain-decomposition demonstration.

5. The method’s dependence on choices of basis sampling (ELM vs SWIM), SVD cutoff, and collocation density appears substantial from the ablations, but sensitivity studies across a broader hyperparameter grid and more seeds would increase confidence in robustness and reproducibility. Some ablations show very large speedups only after aggressive SVD compression. This raises questions about stability and about how to choose compression thresholds in new problems.

**Questions:**

1. What is the computational cost and memory scaling of SWIM resampling and basis projection in high dimensions (d≥10)? Are there practical heuristics (e.g., maximum number of collocation points, subsampling strategies) you recommend to keep SWIM tractable?

2. Could you provide more explicit guidance (or an automated procedure) for choosing the SVD cutoff/target rank so a new practitioner can reliably obtain the claimed speedups without extensive manual tuning? Please include how SVD choice impacts stability of the ODE solver.

3. For PDEs with truly complex spatial structure (e.g., 2D Navier–Stokes), do you expect Frozen-PINN to require domain decomposition or local basis strategies? Do you have preliminary results or runtimes/cost models for such extensions?

4. It will be interesting to see efficiency comparisons with numerical methods for solving PDEs.

---

> ### Author Response · Authors · 2025-11-21
>
> We appreciate your valuable time and thorough feedback. We thank you for highlighting many strengths of our work, including the clear and intuitively appealing idea, broad and systematic empirical evaluation, carbon efficiency, and low hardware requirements. We have carefully implemented your suggestions through new experiments and attempted to address your concerns and questions to the best of our ability. Our point-by-point responses to the identified weaknesses and questions are provided below.
>
>
> > **W1.** Regarding the experimental claims of extreme speedups and accuracy (e.g., “up to 100,000× faster training” and multiple orders-of-magnitude accuracy gains), it should be carefully noted that these comparisons are across different CPU vs GPU settings (CPUs for Frozen PINNs vs GPUs for PINNs), and different solver regimes (low-precision vs high-precision). The paper does acknowledge benchmarking rules (matching accuracy regimes and noting GPU timings), but the mixed hardware/configurations needs to be specified while making broad claims of speedups and accuracy (which could be attributed to the differences in configurations too).
>
> \--> We thank the reviewer for the helpful suggestion and for acknowledging our benchmarking rules. The original *up to 100,000x* statement refers specifically to the CPU-time comparison on identical hardware for the Euler-Bernoulli PDE, which is 95,280 times faster (see Table 1). Nonetheless, we have revised the manuscript to use the more conservative and clearer phrasing: *up to four to five orders of magnitude faster training.*
>
>
> > **W2.** The scope of theoretical guarantees and assumptions is limited in interpretability for practitioners. While the paper references and partially justifies why random-feature style bases and ODE evolution reduce optimization complexity, it lacks clear, practically checkable conditions that explain when SWIM/ELM sampling will produce sufficiently expressive bases (e.g., how many bases are needed as a function of Kolmogorov n-width or solution regularity). The theoretical framing would benefit from tighter bounds or guidance connecting basis counts, SVD truncation thresholds, and expected approximation errors in realistic regimes.
>
> \--> We agree with the reviewer's assessment regarding theoretical guarantees, which is why we have also included it in the' Limitations and Future Work' section. The theoretical guarantees for random features as a basis are generally strong, with good probabilistic bounds for a very large class of functions \[a\]. These results also include the use of the least squares solution with L2 regularization, providing convergence results for the algorithm in both terms of sampling and solving the linear layer, which is significantly stronger than what one can usually expect when using backpropagation.
>
> The difficulty comes when introducing the ODE solver, as we can generally give guarantees with respect to the initial condition (which is already provided in the literature). The problem of furthering understanding is to understand the difference between the true solution of the ODE at time $t$ and the solution provided by the ODE solver. If the trajectory from $t=0$ to $t=T$ is all inside the span of the neurons, there is no problem, and one can rely on classical results for ODE solvers to produce bounds. This is unrealistic to expect, and it is here that the difficulties arise regarding theoretical guarantees. To illustrate this somewhat, we assume a linear ODE $L$. Let $f$ be a function in the span of the neurons. To produce bounds in general, we need to understand the difference between the two operators $Pe^{tL}$ and $ e^{tPL}$, where $P$ is the projection from $L_2$ to the span of the sampled neurons. That is, the operator related to the ODE solver projects down to the function space at each small increment. In contrast, the operator related to the true ODE solution evolves to the specified time $t$ and then projects down. If this can be bounded, one can give theoretical guarantees in general. We attempted to bound these two operators without making too many stringent assumptions about $L$, but this proved quite challenging and has yet to yield success. We have left it as future work for now.

---

> ### Author Response · Authors · 2025-11-21
>
> > **W3.** The overhead and robustness of repeated resampling in high dimensions or with sparse data should be quantified and discussed more explicitly. Some ablations show SVD reduces runtime dramatically, but the cost of SWIM itself (and its scaling w.r.t. dimension and collocation count) needs clearer accounting.
>
> \--> Although the total runtime is discussed and compared to regular PINNs, we do agree that more time could be spent on the cost of SWIM itself. The original authors of SWIM provide some insight into this, but we would also like to offer further insight into our thinking on the matter.
>
> The primary concern from a computational standpoint of SWIM boils down to the linear solver complexity, which is cubic (in computation time) with respect to the number of neurons, as the sampling itself is very quick. We know from several results that the number of neurons needed for many functions is much smaller than $N$. In the literature for random features, it is supported that the number of neurons needed is on the scale of $N^{1/2}log N^{1/2} << N$ \[a\]. This means that our method scales cubically with the number of neurons and linearly with the number of data points, and that the number of neurons required is much less than the number of data points. Hence, there are good reasons to believe that the scaling is relatively good when the dimension increases, something we observed in Figure 6. This is not to say our method will not suffer from the curse of dimensionality at all, but we can expect it to do comparatively well.
>
> > **W4.** The generalization to fully spatially complex PDEs (e.g., Navier–Stokes turbulence or domains requiring domain decomposition) is asserted but not demonstrated. Claims about matching mesh solvers in low-dimensions and scaling to high dimensions are promising but would be stronger with at least one medium-scale 2D/3D fluid example or a domain-decomposition demonstration.
>
> \--> Our point-wise response is given below.
>
> - **Regarding the claims:** We agree with the reviewer that considering PDE benchmarks with added spatial complexity would be beneficial. However, **we do not assert that we can generalize to Navier-Stokes; we merely mention in the Limitations and future work section that:**  *solving PDEs like Navier–Stokes is an exciting next step, where one could leverage domain decomposition to deal with the added complexity \[a,f\].* We also fully agree with the reviewer that claims about matching mesh solvers in low dimensions and scaling to high dimensions would be stronger with more complicated examples. In this work, **we make these claims solely based on our systematic empirical study, and we do not intend to make any broader claims beyond those presented here. We have rephrased the corresponding text in the conclusion section accordingly.**
> - **Ongoing Work with Multilevel Schwarz–Inspired Methods:** We agree that testing on highly nonlinear and multiscale systems (e.g., the Navier–Stokes equations) would further validate the robustness. However, **for such problems, combining Frozen-PINNs with domain-decomposition approaches is essential**, especially given the strong performance of finite-basis PINNs and multi-level FB-PINNs \[b, c\]. We are already pursuing this direction using multilevel Schwarz–inspired domain decomposition \[c\]. **Preliminary results on static PDEs show that Frozen-PINNs can match the accuracy of multi-level FB-PINNs while training significantly faster.** These results are not included here, as the paper focuses on time-dependent PDEs.
> - **Future Extensions:** Extending Frozen-PINNs to time-dependent PDEs and 2D/3D fluid flows is a natural next step, aligning with recent progress in \[d\], but requires effort beyond the scope of this rebuttal.
> - **New Benchmark Added: Wave Equation (Hao et al. \[e\])**: Motivated by the reviewer’s comment, we have added a new benchmark from Hao et al. \[e\] for the wave equation (Sec C.3). This benchmark contains complex, multi-scale solution behavior. We demonstrate that CPU-trained Frozen-PINNs achieve $625$ to $5500$ times faster training than GPU-trained competing PINN variants (vanilla PINN, PINN-NTK, FB-PINN, gPINN) while simultaneously being four to five orders of magnitude higher in accuracy on the wave equation (See Table 1 and Fig. 18).
> - **Additional New Benchmark: Multi-Scale Wave Equation:** To further stress-test the robustness and generality of Frozen-PINNs, we introduce an **even more challenging multi-scale wave-equation benchmark (Sec. C.3, Fig. 19)**. **Frozen-PINNs solve this problem extremely quickly and with high precision**, reinforcing their potential for PDEs with complex, multiscale dynamics.
> - **Additional New Benchmark: Kuramoto-Sivashinsky (KS) Equation:** We tackle the **highly nonlinear KS equation exhibiting spatiotemporal chaos and significant spatial complexity**, and show that Frozen-PINNs capture the characteristic chaotic pattern over a long-time horizon (See Fig. 6).

---

> ### Author Response · Authors · 2025-11-21
>
> > **W5.1** The method’s dependence on choices of basis sampling (ELM vs SWIM), SVD cutoff, and collocation density appears substantial from the ablations, but sensitivity studies across a broader hyperparameter grid and more seeds would increase confidence in robustness and reproducibility.
>
> \-->  We thank the reviewer for this question. We analyze the dependence on ELM versus SWIM sampling, as well as the influence of randomness, in detail i**n Section B.2.1**, under the subsections **"Comparison between Frozen-PINN-swim and Frozen-PINN-elm"** and **"Influence of random sampling on the method.**" These studies consistently show that **both sampling strategies yield stable performance, with SWIM providing slightly lower variance across seeds.**
>
> >**W5.2** Some ablations show very large speedups only after aggressive SVD compression. This raises questions about stability and about how to choose compression thresholds in new problems.
>
> \--> In response to the reviewer's concern regarding SVD truncation, ***we have added new ablation experiments that explicitly quantify the impact of different truncation thresholds.*** For details, please refer to the ***new experiments in Section B.2.1*** concerning the extended discussion on Frozen-PINNs, with the bullet point titled ***\`\`On the choice of SVD truncation threshold''***.
>
> To summarize, Figure 9 illustrates the well-known **trade-off between accuracy and speed, as determined by the SVD truncation threshold.** Retaining fewer singular values reduces the dimensionality and stiffness of the ODE, yielding faster solutions with similar or slightly lower accuracy. Retaining more singular values increases dimensionality and stiffness, which slows the solver but improves accuracy. Importantly, **the performance is robust for SVD truncation thresholds $\epsilon\_{SVD} < 10^{-10}$ in all cases.**
>
> **We recommend setting the default SVD truncation threshold to $10^{-12}$**, which empirically yields a **good trade-off between speed and accuracy across all benchmarks**. One can then further adjust the threshold to navigate the speed-accuracy trade-off as needed.
>
> > **Q1.1** What is the computational cost and memory scaling of SWIM resampling and basis projection in high dimensions (d≥10)?
>
> \--> Please refer to our answer to Weakness 3.
>
> > **Q1.2** Are there practical heuristics (e.g., maximum number of collocation points, subsampling strategies) you recommend to keep SWIM tractable?
>
> \--> The number of collocation points (or even network capacity) highly depends on the underlying solution (i.e., how smooth and spatially complex the features are) and the speed and accuracy requirements. Thus, it is not generally possible to define practical heuristics that will work universally. For instance, when solving the Burgers' equation using Frozen-PINN-swim in a high-precision regime (see Table 1), we evaluate the solution gradient at 20,000 points (ensuring sufficient sampling at the shock locations), which is then used to subsample 1,000 collocation points according to a probability distribution proportional to the solution gradient (see Section 3.4). In the low-precision regime, we only evaluate the gradient at 1000 points to subsample 600 collocation points.
>
> > **Q2.** Could you provide more explicit guidance (or an automated procedure) for choosing the SVD cutoff/target rank so a new practitioner can reliably obtain the claimed speedups without extensive manual tuning? Please include how SVD choice impacts stability of the ODE solver.
>
> \--> Please refer to our answer to W5.2.
>
> > **Q3.** For PDEs with truly complex spatial structure (e.g., 2D Navier–Stokes), do you expect Frozen-PINN to require domain decomposition or local basis strategies? Do you have preliminary results or runtimes/cost models for such extensions?
>
> \--> Please refer to our answer for W4.

---

> ### Author Response · Authors · 2025-11-21
>
> > **Q4.** It will be interesting to see efficiency comparisons with numerical methods for solving PDEs.
>
> \--> We thank the reviewer for the comment. We address this point-wise below.
>
> - **Value of Comparisons with numerical methods:** We completely agree with the reviewer. Comparing neural PDE solvers with classical methods is important and promotes cross-fertilization between scientific computing and machine learning, despite the obvious hurdles in ensuring a fair comparison due to the differences in approaches. Thus, **we compare our approach with classical FEM and Iso-Geometric Analysis FEM methods (see Table 1, Section B.3).**
> - **Fair Benchmarking Protocol:** We benchmark at **(almost) equal accuracy** and against **efficient numerical baselines** (IGA-FEM, FEM), following the two key guidelines for a fair comparison outlined by McGreivy et al. (2024). The results are shown in **Table 1**. Additionally, using the **Burgers’ equation** benchmark (**see Figures 22 and 23, and Appendix C.4.1 for details**), we demonstrate that Frozen-PINN-swim basis functions accurately resolve shocks, **outperforming traditional bases (Fourier, Chebyshev, ELM) in both convergence rate and accuracy**. For the same accuracy, **SWIM requires far fewer basis functions**.
> - **Performance/efficiency comparisons:** For some low-dimensional PDE benchmarks considered here (Advection, Euler Bernoulli (classical), Euler Bernoulli (Winkler)), IGA-FEM outperforms all neural PDE solvers in terms of speed and accuracy by roughly one order of magnitude. For PDEs with shocks and complicated geometries, Frozen-PINNs achieve **14.5× (Burgers)** and **4.83× (nonlinear diffusion)** speedups at equal accuracy compared to IGA-FEM/FEM.
> - **Overall Takeaway:** **Frozen-PINNs** (and neural PDE solvers, in general) **are important computational methods complementing the classical PDE solvers**. Our experiments demonstrate that each method has its pros and cons. **No method is universally better than the others.**
>
> To conclude, we thank you very much for your time and highly valuable feedback. We sincerely hope that we have satisfactorily answered your questions and further improved our work. We eagerly look forward to your response to our rebuttal and would be more than happy to answer any further questions as soon as possible.
>
> **References:**
>
> [a] Generalization Properties of Learning with Random Features, Rudi [et.al](http://et.al)., 2021
>
> \[b\] Moseley, Ben, Andrew Markham, and Tarje Nissen-Meyer. "Finite basis physics-informed neural networks (FBPINNs): a scalable domain decomposition approach for solving differential equations." Advances in Computational Mathematics 49.4 (2023): 62.
>
> \[c\] Dolean, Victorita, et al. "Multilevel domain decomposition-based architectures for physics-informed neural networks." Computer Methods in Applied Mechanics and Engineering 429 (2024): 117116.
>
> \[d\] Wang, Sifan, et al. "Simulating three-dimensional turbulence with physics-informed neural networks." *arXiv preprint arXiv:2507.08972* (2025).
>
> \[e\] Zhongkai, Hao, et al. "Pinnacle: A comprehensive benchmark of physics-informed neural networks for solving pdes." *Advances in Neural Information Processing Systems* 37 (2024): 76721-76774.
>
> \[f\] Howard, Amanda A., et al. "Finite basis kolmogorov-arnold networks: domain decomposition for data-driven and physics-informed problems." *arXiv preprint arXiv:2406.19662* (2024).

---

### Author Response · Authors · 2025-12-03
**Summary for Area Chair (Part 1: Key Reviewer Questions and Author Responses)**

Dear AC and Reviewers,

We sincerely appreciate the time and effort you have dedicated to reviewing our manuscript. We fully understand the additional workload and challenges brought about by the current situation, and we truly appreciate your continued efforts. To facilitate your decision-making process, we would like to provide a summary of the discussion phase.

### **Summary of key reviewer concerns and author responses:**

**Reviewer 19cn**:

- Primarily raised questions regarding: (a) the scope of theoretical guarantees and assumptions and computational cost of SWIM resampling in high dimensions, (b) explicit guidance for choosing the SVD cutoff for the SVD layer, (c) PDEs with truly complex spatial structure
- We addressed these concerns as follows: (a) we have **provided a preliminary cost analysis of SWIM re-sampling** and clarified the precise difficulties that make deriving tight error bounds non-trivial (b) we have added **new ablation experiments that explicitly quantify the impact of different SVD truncation thresholds and provide concrete guidance on setting the SVD threshold** (Section B.2.1), (c) we have **added multiple new benchmarks** **for the wave equation** and **Kuramoto-Sivashinsky** **equation** (Sec C.3, C.7) exhibiting complex, multi-scale solution behavior.

**Reviewer 6QHM**

- Primarily raised questions regarding: (a) missing literature regarding model-architecture-based temporal PINN methods, and (b) the choice between two proposed strategies for enforcing boundary conditions: boundary-compliant layer vs. augmented ODE.
- We addressed both points by **expanding the discussion of relevant literature** (Section A) and **clarifying the trade-offs between the two boundary-condition strategies** (Section B.2.1).

**Reviewer iLK9:**

- Primarily raised questions regarding: (a) an analysis of the residual initial condition loss and its impact on model performance, (b) investigating the sources of different loss components and their effects on accuracy, and (c) the mechanism in Frozen-PINNs to leverage increased computational resources for better initial condition satisfaction.
- We addressed these concerns by: (a) adding **an empirical analysis of the residual initial condition loss and its impact on model performance** (Section B.2.1), (b) **adding new experiments to quantify the sources of different loss components** (Section B.2.1), and (c) clarifying the effect of extra computation to improve initial condition loss.

**Reviewer Ukoa:**

- Primarily raised questions regarding: (a) possible limitations on the method’s “learning” component, (b) unclear conceptual novelty in combining PINNs with classical ODE solvers and whether this offers advantages over classical PDE solvers, (c) applicability to PDEs with complex and multi-scale dynamics, (d) justification of the shift in focus from classical gradient-descent-based training vs gradient-descent-free training, (e) transferability of insights to other neural operator or physics-informed frameworks, and (f) missing ablations on SVD truncation threshold and L2 regularization.
- We addressed the points as follows: (a) we argue that the learning component of the method is **not inherently limited** by **showing clear advantages compared to classical spectral PDE solvers** for Burgers' equation (with solution-aware resampling of spatial basis functions) and non-linear diffusion equation (with direction-aware sampling of basis functions), (b) arguing that the main **cenceptual novelty** lies in **leveraging tools from scientific computing and random feature methods** (space-time separation, ODE solvers) **to train PINNs (construct PINN parameters)**, alleviating some of the long-standing and key optimization (training) issues of PINNs (as appreciated by all other reviewers), (c) **adding multiple new benchmarks** **for the wave equation** and **Kuramoto-Sivashinsky** **equation** (Sec C.3, C.7) exhibiting complex, multi-scale solution behavior, (d) justifying the change in focus towards gradient-descent-free training by demonstrating **substantial benefits across** **nine PDE benchmarks**: **major speed-ups**, **significant accuracy gains**, **respecting temporal causality by construction**, and **reduced dependence** **on specialized hardware** like GPUs, (e) discussing how **key ideas in Frozen-PINNs are have already synergized with broader frameworks to accelerate training** of neural operator frameworks like POD-DeepONet and Fourier Neural Operator, Hamiltonian neural networks and Hamiltonian graph neural networks, and the current status of our ongoing work on integrating Frozen-PINNs with domain decomposition-based frameworks: FB-PINNs and multi-level FB-PINNs to solve PDEs with multi-scale solutions, (f) **adding ablation studies on SVD truncation threshold and L2 regularization** (Figure 9, Figure 10, Section B.2.1).

---

### Author Response · Authors · 2025-12-03
**Summary for Area Chair (Part 2: Summary of Key Revisions)**

.... (continued from summary 1)

### **Summary of key revisions:**

The initial scores at the beginning of the rebuttal phase (before the reviewers’ later improvements) were **8, 8, 8, 4**. We have carefully incorporated all revisions and additional clarifications provided during the discussion into the updated manuscript, with **changes highlighted in blue** for your convenience. In particular, we have **added the following new experiments during the rebuttal phase:**

1. **Empirical analysis quantifying the effect of Frozen-PINN hyperparameters on different loss components (Section B.2.1) and practical tips on setting those,**
2. **Empirical analysis of the residual initial condition loss and its impact on performance (Section B.2.1),**
3. **Benchmark on the Wave equation (Sec. C.3, Table 1, Figures 18 and 19),**
4. **Benchmark on the Kuramoto-Sivashinsky equation (Sec. C.7, Figures 6 and 28),**
5. **Ablation for SVD threshold (Figure 10),**
6. **Empirical analysis of L-2 regularization (Figure 9),**
7. **Ablations for PINN with varying numbers of layers for deeper networks (Tables 6 and 7).**

We sincerely believe that we have fully addressed all reviewer questions, including those from reviewers who were unable to participate in the discussion, and that the paper has improved substantially as a result. We trust your judgment in the final decision and hope this summary supports your reevaluation. Thank you for your time and consideration.

---

### Meta-Review · Area_Chair_Rgrm · 2026-01-06

**Summary:**

The paper proposes "Frozen-PINN," a novel framework for solving time-dependent Partial Differential Equations (PDEs) without relying on gradient descent-based optimization. The core idea utilizes space-time separation: spatial dependencies are modeled using fixed (frozen) random feature bases (via ELM or SWIM sampling), while the temporal evolution is handled by solving the resulting system of Ordinary Differential Equations (ODEs) using classical solvers.

The reviewers generally appreciated the paper's significant contribution to addressing the long-standing bottlenecks of training speed and temporal causality in PINNs. Three out of four reviewers (19cn, 6QHM, iLK9) were highly positive, citing the "extreme efficiency" (speedups of several orders of magnitude), the enforcement of causality by construction, and the method's ability to handle high-dimensional problems. One reviewer (Ukoa) initially raised concerns regarding the conceptual novelty (viewing it as closer to classical numerical methods than deep learning) and the strength of baselines.

The authors provided a comprehensive rebuttal, adding significant new experiments (Wave equation, Kuramoto-Sivashinsky equation, ablations on deep PINNs and SVD thresholds) and clarifying the comparison metrics. Based on the strong empirical results and the effective synthesis of random feature methods with scientific computing, I recommend acceptance.

**Reviewer Concerns:**

Theoretical Guarantees: Reviewer 19cn noted the lack of tight theoretical bounds, specifically regarding the error propagation when combining random feature approximation with ODE solvers. The authors acknowledged this difficulty and framed it as future work. While an open problem, this does not diminish the empirical value of the current work.

Conceptual "Learning" Component: Reviewer Ukoa's fundamental concern that the method "shifts focus" away from end-to-end learning to a hybrid numerical approach remains a philosophical difference.

**Reviewer Scores:**

Reviewer 6QHM: 8 (Accept) -> 10 (Strong Accept). Reasoning: The reviewer explicitly raised their score after the authors successfully addressed concerns regarding missing literature (PINNsFormer) and provided a clear decision rule for boundary condition strategies.

Reviewer 19cn: 8 (Accept) -> 8 (Accept). Reasoning: The reviewer was positive from the start. The rebuttal addressed the call for more complex benchmarks (Wave/KS equations), reinforcing the initial positive assessment, though theoretical concerns remain as future work.

Reviewer iLK9: 8 (Accept) -> 8 (Accept). Reasoning: The reviewer explicitly stated "I have no further questions" after the rebuttal provided a satisfactory analysis of the Initial Condition residual loss and its impact.

Reviewer Ukoa: 4 (Reject) -> 5 (Weak Accept). Reasoning: While the reviewer questioned the conceptual novelty ("limited learning component"), the rebuttal objectively addressed the factual criticisms regarding weak baselines and simple benchmarks. The strong empirical evidence warrants a weak accept despite the philosophical differences.

---

### Decision · Program_Chairs · 2026-01-26

Accept (Oral)